# Site-specific ubiquitylation acts as a regulator of linker histone H1

Eva Höllmüller[1,2,3,4], Simon Geigges[1,3,4], Marie L. Niedermeier[2,3], Kai-Michael Kammer[2,3], Simon M. Kienle[2,3], Daniel Rösner[1,3], Martin Scheffner [2,3✉], Andreas Marx [1,3✉] & Florian Stengel [2,3✉]

Decoding the role of histone posttranslational modifications (PTMs) is key to understand the fundamental process of epigenetic regulation. This is well studied for PTMs of core histones but not for linker histone H1 in general and its ubiquitylation in particular due to a lack of proper tools. Here, we report on the chemical synthesis of site-specifically mono-ubiquity-lated H1.2 and identify its ubiquitin-dependent interactome on a proteome-wide scale. We show that site-specific ubiquitylation of H1 at position K64 modulates interactions with deubiquitylating enzymes and the deacetylase *SIRT1*. Moreover, it affects H1-dependent chromatosome assembly and phase separation resulting in a more open chromatosome conformation generally associated with a transcriptionally active chromatin state. In summary, we propose that site-specific ubiquitylation plays a general regulatory role for linker histone H1.

[1] Department of Chemistry, University of Konstanz, Konstanz, Germany. [2] Department of Biology, University of Konstanz, Konstanz, Germany. [3] Konstanz Research School Chemical Biology, University of Konstanz, Konstanz, Germany. [4]These authors contributed equally: Eva Höllmüller, Simon Geigges. ✉email: martin.scheffner@uni-konstanz.de; andreas.marx@uni-konstanz.de; florian.stengel@uni-konstanz.de

In eukaryotic cells, genetic information is tightly packaged and organized in a nucleoprotein complex termed chromatin. Nucleosomes form the basic repeating structural units of chromatin and consist of DNA wrapped around an octamer of the four core histones (H2A, H2B, H3, and H4). The linker histone H1 additionally binds at the nucleosome entry and exit sites in a dynamic manner to form higher-order chromatin structures. Thereby, the small basic protein stabilizes the nucleosomes and provides the structural and functional flexibility of chromatin. Human cells contain eleven variants of linker histone H1, including seven somatic subtypes (H1.1 to H1.5, H1.0, and H1x). The subtypes H1.1 to H1.5 and H1x are ubiquitously expressed in almost every cell type, with H1.2 and H1.4 being the predominant variants in most human cells. Linker histones of higher eukaryotes have a tripartite structure. They consist of a central structured and highly conserved globular domain (GD, 70–80 amino acids) flanked by unstructured N-terminal (N-terminal domain, NTD; ~40 amino acids) and C-terminal (C-terminal domain, CTD; ~100 amino acids) tails, which exhibit significant sequence divergence within the same species[1–5]. Biomolecular condensate formation by liquid-liquid phase separation (LLPS) leads to local enrichment of proteins and is increasingly recognized as a general mechanism of how cells can organize biochemical reactions in time and space[6,7]. Recent studies indicate that LLPS plays a role in chromatin maintenance and chromatin organization, even though the extent to which these processes are affected by LLPS in cells requires further investigations[7–10].

Histones are intensely post-translationally modified resulting in a complex chemical language, the histone code, which is interpreted by specific protein complexes and enzymes to mediate transcriptional responses and downstream functions[11–16]. While the functional relevance of specific histone marks is increasingly well understood for core histones, this is not the case for linker histone H1. Unlike the core histones, H1 binds dynamically to chromosomes and plays a fundamental role in the formation of higher-order chromatin. Yet, although H1 is closely linked to the regulation of DNA structure and dynamics[1,3–5], a lack of appropriate technologies, in particular the absence of site-specific and modification-specific antibodies, has handicapped research on H1 and has significantly hampered our ability to decipher its contribution to the histone code. Moreover, intrinsic characteristics have even hindered in vitro analyses of modified H1 following recombinant expression. In particular, the long, highly unstructured and lysine-rich CTD is prone to degradation and yields insoluble and truncated proteins[17]. Yet, there is growing evidence that linker histone H1 regulates cellular functions by direct protein-protein interactions[18–21]. Furthermore, ubiquitylation of H1 was linked to activation of gene expression[22] and antiviral protection[23], and more recently H1 ubiquitylation has also been put forward as a histone mark relevant in the response to DNA damage[24,25].

The combination of chemical methods to modify histones together with mass spectrometry (MS) has been proven powerful in identifying the relevant writers, readers, and erasers for specific (core) histone-PTMs[15,26–28], as chemical protein synthesis allows for the generation of histones of defined, homogeneous modification states[15,29,30]. Methods include cysteine bioconjugation[31,32], protein semisynthesis, such as native chemical ligation[33,34], expressed protein ligation[35] or sortase-mediated ligation[36], as well as genetic code expansion[37,38]. However, the generation of ubiquitylated H1 poses unique chemical challenges due to both the size of this PTM and the unstructured, highly basic nature of H1 itself. Based on genetic code expansion by stop codon suppression followed by Cu(I)-catalyzed azide-alkyne cycloaddition (CuAAC) (click reaction), we and others[39,40] established an efficient approach for the generation of ubiquitylated conjugates. The bioorthogonal click reaction is a stereospecific 1,3-dipolar Huisgen cycloaddition between an azide and a terminal alkyne, forming a 1,4-disubstituted 1,2,3-triazole. It mimics the natural isopeptide bond by similar electronic and topological features[41] while resisting proteolytic cleavage[42]. The respective chemical functionalities can be incorporated using unnatural amino acids, resulting in the site-specific ubiquitylation of the target protein[43].

Here, we have adapted and critically expanded this approach allowing us to generate pure, non-hydrolyzable, mono-ubiquitylated H1 variants in sufficient quantities to study the consequences of site-specific ubiquitylation by proteomic profiling and biochemical approaches. We used the H1.2 ubiquitin (Ub) conjugates to identify and comprehensively characterize the ubiquitylation-dependent cellular interactome for H1.2 and show that site-specific ubiquitylation results in overlapping, but distinct interactomes. We then moved on to functionally probe position-dependent biochemical effects of H1 ubiquitylation in more detail and demonstrate that site-specific ubiquitylation of H1 modulates and regulates interactions with enzymes relevant for deubiquitylation and deacetylation. We finally show that ubiquitylation at position K64 functionally impacts both H1-induced phase separation and chromatosome assembly. In summary, we established the combination of chemical synthesis of Ub-conjugates by CuAAC and proteomic profiling as a powerful approach to study histone marks for linker histone H1 and demonstrated that site-specific ubiquitylation is an important contributor in the shaping of the H1 interactome and acts as a general modulator of H1 function.

## Results

**Generation of site-specifically ubiquitylated H1.2.** Site-specifically ubiquitylated H1.2 was generated by CuAAC as illustrated in Fig. 1a. To obtain alkyne-functionalized H1, the pyrrolysine (Pyl) analog propargyl-derivatized lysine (Plk) was incorporated into H1.2 by amber stop codon suppression using the orthogonal Pyl tRNA synthetase (PylRS)/tRNA$^{Pyl}$ pair of *Methanosarcina barkeri*[44]. Thereby, H1.2 variants were generated harboring Plk instead of lysine at distinct positions (H1.2 KxPlk)[39,45]. To investigate the effects of site-specific ubiquitylation of H1, we generated three variants containing Plk in the proteins' three structural domains: the NTD (H1.2 K17Plk), the GD (H1.2 K64Plk), and the CTD (H1.2 K206Plk). These positions had been identified to be ubiquitylated in several previous studies[46–51] (note that ubiquitylation of endogenous H1.2 at positions K17 and K64 was confirmed in HEK 293T cells; Supplementary Fig. 1 and Supplementary Table 1). The incorporation of Plk was verified by MS and the correct structural fold of the H1.2 KxPlk variants was validated by circular dichroism (CD) spectroscopy (Supplementary Fig. 2a-c). In addition, H1.2 was equipped with an N-terminal Strep-tag II for affinity purification. Azide-functionalized Ub was generated by the incorporation of the methionine analog azidohomoalanine (Aha) at the C-terminal position (Ub G76Aha) via selective pressure incorporation in methionine auxotrophic *Escherichia coli* (*E. coli*) B834 (DE3) cells as described by us previously (Supplementary Fig. 2a and c)[45]. Finally, H1.2 KxPlk proteins were linked to Ub G76Aha by CuAAC (Fig. 1b) to form a stable triazole linkage[45,52]. Ubiquitylated histones (H1.2 KxUb) were characterized by LC-MS and LC-MS/MS analysis (Fig. 1c and Supplementary Fig. 2d), and the correct fold of the secondary structure after click reaction and purification was again confirmed by CD spectroscopy (Supplementary Fig. 2b). Furthermore, the H1.2 KxUb conjugates still served as substrates in an in vitro ubiquitylation assay confirming their structural integrity (Supplementary Fig. 2e) and indeed resisted proteolytic cleavage during incubation in human cell lysate (Fig. 1d).

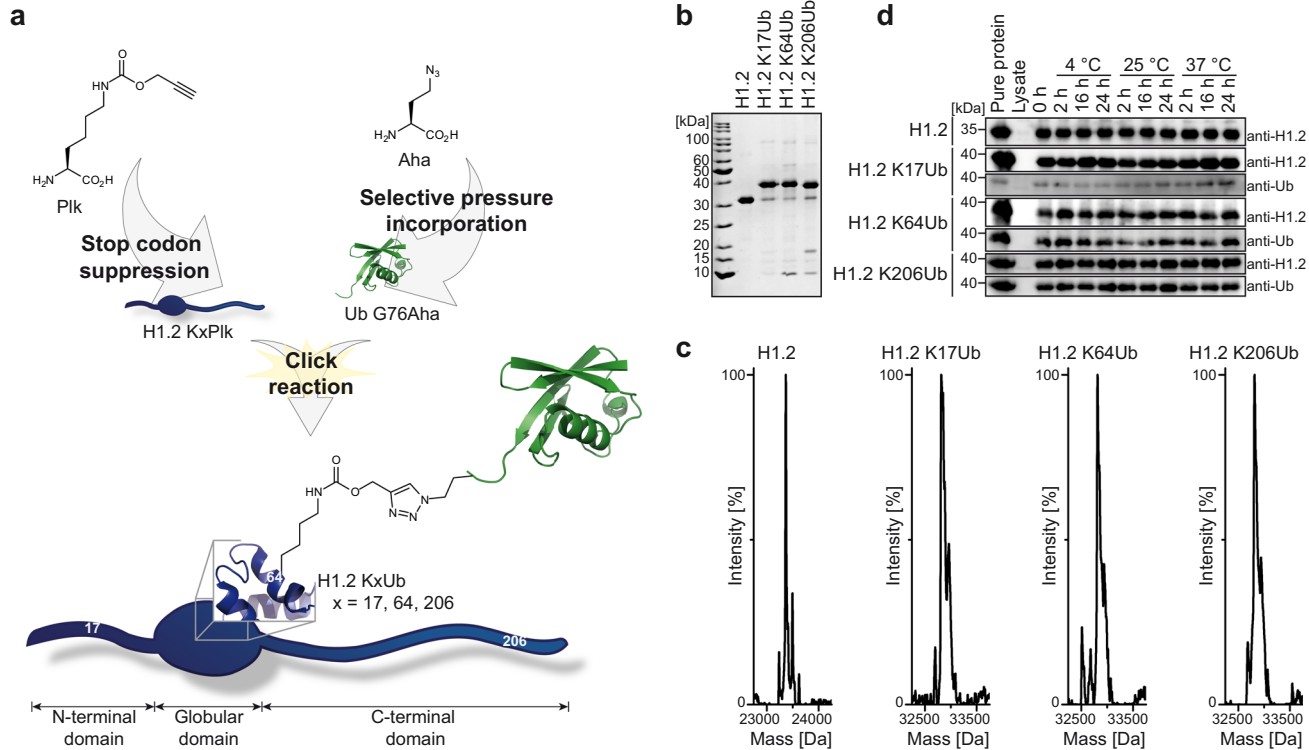

**Fig. 1 Generation of site-specifically ubiquitylated H1.2. a** Scheme representing the generation of ubiquitylated histones. Aha was incorporated into Ub via selective pressure incorporation and Plk into H1.2 by stop codon suppression. Finally, the proteins were linked by CuAAC. H1.2 was ubiquitylated at positions K17, K64, and K206. **b** Non-ubiquitylated H1.2 and H1.2 KxUbs resolved by SDS-PAGE followed by Coomassie blue staining. **c** LC-MS analysis of full-length proteins. H1.2 calc. 23370 Da, measured 23370 Da; H1.2 K17Ub calc. 32813 Da, measured 32818 Da; H1.2 K64Ub calc. 32813 Da, measured 32816 Da; H1.2 K206Ub calc. 32813 Da, measured 32815 Da. **d** Stability assay of H1.2 and H1.2 KxUbs. Proteins were incubated in HEK 293T cell lysate and analyzed by western blot showing constant levels of the respective proteins, indicating the stability of the conjugates over time at various temperatures.

**Identification of the ubiquitylation-dependent modular H1.2 KxUb interactome**. With sufficient amounts of non-hydrolyzable site-specifically ubiquitylated H1 in hand, we set out to dissect the ubiquitylation-dependent modular interactome. Building on previous efforts[53,54], we adapted an affinity purification-mass spectrometry (AP-MS) based approach to identify Ub-dependent interaction partners of H1 using its N-terminal Strep-tag II for enrichment (Fig. 2a, Supplementary Fig. 3a and Supplementary Data 1). Using HEK 293T cell lysates and the unmodified H1.2 and H1.2 KxUbs as bait proteins, as well as free (i.e., non-conjugated) Ub and empty beads as control, we identified 270 proteins that were consistently and significantly enriched over three biological replicate AP-MS experiments (ANOVA statistics, FDR = 0.001, S0 = 2).

The binding behavior of all significantly enriched proteins was sorted in an unbiased way by hierarchical clustering and the results were visualized in a heatmap, resulting in a total of six protein clusters that show similar binding behavior. Plotting the respective profile plots for each protein from each cluster highlights the specificity of the respective binding profiles for each cluster even more (Fig. 2b and Supplementary Fig. 3b). Interestingly, we find that the largest cluster (cluster 3) harbors proteins that bind to both non-modified and ubiquitylated H1.2, indicating a functional role for ubiquitylation besides the regulation of protein-protein interactions. We also identify proteins that seem to interact with non-modified H1.2 and ubiquitylated H1.2 except when ubiquitylated at position K64 (cluster 2). In cluster 6, we find proteins that exclusively interact with free Ub. The other three clusters (clusters 1, 4, and 5) contain proteins that exhibit Ub-specific interactions of H1.2. Here we find proteins that interact with all H1.2 KxUbs but

excitingly also proteins that interact specifically with only a subset of the H1.2 KxUb variants.

While the majority of the H1.2 interactors identified in this study bind to both non-modified and ubiquitylated H1.2, around twenty percent of the enriched interactors were specifically enriched for ubiquitylated H1.2 only (Fig. 2c and Supplementary Fig. 3b). Gene Ontology (GO) analysis indicates that the ubiquitylation-specific H1.2 interactors mainly represent proteins involved in nucleic acid binding, scaffold/adaptor proteins, and protein modification enzymes (Fig. 2c and Supplementary Fig. 3c). As for the latter, we find Ub ligases including *CUL4A* and *HUWE1*, as well as several deubiquitylating enzymes (DUBs) such as *USP15*, *USP13*, *UCHL5*, or *BRCC3* but also members of other enzyme classes, for example, the deacetylase *SIRT1*, the ATPase *WRNIP1*, and the phosphatase modulators *PPP6R1/R3*.

To verify and independently validate the results obtained by our LC-MS/MS analysis, we selected several proteins from each of the clusters and observed their binding behavior also by immunoblotting (Fig. 2d). Comparison of our western blot analysis and MS data revealed highly similar enrichment patterns, thereby validating not only the generally ubiquitylation-dependent interactions but also the site-specific interaction patterns detected by proteomic profiling. Finally, selected interactors co-immunoprecipitated with endogenous H1.2 further confirming our AP-MS results (Supplementary Fig. 3d). In summary, using proteomic profiling we show that site-specific ubiquitylation is a general modulator of the H1 interactome.

**Functional consequences of site-specific ubiquitylation of H1**. After having established site-specific ubiquitylation as a driver in shaping the modular interactome, we wanted to better

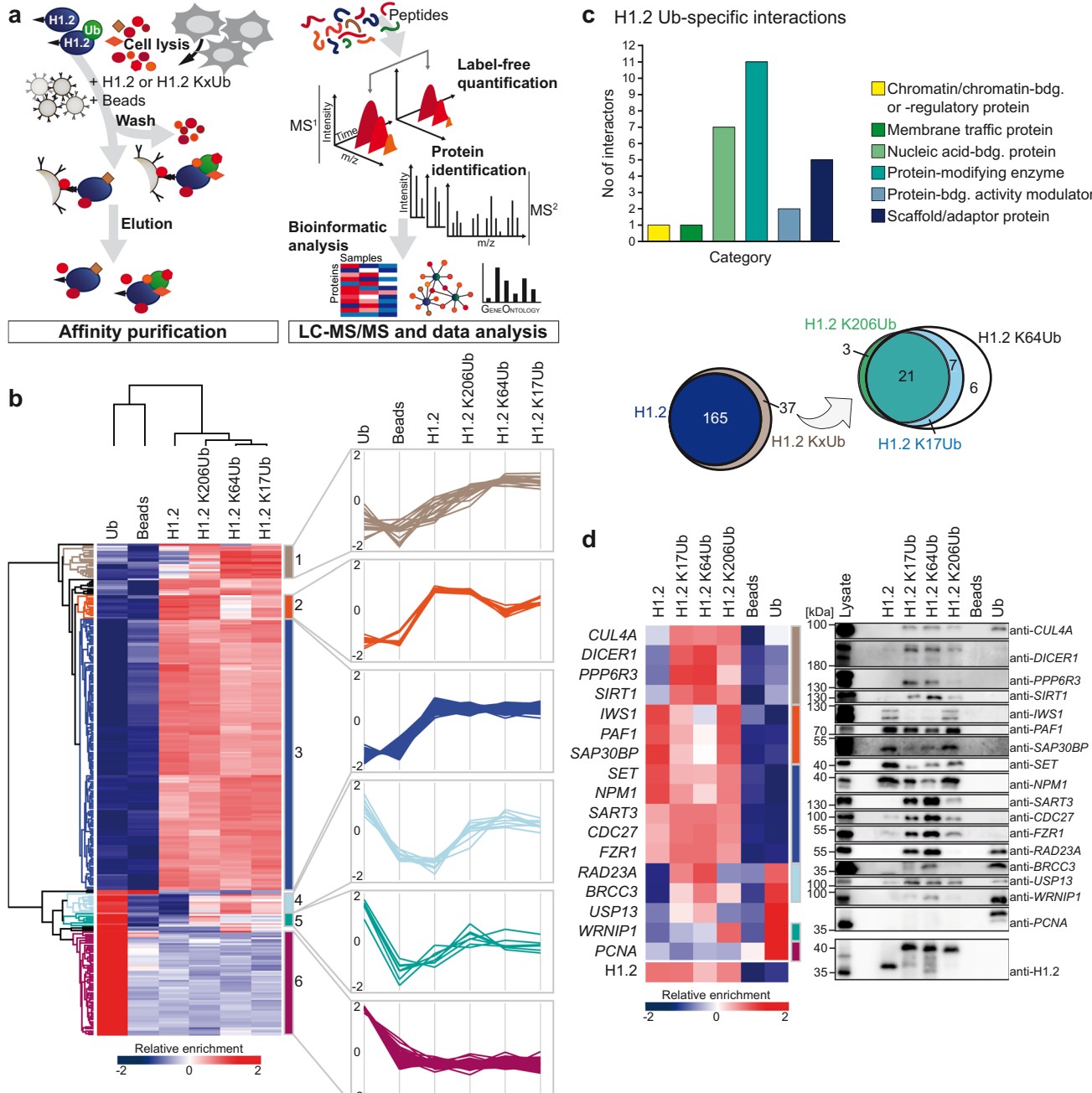

**Fig. 2 Identification of the ubiquitylation-dependent modular H1.2 KxUb interactome. a** Schematic overview of the AP-MS-based workflow to identify interactors of H1.2 KxUb variants. **b** Hierarchical clustering of statistically significant interactors following ANOVA analysis (FDR = 0.001, S0 = 2, n = 3). Interacting proteins are shown in rows, columns represent bait proteins used in AP-MS experiments. AP-MS experiments were carried out in biological triplicates in HEK 293T whole cell lysates. Empty beads (no bait protein) and beads decorated with free Ub were added as controls. Clusters of proteins with similar interaction behavior are depicted in different colors (left). Profile plots of clusters with specific binding patterns are shown on the right. **c** Venn diagrams (bottom) of all proteins that specifically interact with unmodified H1.2 or H1.2 KxUbs, respectively (left) and all H1.2 KxUb-specific interacting proteins with site-specific resolution (right), not including interactors of free Ub (for those see cluster 6). Further GO-term analysis of all ubiquitylation-specific H1.2 KxUb-interacting proteins based on PANTHER classification (top). **d** Heatmap showing the relative enrichment of selected proteins as identified by AP-MS (left) and by immunoblotting (right). Lysate and elution fractions of the affinity purification assay were subjected to western blot analysis with antibodies specific for the indicated proteins.

understand the functional implications of this PTM for H1. We, therefore, chose examples that were specifically enriched as Ub-specific interactors of H1, focusing on protein modification enzymes in general and protein ubiquitylation and deubiquitylation in particular as the most probable effectors/antagonists for the establishment of the dynamic ubiquitome.

Altogether, we identified six DUBs that were enriched for H1.2 KxUbs (Supplementary Fig. 4a). In the course of this study, we decided to focus on three of these DUBs—*USP15*, *USP13,* and *UCHL5*—due to their differential enrichment pattern and as they were amenable to further biochemical in vitro characterization. The quantified enrichment patterns from our AP-MS showed

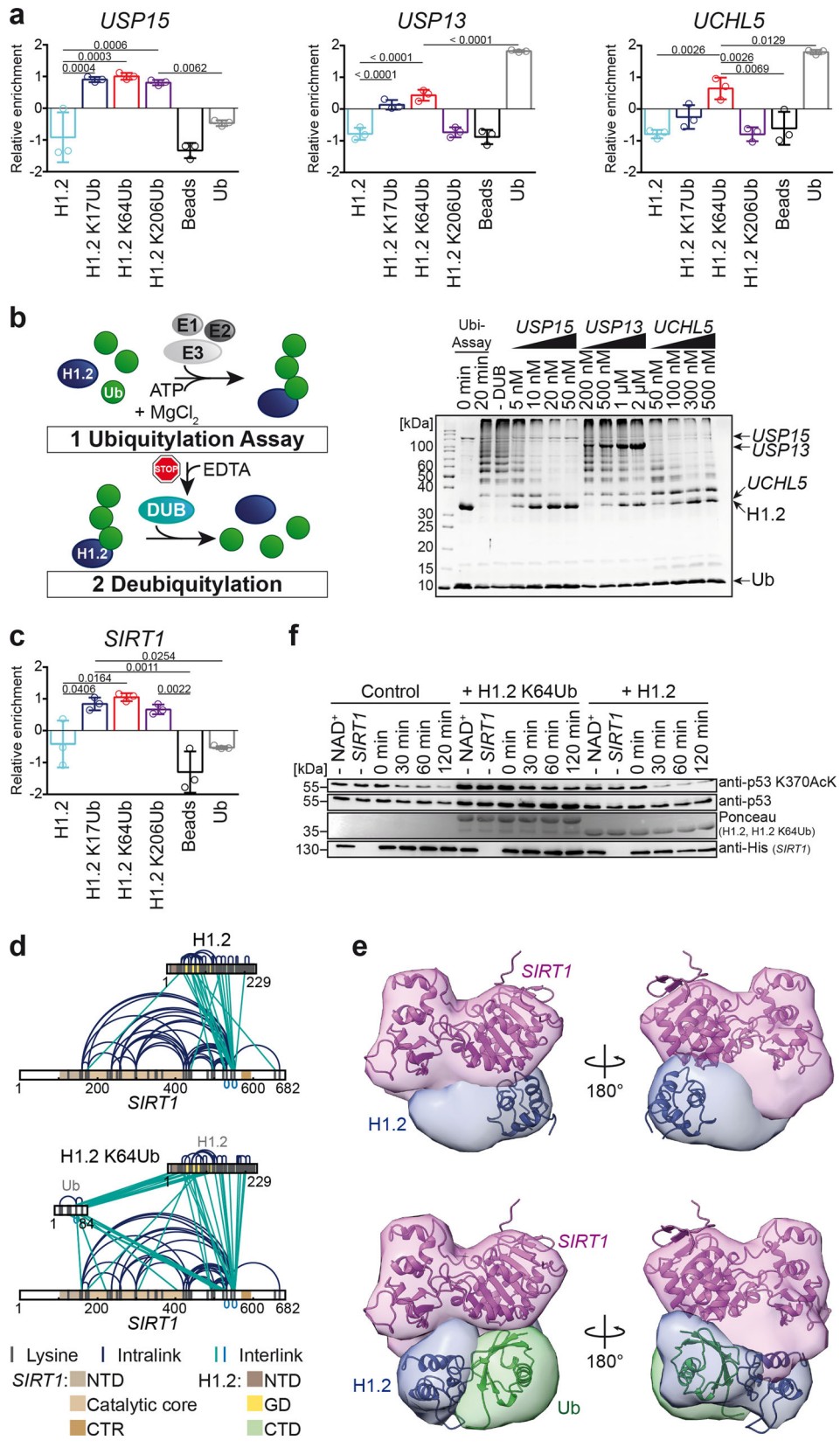

that, while all three investigated DUBs are significantly enriched for ubiquitylated H1.2, they differ in their site-specificity (Fig. 3a, Supplementary Fig. 4a and 5). While USP15 is uniformly interacting with all H1.2 KxUbs, USP13 and in particularly UCHL5 interact primarily with H1.2 K64Ub. To have a closer look at the deubiquitylation-activity of these DUBs towards the linker histone, we generated ubiquitylated H1.2 by in vitro ubiquitylation (Supplementary Fig. 4b) using the catalytic core of HUWE1 as an E3 ligase, which was specifically enriched for ubiquitylated H1.2 in our dataset and has also been previously

**Fig. 3 Functional characterization of H1.2 KxUb-specific interactions with DUBs and *SIRT1*. a** Relative enrichment patterns of DUBs *USP15*, *USP13*, *UCHL5*. Plotted is the mean of the Z-scores from the quantitative proteomics data. Shown are mean values $+/-$ standard deviation, $n = 3$, one-way ANOVA (repeated measures) with Tukey's multiple comparisons test with $\alpha = 0.05$, 95% confidence interval. Selected significance lines and exact p values with $p \leq 0.05$ are shown, for complete results of significance analysis see Supplementary Fig. 5. **b** Scheme representing the workflow of the deubiquitylation assay (left). H1.2-deubiquitylation assay for *USP15*, *USP13*, and *UCHL5* (right). −DUB indicates that no DUB was added after the ubiquitylation assay. **c** Relative enrichment pattern of *SIRT1*. Plotted is the mean of the Z-scores from the quantitative proteomics data. Shown are mean values $+/-$ standard deviation, $n = 3$, one-way ANOVA (repeated measures) with Tukey's multiple comparisons test with $\alpha = 0.05$, 95% confidence interval. Selected significance lines and exact p values with $p \leq 0.05$ are shown, for complete results of significance analysis see Supplementary Fig. 5. **d** Overall crosslinking pattern of H1.2: *SIRT1* (top) or H1.2 K64Ub: *SIRT1* (bottom) complexes. NTD = N-terminal domain, CTR = C-terminal regulatory domain of *SIRT1*; structural domains of H1.2 as described before. **e** Models of the H1.2: *SIRT1* (top) and H1.2 K64Ub: *SIRT1* (bottom) complex. Bayesian crosslinking guided integrative structural modeling using the crystal structures of *SIRT1* (PDB: 4ig9), Ub (PDB: 1ubq), and the chicken H1 GD (PDB: 1ghc) together with our crosslinking data as input. Shown are crystal structures overlaid with the average cluster density. The second cluster for H1.2 K64Ub: *SIRT1* is depicted in Supplementary Fig. 7. **f** In vitro deacetylation assay for *SIRT1* and model substrate protein p53 K370AcK in the presence of H1.2, H1.2 K64Ub or in the absence of any histone (Control). −NAD$^+$ and −*SIRT1* indicate that the respective component was not present in the reaction mixture. Protein and acetylation intensities were visualized by western blot and Ponceau S staining.

shown to ubiquitylate linker histone H1[55]. Thereby, H1.2 was ubiquitylated at several positions, including also the studied positions K17, K64, and K206 (Supplementary Fig. 4c). These ubiquitylated histones were then incubated with *USP15*, *USP13*, or *UCHL5* resulting in both concentration-dependent and time-dependent deubiquitylation of H1.2 for all DUBs tested. While this confirms that these DUBs can act on H1.2 in vitro and concurrently indicates that they also act as DUBs for H1 in vivo (Fig. 3b and Supplementary Fig. 4d), our data and the differences in relative deubiquitylation rates also indicate a ubiquitylation-site mediated specificity for DUBs.

We then had a look at the human NAD$^+$-dependent deacetylase *SIRT1*, which is involved in epigenetic regulation and a known interactor of linker histones[56]. Interestingly, while we observed at best a marginal interaction between *SIRT1* and non-modified H1.2 in our AP-MS analysis, we observed significantly higher enrichment of the deacetylase with ubiquity-lated H1.2. Both our quantified enrichment patterns from AP-MS and our immunoblotting data show that *SIRT1* is significantly enriched for ubiquitylated linker histone with a site-specific preference for H1.2 K64Ub (Figs. 2d, 3c and Supplementary Fig. 5).

To further investigate this site-specific interaction of H1.2 K64Ub with *SIRT1*, we performed chemical cross-linking coupled to mass spectrometry (XL-MS) (Fig. 3d and Supplementary Fig. 6a, Supplementary Data 2). XL-MS uses covalent bonds formed by crosslinking reagents to identify crosslinking sites by MS that reflect the spatial proximity of regions within a given protein (intralinks) or between different proteins or subunits in a protein complex (interlinks) that can be used to precisely map regions and domains of interaction[57]. Our crosslinking pattern shows a large number of interlinks between H1.2 and *SIRT1* in the H1.2: *SIRT1* complex and between *SIRT1* and both H1.2 and Ub in the H1.2 K64Ub: *SIRT1* complex, demonstrating that ubiquitylated linker histone H1.2 and *SIRT1* bind each other and form a protein complex in vitro (Fig. 3d). If we add, however, purified Ub to H1.2 and *SIRT1* or *SIRT1* alone as a control, no interaction between Ub and either H1.2 or *SIRT1* can be detected, strongly indicating that we have indeed mapped the location of Ub in the H1.2 K64Ub: *SIRT1* complex (Supplementary Fig. 6a). Furthermore, the distribution of detected crosslinks indicates that *SIRT1* interacts mainly via a domain close to its C-terminal regulatory domain with H1.2. To generate a model of the intact H1.2 K64Ub: *SIRT1* complex we additionally performed Bayesian crosslinking guided integrative structural modeling using the crystal structures of *SIRT1*, Ub, and the H1 GD together with our crosslinking data as input[58]. This resulted in unbiased, highly reproducible, and robust models for H1.2: *SIRT1*

and H1.2 K64Ub: *SIRT1* (Fig. 3e, Supplementary Fig. 7a and b; see "Methods" section for more details). While our models show more than one possible localization for Ub within the H1.2 K64Ub: *SIRT1* complex, suggesting multiple conformational states, they also unambiguously demonstrate that ubiquitylation of histone H1 significantly impacts the positions of H1.2 and *SIRT1* relative to each other, suggesting conformational changes within the H1.2: *SIRT1* complex upon ubiquitylation.

Having established that H1.2 K64Ub forms a distinct complex with *SIRT1*, we speculated that the enzymatic activity of *SIRT1* is modulated by H1.2 in a ubiquitylation-dependent manner. To test this hypothesis, we used the tumor suppressor p53, a known substrate of *SIRT1*[59,60], which we had acetylated at position K370 using unnatural amino acid technology[37], and incubated it together with *SIRT1* and either H1.2 or H1.2 K64Ub. We found that the deacetylation activity of *SIRT1* is slightly decreased after incubation with H1.2 K64Ub relative to H1.2, suggesting that the ubiquitylation status of H1.2 at position K64 affects the enzymatic activity of *SIRT1* (Fig. 3f and Supplementary Fig. 6b).

Taken together, we show that site-specific ubiquitylated H1.2, particularly at position K64, selectively interacts with and functions as a substrate for distinct DUBs and modulates the enzymatic activity of *SIRT1* in vitro.

**Site-specific ubiquitylation of H1 impacts chromatosome assembly.** To study the effect of site-specific H1.2 ubiquitylation at the level of the intact chromatosome, we generated both 12-mer nucleosome and chromatosome arrays and characterized their behavior in multiple assays. To generate intact chromatosomes, a DNA sequence containing twelve repeating units of the nucleosome positioning sequence was mixed with core histones to assemble nucleosomes before unmodified H1.2 or H1.2 KxUbs were incorporated (Fig. 4a and Supplementary Fig. 8a).

Unmodified H1.2, as well as the different H1.2 KxUb conjugates, were successfully incorporated into the array (Fig. 4b, Supplementary Fig. 8a and b). We first looked at the precipitation of the arrays with MgCl$_2$ as an indicator of the general solvent accessibility of the linker DNA and thus a criterion for their overall compaction state (Supplementary Fig. 8c)[61]. While we found that chromatosome arrays generally precipitated at lower MgCl$_2$ concentrations than nucleosome arrays, we could not detect any significant differences between chromatosomes containing unmodified H1.2 or the different H1.2 KxUbs.

However, ubiquitylation of H1.2 resulted in a slightly retarded migration behavior using an electrophoretic mobility shift assay (EMSA) (Fig. 4b), which may result from the additional protein mass of Ub, from a more open or relaxed chromatosome structure or both. We, therefore, probed the accessibility of DNA

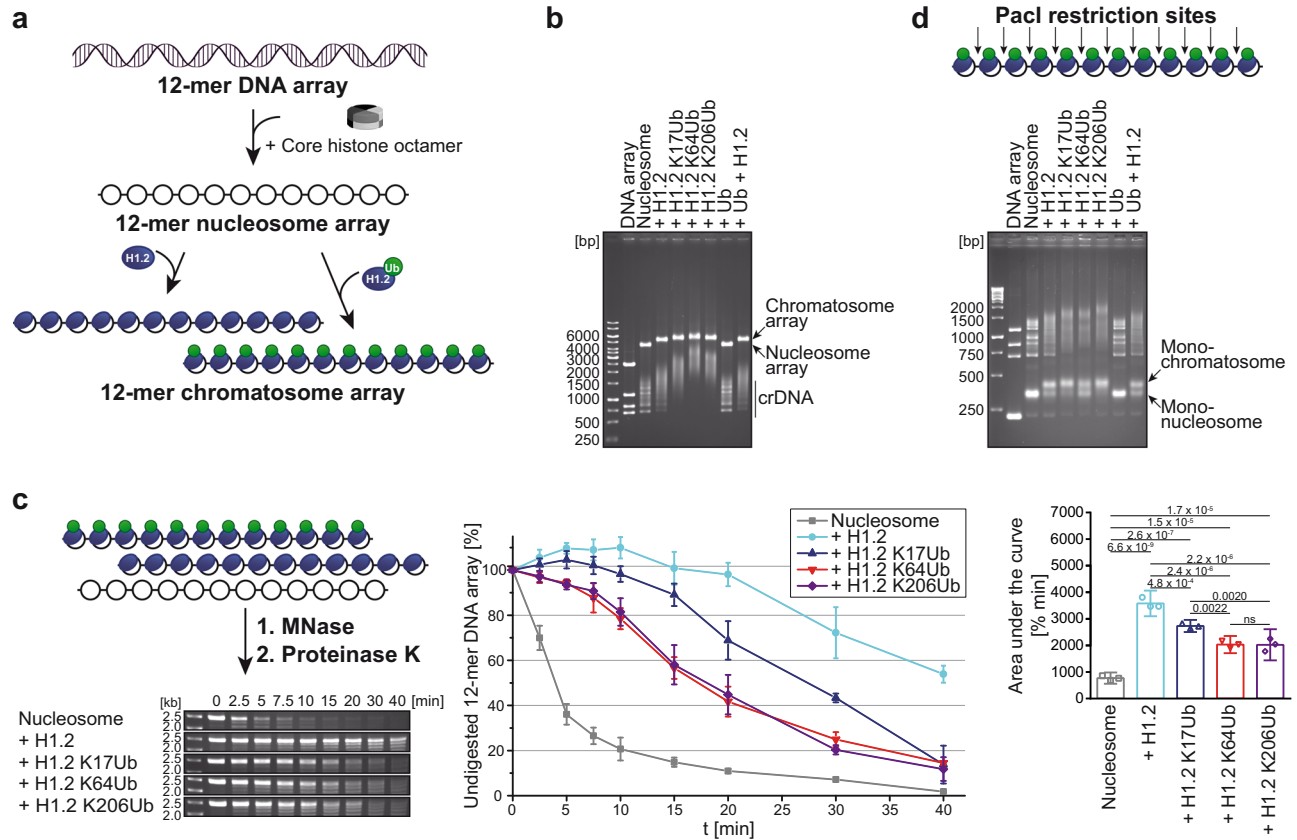

**Fig. 4 Impact of H1.2 KxUb variants on chromatosome assembly. a** Schematic overview of 12-mer chromatosome array formation containing H1.2 or H1.2 KxUbs. **b** EMSA of different arrays analyzed; crDNA indicates competitor DNA. **c** Overview of MNase digestion assay which probes DNA accessibility of nucleosome and chromatosome arrays (left). The arrays were treated with MNase for indicated times, digested by proteinase K, and analyzed by agarose gel electrophoresis. Undigested arrays were quantified and plotted over time (middle). Shown are mean values $+/-$ standard deviation, $n = 3$. For statistical analysis, the area under the curve for the digestion of the different arrays was determined and analyzed (right); shown are mean values $+/-$ standard deviation, $n = 3$, one-way ANOVA with Tukey's multiple comparisons test with $\alpha = 0.05$, 95% confidence interval, exact $p$ values with $p \leq 0.05$ are shown, ns (not significant) $p > 0.05$. **d** PacI digestion assay. PacI cleaves the array site-specifically within the linker DNA between the repetitive nucleosome positioning sequences resulting in the formation of mono-nucleosomes and mono-chromatosomes.

by digestion with the micrococcal endo-exonuclease MNase (Fig. 4c)[62]. Here we found that chromatosomes are generally more stable against MNase degradation than nucleosomes and that chromatosome arrays containing unmodified H1.2 are significantly more stable than arrays with ubiquitylated H1.2. Importantly, we observed significant differences between the different H1.2 KxUb variants, with chromatosomes containing H1.2 K17Ub being significantly more stable than H1.2 K64Ub or H1.2 K206Ub containing arrays. This data, therefore, demonstrates that DNA in chromatosomes containing ubiquitylated H1 variants is more accessible and therefore strongly suggests a more open conformation for chromatosomes, particularly for arrays containing H1.2 K64Ub or H1.2 K206Ub. Furthermore, digestion of chromatosome arrays with the restriction enzyme PacI, which cleaves between nucleosomes, resulted in a high proportion of mono-nucleosomes for H1.2 K64Ub (Fig. 4d). In contrast, mono-chromatosomes were more abundant when unmodified H1.2 and the ubiquitylated variants H1.2 K17Ub and H1.2 K206Ub were used. This indicates that H1.2 K64Ub has a lower affinity to nucleosomes than unmodified and N-terminal or C-terminal ubiquitylated H1.2.

In conclusion, while the MgCl$_2$ precipitation data indicates that ubiquitylation induces no major changes to the overall compaction state, our more detailed characterization of the assembly state of H1.2 KxUb-bound chromatosomes by both EMSA and MNase

digestion strongly suggests that ubiquitylation of H1 induces specific conformational changes resulting in chromatosomes with a more open conformation. This is, in particular, the case when Ub is site-specifically attached at positions K64 or K206 of H1.2 and is potentially caused by lower binding affinities for nucleosomes as shown by the PacI digest.

## H1 condensate formation is modulated by site-specific ubiquitylation.
As H1 has recently been found to form condensates and to modulate chromatin LLPS[9,10], we wanted to study a potential effect of ubiquitylation on H1.2-dependent phase separation. To do so, we incubated fluorophore-labeled versions of the various H1.2 KxUb conjugates with fluorescently labeled DNA[63] or 12-mer nucleosome arrays, respectively, and monitored condensate formation by fluorescence microscopy (Fig. 5a and Supplementary Fig. 9a).

Focusing first on H1.2/H1.2 KxUb-DNA condensates, both H1.2 and all H1.2 KxUb variants formed liquid-like droplets (Fig. 5b), whereas with H1.2 or DNA on their own no condensate formation was observed (Supplementary Fig. 9b). An increase in the ionic strength by the addition of NaCl, as well as lowering the amount of DNA and H1.2 resulted in a decreased number of condensates (Supplementary Fig. 9c). All condensates also recovered their fluorescence signal in fluorescence recovery

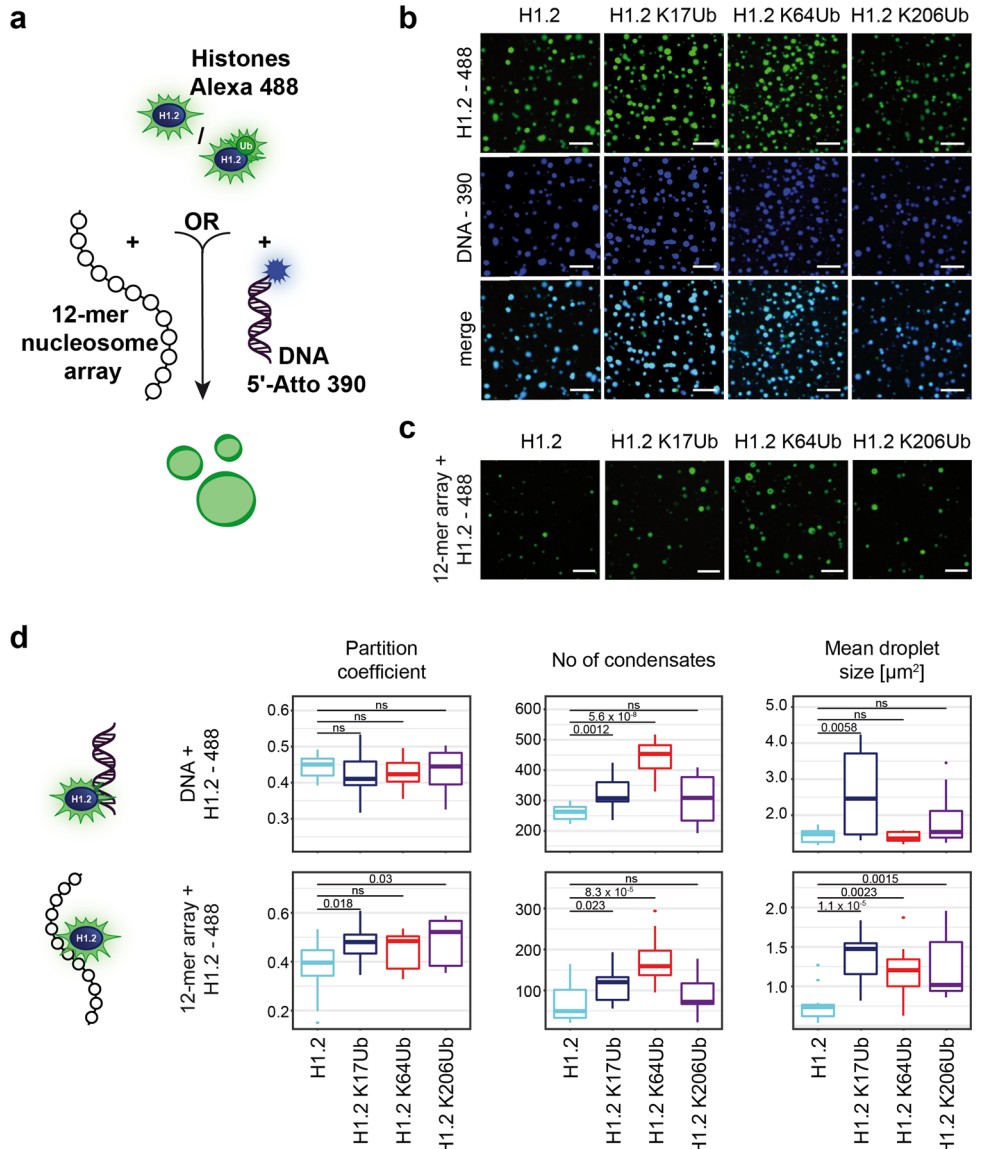

**Fig. 5 H1.2 KxUb-dependent condensate formation. a** Overview of condensate formation assay using fluorescently labeled H1.2 and H1.2 KxUbs where either the nucleosome positioning DNA or intact 12-mer nucleosome arrays were added. The different linker histone variants were chemically labeled with Alexa Fluor 488-NHS, DNA was amplified with a 5′-Atto-390-modified primer while the DNA array was not labeled. **b** Representative microscopic images of various H1.2- and H1.2 KxUb-DNA condensates. Shown are images after excitation with a 488 nm laser (green) or a 405 nm laser (blue) and the merged images (light blue). Scale bar 10 μm. **c** Representative microscopic images of H1.2- and H1.2 KxUb-array condensates. Shown are images after excitation with the 488 nm laser (green). Scale bar 10 μm. **d** Characterization of H1.2- and H1.2 KxUb-DNA (top) and H1.2- and H1.2 KxUb-nucleosome array (bottom) condensates. Shown are the partition coefficient (left), the number of droplets (middle), and mean droplet size (right) 20 min after mixing. Data were extracted from the images in the histone (green) channel and are shown as boxplots where the midline represents the medians, the upper and lower bounds the interquartile ranges, and the whiskers extend to 1.5 times the interquartile range (two-sample t-test; n = 4, exact p values with p ≤ 0.05 are shown, ns (not significant) p > 0.05).

after photo-bleaching (FRAP) experiments, confirming liquid-like properties (exemplarily shown for H1.2 in Supplementary Fig. 9d). These results confirm that both H1.2 and H1.2 KxUb form condensates together with their cognate DNA and show that the ionic interaction of both molecules is required for this process.

To determine a potential impact of site-specific ubiquitylation, we closely monitored the morphology and dynamics of H1.2 KxUb condensates and analyzed the partition coefficient, number, and size of formed droplets (Fig. 5d, top) and their liquid-like dynamics as assessed by FRAP (Supplementary Fig. 9f and g). We found no significant effect of ubiquitylation on the partition coefficients of the resulting condensates (Fig. 5d, left)

and detected only a marginal influence on their size (Fig. 5d, right) and dynamics (Supplementary Fig. 9f and g). However, we observed that H1.2 and the H1.2 KxUb variants differ in the total number of formed condensates (Fig. 5d, middle). We additionally assessed phase separation behavior of labeled H1.2 and H1.2 KxUb variants with the 12-mer nucleosome arrays (Fig. 5a and c) and found that their behavior regarding partition coefficient, number, and size mirrors our observations with the H1.2/H1.2 KxUb-DNA condensates (Fig. 5d, bottom). While the effect on mean droplet size is even more pronounced for intact chromatosomes, histones ubiquitylated in their NTD (H1.2 K17Ub) or GD (H1.2 K64Ub) formed significantly more droplets than

non-ubiquitylated H1.2 in both settings (Fig. 5d). These results suggest that ubiquitylation of the NTD or GD of H1.2 leads to the formation of more but less concentrated condensates as partition coefficients were unchanged while the number and size of formed condensates increased.

Taken together, we, therefore, find that site-specific ubiquitylation affects and modulates H1.2-mediated condensate formation in the presence of DNA alone or of intact 12-mer nucleosome.

## Discussion

While the role and functional relevance of specific PTMs is increasingly well understood for core histones, this is not the case for linker histone H1, even though it plays a fundamentally important role in the formation of higher-order chromatin structures and epigenetic regulation. Due to the lack of appropriate technology, research on PTM-specific regulation of H1 has so far been difficult, as evidenced by the fact that various recent reviews on epigenetic control do not even mention linker histone H1[14–16]. As a consequence, our understanding of the contribution of linker histone H1 to the histone code clearly lags behind research on core histones[1,3–5].

In this study, we have established the use of CuAAC to generate site-specifically ubiquitylated variants of linker histone H1. By adapting and optimizing established protocols based on amber stop codon suppression and selective pressure incorporation we were able to modify linker histone H1.2 at defined positions and to generate sufficient quantities of pure, non-hydrolyzable H1.2 KxUb variants to study the consequences of site-specific ubiquitylation by proteomic profiling and biochemical characterization. Our study, therefore, serves as a general example of how chemical protein synthesis, proteomic profiling, and biochemical characterization can be deployed in a powerful approach to dissect the PTM-specific modular proteome and to study its functional consequences on a molecular level.

We applied the H1 Ub conjugates to proteome-wide protein-protein interaction studies to find that site-specific ubiquitylation results in overlapping, but distinct interactomes. While the majority of the interactors identified in this study bind to both non-modified and ubiquitylated H1.2, around twenty percent of the enriched proteins interacted in a Ub-specific manner (Fig. 2c). These H1.2 ubiquitylation-specific interactors comprise a large number of enzymes involved in protein modification. This suggests that ubiquitylation of H1 does indeed regulate cellular functions by direct protein-protein interactions and acts via specific writers, readers, and erasers to mediate transcriptional responses, much as described for other histone marks[11–16].

This is very much in line with our other data, where we moved on to demonstrate that site-specific ubiquitylation of H1 indeed carries functional consequences. We first had a closer look at some of the enzymes that we had identified as Ub-specific interactors. H1.2 ubiquitylated at position K64 preferentially interacts with a defined subset of DUBs, which also deubiquitylated H1 Ub in vitro. We next showed that H1.2 K64Ub also modulates the interaction with the deacetylase SIRT1 in vitro. Comparing our integrative model of the H1.2 K64Ub: SIRT1 complex with its non-ubiquitylated counterpart demonstrates that the site-specific ubiquitylation of histone H1.2 at position K64 results in a conformational change within the complex thereby pointing at the molecular underpinnings of H1.2 K64Ub-mediated regulation of SIRT1 deacetylase activity. The fact that we identify at least two main clusters of structural solutions for the H1.2 K64Ub: SIRT1 complex may also indicate multiple functional Ub-dependent conformations (Fig. 3e and Supplementary Fig. 7). Since SIRT1 itself targets various nuclear proteins including core histones, as well as linker histones which eventually leads to transcriptional

repression[56], this indicates H1-dependent crosstalk between the different epigenetic modifications. Our data also suggests that ubiquitylation of H1.2 at position K64 affects the deacetylation capacities of SIRT1, thereby potentially counteracting its transcriptional repressive function.

The idea that H1 ubiquitylation counteracts transcriptional repression is consistent with our other data that also suggests a role for H1 in transcriptional control. In our in vitro studies on H1 molecular condensates, we found that site-specific ubiquitylation affects and modulates condensate formation of H1. Our results showed that ubiquitylation of H1.2 at position K64 leads to the formation of more but less concentrated H1-dependent condensates. Remarkably, these site-specific ubiquitylation effects for condensates consisting of H1 and its nucleosome positioning DNA are fully mirrored by condensates consisting of intact chromatosomes. Reducing compactness by obtaining a more open conformation would explain the observed reduction in the concentration of H1-dependent condensates.

This observation ties in very well with the data from our other assays with intact chromatosomes, which clearly showed that ubiquitylation of the linker histone, particularly at position K64, results in a more open conformation, where the DNA is more accessible. This is exciting, as previous studies had shown that the binding of unmodified H1 induces the nucleosome to adopt a more compact and rigid conformation[64] and to promote more condensed chromatin structures[65]. Our data now suggest that ubiquitylation of H1.2, especially at position K64, counteracts these processes and relaxes chromatin via distinct but likely rather subtle conformational changes. This is in line with data on other histones, where it had been shown that the attachment of Ub to histone H2B inhibits nucleosome array compaction[66]. Generally, it is quite astonishing, given the size of Ub (8.5 kDa) in relation to H1.2 (21.4 kDa), that our data shows that ubiquitylation does neither prevent the formation of higher-order chromatin structures nor lead to a disordered arrangement of the chromatosome array in the first place.

Taken together our data suggests that the site-specific ubiquitylation of linker histone H1 promotes a transcriptionally active state: both by direct interaction and regulation of deacetylases such as SIRT1 and by promoting a more open conformational chromatosome state, generally associated with a transcriptionally active state. Finally, ubiquitylation might also counteract phase-separation mediated heterochromatin formation by modulation of H1-dependent condensates[8]. While the necessary technology to address our model of an H1-mediated concept of ubiquitylation and phase separation-dependent transcriptional control in vivo has yet to be developed, it constitutes an important step forward in the quest for a comprehensive understanding of the histone code and a significant step forward towards the design of functional, intact synthetic chromatin.

## Methods

**Sample preparation for mass spectrometry (MS).** For in-solution digestion, freeze-dried samples were denatured in 8 M urea, reduced by the addition of 5 mM TCEP at 37 °C for 30 min, and subsequently alkylated using 10 mM 2-iodoacetamide at RT for 30 min in the dark. After the addition of 50 mM $NH_4HCO_3$ to a final concentration of 1 M urea, samples were digested using trypsin (1:50, w/w) (Promega) overnight at 37 °C. Finally, samples were acidified with TFA and lyophilized or vacuum concentrated. For in-gel digestion, proteins were separated by SDS-PAGE and visualized by Coomassie staining. The spots or lanes of interest were excised from the gel, destained in 50 mM $NH_4HCO_3$, 50% (v/v) ACN, and washed with 50 mM $NH_4HCO_3$. Next, disulfide bonds were reduced by the addition of 10 mM DTT in 50 mM $NH_4HCO_3$ for 1 h at 56 °C, followed by alkylation in 50 mM 2-iodoacetamide or 2-chloroacetamide in 50 mM $NH_4HCO_3$ for 1 h at RT in the dark. After washing in 50 mM $NH_4HCO_3$, 50% (v/v) ACN and dehydration in ACN, proteins were digested overnight at 37 °C using trypsin. For digestion with pepsin (Sigma-Aldrich), the protease was added in 0.1% TFA. Peptides were extracted in 30% ACN, 5% formic acid and 60% ACN, 5% formic acid. The combined extracts were freeze-dried.

Before LC-MS/MS analysis, samples were desalted using standard ZipTips (C18) or Micro-ZipTips (μ-C18) (Merck). Recombinant protein crosslinking samples were desalted with C18 cartridges (Waters) followed by further processing.

**Expression and purification of H1.2 and H1.2 KxPlk**. Plk was synthesized essentially as described[67]. In short, Boc-Lys-OH was dissolved in 1 M NaOH and THF, followed by cooling to 0 °C. Propargyl chloroformate was added dropwise and the reaction was allowed to stir at RT for 16 h. The solution was cooled again, washed with ice-cold $Et_2O$, acidified with 1 M HCl, and extracted with EtOAc. The combined organic layers were dried over $MgSO_4$ and the solvents evaporated to give Boc-Plk as a white foam in 71% yield. The propargyl carbamate was dissolved in DCM and TFA was added dropwise and the reaction was allowed to stir at RT for 1 h. The solvents were evaporated, and the product precipitated through the addition of $Et_2O$, filtered, and dried, affording Plk as a white solid in 98% yield.

The gene encoding human H1.2 and a C-terminal His$_6$-tag and an N-terminal Strep-tag II (Supplementary Table 2) was inserted into the multiple cloning site of pET11a containing also the expression cassette for the tRNA$^{Pyl}$ gene in its backbone. For incorporation of Plk by stop codon suppression, the K17, K64, or K206 codon of H1.2 was replaced by an amber stop codon using site-directed mutagenesis. E. coli BL21 (DE3) cells were co-transformed with these plasmids and the pRSFDuet-1 vector, containing the PylRS gene from Methanosarcina barkeri. Cells were cultured in LB medium supplemented with appropriate antibiotics at 37 °C. At $OD_{600} = 0.3$–0.4, 2 mM Plk was added, and once $OD_{600} = 0.6$–0.8 was reached, 1 mM IPTG was added. After 14–16 h cells were harvested by centrifugation. For expression of H1.2, only the respective pET11a plasmid was transformed into E. coli BL21 (DE3) and no Plk was added during expression.

To isolate H1.2 or H1.2 KxPlk from inclusion bodies, cells were lysed by sonication in 50 mM Tris base, 100 mM NaCl, 1 mM EDTA, 1% (v/v) Triton X-100, 2 mM PMSF (pH 8.5) and the pellet was washed three times. Then, pellets were incubated in 50 mM Tris base, 1 M NaCl, 6 M urea, 10 mM β-mercaptoethanol (pH 7.0) and after centrifugation, cOmplete His-Tag Purification Resin (Roche) was added to the supernatant and incubated overnight at 4 °C. Beads were washed with increasing concentrations of imidazole and finally eluted with up to 500 mM imidazole. Elution fractions were pooled, dialyzed in water, and concentrated by ultrafiltration. Protein concentration was determined by BCA assay (Thermo Fisher Scientific) and SDS-PAGE analysis followed by Coomassie blue staining. The H1.2 KxPlk variants were obtained in 0.6 mg (H1.2 K17Plk), 0.3 mg (H1.2 K64Plk), and 1.3 mg (H1.2 K206Plk) yield per liter of expression culture.

**Expression and purification of Ub (used for AP-MS controls)**. The gene encoding human Ub with an N-terminal Strep-tag II was inserted into pGEX-2TK where the GST-coding sequence was removed. The plasmid was transformed into E. coli BL21 (DE3). For expression of Ub, cells were cultured in LB medium at 37 °C. At $OD_{600} = 0.6$–0.8, 1 mM IPTG was added, and after 8 h cells were harvested by centrifugation.

Cells were resuspended in 100 mM Tris-HCl (pH 8.0), 150 mM NaCl, 1 mM EDTA, 1 mM PMSF, 0.1 mg/ml lysozyme, 0.01% (v/v) Triton X-100, incubated on ice and lysed by sonication. After centrifugation, the supernatant was pre-cleared by heat denaturation. Therefore, the mixture was incubated at 60 °C for 20 min, followed by centrifugation. The resulting supernatant was manually loaded onto a Strep-Tactin Superflow Plus Cartridge (Qiagen), washed with 100 mM Tris-HCl (pH 8.0), 150 mM NaCl, 1 mM EDTA, followed by elution with 2.5 mM desthiobiotin. Elution fractions were concentrated by ultrafiltration and the concentration determined by BCA assay/SDS-PAGE and Coomassie blue staining.

**Expression and purification of Ub G76Aha**. Ub G76Aha was generated by selective pressure incorporation essentially as described[45]. In brief, methionine auxotrophic E. coli B834 (DE3) cells were transformed with pGEX-2TK/Ub G76M, coding for an N-terminal GST fusion to Ub and where the C-terminal G codon was replaced by an ATG/AUG codon. For recombinant expression, cells were cultured at 37 °C in NMM (new minimal medium) containing 0.04 mM methionine until the stationary growth phase was reached. Next, cells were harvested by centrifugation and resuspended in fresh NMM without methionine but supplemented with 0.5 mM Aha (Iris Biotech). After 30 min at 37 °C, expression of the Ub G76Aha construct was induced by the addition of 1 mM IPTG. After another 14–16 h at 25 °C, cells were harvested by centrifugation.

For purification, bacteria were resuspended in 1× PBS buffer containing 1% (v/v) Triton X-100 and lysed by sonication. After centrifugation, the supernatant was incubated with Glutathione Sepharose beads (GE Healthcare). Then, beads were washed with 1× PBS and finally, cleavage of the bound GST fusion protein was allowed by the addition of thrombin and incubation overnight at RT. Ub G76Aha was eluted in 1× PBS and further clarified by heat denaturation of co-purifying proteins (20 min at 60 °C), followed by centrifugation. The supernatant containing Ub G76Aha was concentrated by ultrafiltration and protein concentration was determined by BCA assay.

**Generation of H1.2 KxUb by Cu(I)-catalyzed azide-alkyne cycloaddition (CuAAC) (click reaction)**. 10 μM H1.2 KxPlk was mixed with 15–60 μM Ub

G76Aha in 1× PBS or Tris-HCl (pH 7.4), 0.2–1.2 mM SDS and 10 mM THPTA, 5 mM $Cu(MeCN)_4BF_4$. To prevent protein oxidation, reaction vessels were flushed several times with argon. After incubation on ice for 1 h, the reaction was stopped by the addition of 50 mM EDTA and centrifugation for 10 min at 21,000×$g$, 4 °C. Product yield and solubility were analyzed by SDS-PAGE and Coomassie blue staining. CuAAC conversion rates were optimized for each batch of protein to obtain >90–95% of conversion. Supernatants were pooled. H1.2 KxUb conjugates were purified and washed by ultrafiltration and finally stored in water. Pellets obtained after click reactions which also contained H1.2 KxUb were resuspended in 50 mM Tris base, 1 M NaCl, 6 M urea, 10 mM β-mercaptoethanol (pH 7.0). To enable protein refolding, the conjugates were dialyzed in a stepwise gradient to pure water at 4 °C for at least three days in total. Finally, proteins were concentrated by ultrafiltration. In case any Ub G76Aha was left, the conjugates were further purified by size-exclusion chromatography (HiLoad 16/600 Superdex 75 pg) (GE Healthcare) in 20 mM Tris-HCl (pH 7.4), 300 mM NaCl followed by concentration via ultrafiltration. Protein concentration was determined by BCA assay and SDS-PAGE with Coomassie blue staining. Isolated yields after conjugation varied with around 25% (H1.2 K17Ub), 5% (H1.2 K64Ub), and 30% (H1.2 K206Ub) of used H1.2 KxPlk, respectively.

The formation of the triazole linkage was verified by LC-MS/MS analysis. Therefore, proteins were loaded on an SDS-PAGE gel, visualized with Coomassie blue staining, the respective band cut from the gel, and prepared for analysis by in-gel digestion with trypsin or pepsin.

Samples were measured on an Orbitrap Fusion Tribrid mass spectrometer (Thermo Fisher Scientific) operated with Tune 2.1.1456.23 (Thermo Fisher Scientific) coupled to an EASY-nLC 1200 system (Thermo Fisher Scientific). Peptides were separated in a 40 min or 45 min gradient starting from 5% ACN, 0.1% formic acid to 95% ACN, 0.1% formic acid. MS spectra were recorded in the orbitrap at 120000 (at $m/z$ 200) resolution, scan range 400–1500 $m/z$, automatic gain control ion target value of 4e5, and maximum injection time of 50 ms. The intensity threshold was set to 5e3, included charge states to 2–7. Dynamic exclusion duration was set to 60 s or 20 s, respectively. Fragmentation was performed by collision-induced dissociation (CID) with 35% collision energy and ions were subsequently detected in the ion trap. Automatic gain control was set to 2e3 or 2e4 with a maximum injection time of 300 ms. The system was operated in data-dependent top-speed mode with a 3 s cycle time.

Raw files were analyzed using the Mascot 2.5.1 search engine MS/MS ions search (Matrix Science). Oxidation (M) was set as variable modification, as well as the triazole linkage resulting from click reaction with the additional G of Ub G76Aha which is still present after tryptic digest (K) $(+C(10)H(13)N(5)O(5);$ monoisotopic: 283.0917 Da, average: 283.2407 Da). Trypsin cleavage allowing up to two missed cleavages was chosen. For the analysis of peptide spectra of H1.2 K206Ub, which were generated by pepsin cleavage, no enzyme was chosen to allow for the search of every sub-sequence of the H1.2 sequence.

For the determination of full-length protein mass, recombinant proteins were analyzed on a micrOTOF II (Bruker) coupled to an Agilent 1200 HPLC system. Compass DataAnalysis 4.1 (Bruker) software was used for re-calibration and spectra deconvolution.

**Expression and purification of SIRT1**. The pET21a plasmid containing the coding sequence for SIRT1 with N-terminal deletion (669 aa (amino acids)/aa 1–3 + 82 − 747; the plasmid with human SIRT1 with flag tag was a kind gift from Eric Verdin, Addgene plasmid #13812[68]) and a C-terminal His$_6$-tag was transformed into E. coli BL21 (DE3). Cells were cultured at 37 °C in LB medium and at $OD_{600} = 0.6$–0.8, 1 mM IPTG was added. After 6 h, cells were harvested.

Cells were lysed by sonication in 50 mM Tris-HCl (pH 8.0), 250 mM NaCl, 2 mM $MgCl_2$, 1 mM β-mercaptoethanol, 1 mM PMSF, 100 μM Pefabloc SC, 1 μg/ml Leupeptin, 1 μg/ml Aprotinin, 10 μg/ml DNase I, 0.01% (v/v) Triton X-100. After centrifugation, SIRT1 was purified on a HisTrap FF column (GE Healthcare). Protein was eluted with a linear gradient with up to 500 mM imidazole. Elution fractions containing SIRT1 were pooled and dialyzed in 50 mM Tris base, 250 mM NaCl, 1 mM β-mercaptoethanol (pH 8.0), concentrated via ultrafiltration and further purified by size-exclusion chromatography (HiLoad 16/600 Superdex 75 pg). Purification was analyzed by SDS-PAGE and fractions with SIRT 1 were pooled and concentrated via ultrafiltration. Protein concentration was determined by BCA assay and SDS-PAGE with Coomassie blue staining. For crosslinking experiments, SIRT1 was used immediately; for other experiments, the proteins were stored in 10% glycerol at −80 °C.

**Circular dichroism (CD) spectroscopy**. CD spectroscopy was performed on a J-815 CD Spectropolarimeter (Jasco). Spectra were measured in a range of 250–190 nm with 0.1 nm data intervals and 200 nm/min scanning speed. Proteins were analyzed in a quartz cuvette of a 1 mm light path with a total volume of 200 μl. CD spectra were averaged from five scans. Histone H1 obtained from calf thymus (Calbiochem) served as a reference.

**HEK 293T cell culture and lysate preparation**. HEK 293T cells were cultured in DMEM supplemented with 10% (v/v) fetal bovine serum at 37 °C, 5% CO$_2$. Cells

were harvested by centrifugation for 10 min at 500×g, 4 °C, and washed with ice-cold 1× PBS. Pellets were frozen in liquid nitrogen and stored at −80 °C.

HEK 293T pellets were resuspended in 1× PBS, 2 mM MgCl$_2$, 1 mM DTT, 100 µM Pefabloc SC, 1 µg/ml Leupeptin, 1 µg/ml Aprotinin, incubated on ice for 10 min and lysed by sonication. The lysate was cleared by centrifugation (30 min, 21,885×g, 4 °C) and the protein concentration of the supernatant was determined by Bradford assay.

**H1.2 KxUb stability assay**. To determine the stability of the H1.2 KxUb conjugates in human cell lysates, 4.7 µM H1.2 and H1.2 KxUb were incubated in HEK 293T cell lysates (5 mg/ml). Samples were incubated at 4 °C, 25 °C, or 37 °C for up to 24 h and finally analyzed by western blot with anti-H1.2 (Abcam ab17677) in dilution 1:1000 and anti-Ub (BioLegend P4G7), dilution 1:1000, antibodies. For unprocessed scans see also the Source Data file.

**Identification of endogenous H1.2 posttranslational modifications in HEK 293T cells**. To identify endogenous posttranslational modifications (PTMs) of H1.2 in HEK 293T cells, H1.2 was enriched by immunoprecipitation followed by identification of potential PTMs and modification sites via LC-MS/MS. In the first step, the primary antibody H1.2 (Abcam ab17677) was immobilized (5 µg antibody/12.5 µl beads) on Pierce Protein G Magnetic Beads (Thermo Fisher Scientific). After washing, 0.5 mg HEK 293T cell lysate was added, incubated for 2 h at 4 °C and washed with 1× PBS, 2 mM MgCl$_2$, 1 mM DTT, 100 µM Pefabloc SC, 1 µg/ml Leupeptin, 1 µg/ml Aprotinin, 1x cOmplete Protease Inhibitor Cocktail. Samples were eluted in SDS-PAGE loading dye supplemented with 200 µM DTT and fractionated by SDS-PAGE followed by tryptic digest using trypsin and 2-chloroacetamide as alkylation agent.

Digested and desalted peptides were analyzed on an Orbitrap Fusion Tribrid mass spectrometer coupled to an EASY-nLC 1200 system. Peptides were separated in a 90 min gradient starting from 5% ACN, 0.1% formic acid to 35% ACN, 0.1% formic acid in 70 min, followed by 10 min to 45% ACN, 0.1% formic acid, and a washing step of 10 min at 80% ACN, 0.1% formic acid. MS spectra were recorded in the orbitrap at 120,000 (at m/z 200) resolution and scan range 300–1500 m/z, automatic gain control ion target value of 4e5, and a maximum injection time of 50 ms. The intensity threshold was set to 5e3, included charge states to 2–7. The exclusion duration was 45 s. Fragmentation was performed by CID in the ion trap using 35% collision energy. The automatic gain control was set to 2e3 with a maximum injection time of 300 ms. The system was operated in data-dependent top-speed mode with a 3 s cycle time. All samples were measured in technical duplicates.

Samples were analyzed with Proteome Discoverer 2.2.0.388 (Thermo Fisher Scientific). Dynamic modifications were set to oxidation (M), acetylation and methylation (K), attachment of GG or LRGG (K), and phosphorylation (S, T, or Y). Possible protein N-terminal modifications included were acetylation, M-loss, or M-loss with acetylation. The modification site-probability threshold was set to 75%. Results were further filtered for no more than three modifications per peptide and only unambiguous PSMs were considered.

**Identification of the H1.2 KxUb interactome by affinity purification-mass spectrometry (AP-MS)**. To identify protein-protein interactions, pulldown assays were performed with H1.2 and H1.2 KxUb conjugates. Samples using Ub as bait proteins or without any bait protein served as control. 2.5 mg HEK 293T cell lysate (preparation as described above) and 2.35 nmol bait protein in 500 µl total volume were incubated on ice for 10 min and centrifuged for 10 min at 21,885×g at 4 °C. The supernatant was added to 20 µl Strep-Tactin Superflow resin (IBA Lifescience) and incubated overnight at 4 °C in an overhead shaker. Beads were washed five times with 1× PBS, 2 mM MgCl$_2$, 1 mM DTT, 100 µM Pefabloc SC, 1 µg/ml Leupeptin, 1 µg/ml Aprotinin followed by elution of bound proteins in 2.5 mM desthiobiotin. Samples were analyzed by SDS-PAGE (with 0.02% input and flow-through, 0.05% first washing fraction, 1.5% last washing fraction, 2.75% elution fraction loaded), and Krypton staining (Thermo Fisher Scientific). For identification of co-purified proteins, elution fractions were freeze-dried, followed by in-solution digestion, desalting, and LC-MS/MS analysis. Each pulldown was performed in independent biological triplicates.

Tryptic peptides were separated on an EASY-nLC 1200 system at a flow rate of 300 nl/min using a 190 min gradient from 5% ACN, 0.1% formic acid to 35% ACN, 0.1% formic acid, and 10 min to 45% ACN, 0.1% formic acid followed by a washing step of 10 min at 80% ACN, 0.1% formic acid. Mass spectra were recorded on an Orbitrap Fusion Tribrid mass spectrometer operated in data-dependent top-speed mode with dynamic exclusion set to 45 s and a total cycle time of 3 s. Full scan MS spectra were acquired in the orbitrap at a resolution of 120,000 (at m/z 200) and scan range 300–1500 m/z with an automatic gain control ion target value of 4e5 and a maximum injection time of 50 ms. Most intense precursors with charge states of 2–7 and intensities greater than 5e3 were selected for MS/MS experiments using a linear ion trap and CID with 35% collision energy. Isolation was performed in the quadrupole with an automatic gain control ion target value of 2e3 and a maximum injection time of 300 ms. Each of the biological triplicates was measured in technical duplicate.

Raw files from LC-MS/MS measurements were analyzed using MaxQuant 1.6.1.0[69] with match between runs and label-free quantification (minimum ratio count set to one) enabled. The minimal peptide length was set to five with a mass tolerance of 4.5 ppm and 0.5 Da for parent ions and fragment ions, respectively (further settings: default). For protein identification, the human reference proteome downloaded from the UniProt database (download date: 2018-02-22) and the integrated database of common contaminants were used.

Further data processing was performed using Perseus 1.6.1.3 software[70]. Identified proteins were filtered for reverse hits, common contaminants, and proteins only identified by site. LFQ intensities were log2 transformed, filtered to be detected in at least five out of six replicates (three biological replicates, each measured as technical duplicates), and missing values were imputed from a normal distribution (width = 0.3 and shift = 1.8 for total matrix), based on the assumption that these proteins were below the detection limit. Significantly enriched proteins were identified by an ANOVA test (FDR = 0.001, S0 = 2), normalized by Z-scoring, and averaged. Finally, the enriched proteins were analyzed by hierarchical clustering (Euclidean distance) and plotted as a heatmap. Various clusters were identified and proteins plotted in profile plots. Ub-associated proteins were selected manually.

Proteins with a minimum Z-score of 0.3 were considered for further analysis in the respective groups (see also Supplementary Data 1). The network of interacting proteins was generated using Cytoscape 3.8.0 (Cytoscape Consortium). Further functional protein classification was examined by PANTHER 15.0 focusing on the 'Protein Class'. Graphs showing relative enrichment of single proteins (USP15, USP13, UCHL5, SIRT1) were generated using GraphPad Prism 6.01 (GraphPad Software). For statistical analysis, a one-way ANOVA (repeated measures) with Tukey's multiple comparisons test was performed ($\alpha$ = 0.05, 95% confidence interval). Only selected significance lines are shown in the graphs. The complete results of Tukey's multiple comparisons test are depicted in graphs showing differences between group means (Supplementary Fig. 5).

**Immunoblotting**. For orthogonal validation of protein-protein interactions by immunoblotting, HEK 293T cell lysate and samples resulting from the affinity purification assay were analyzed by western blot. Primary antibodies were directed against CUL4A (dilution 1:1000, Bethyl Laboratories A300-739A), IWS1 (1:1000, Bethyl Laboratories A304-609A), PAF1 (1:1000, Bethyl Laboratories A300-172A), DICER1 (1:500, Abcam ab227518), SIRT1 (1:5000, Abcam ab32441), SET (1:10000, Abcam ab181990), NPM1 (1:1000, Abcam ab10530), RAD23A (1:10000, Abcam ab102593), USP13 (1:1000, Abcam ab109264), WRNIP1 (1:2000, Abcam ab99316), H1.2 (1:1000, Abcam ab17677), PPP6R3 (1:2000, Novus Biologicals ABIN188586), SAP30BP (1:2000, Novus Biologicals ABIN4351992), SART3 (1:1000, Abbexa ABIN5998745), BRCC3 (1:1000, antibodies-online ABIN1586883), PCNA (1:2000, antibodies-online ABIN152935), CDC27 (1:500, in-house generation), FZR1 (1:500, in-house generation).

For the co-immunoprecipitation analysis of potential interactors, endogenous H1.2 was enriched from HEK 293T cells as described above. As a control sample, a rabbit IgG isotype control antibody (Thermo Fisher Scientific 10500 C, lot TH275005) was immobilized. If probed for CDC27, beads were additionally incubated before elution in 1× PBS, 1 mM MnCl$_2$, lambda protein phosphatase, 1× cOmplete Protease Inhibitor Cocktail (Roche) for 40 min at RT. Finally, elution fractions were analyzed by western blot and Ponceau S (Merck) staining. For western blot analysis, primary antibodies were directed against H1 (AE-4, dilution 1:1000, Santa Cruz sc-8030), CDC27 (1:500, in-house generation), PPP6R3 (1:2000, Novus Biologicals ABIN188586), SIRT1 (10E4, 1:1000, Merck 04-1557), NPM1 (1:1000, Abcam ab10530) and p150 (1:1000, BD Transduction Laboratories 610473).

**Ubiquitylation assay**. In vitro ubiquitylation assays were performed with Ub-activating enzyme E1, UBA1, Ub-conjugating enzyme E2, UBCH5B, and E3 ligase HUWE1 (catalytic domain, 1147 aa/aa 3228–4374). 8.2 ng/µl E1, 8.3 ng/µl E2 and 25 ng/µl E3 were incubated with 0.3 µg/µl Ub (Sigma-Aldrich) (Ub (SA)) in 2 mM ATP, 2 mM MgCl$_2$, 1 mM DTT, 25 mM Tris-HCl (pH 7.4), 50 mM NaCl. For H1.2 ubiquitylation, 4.7 µM H1.2 were added and samples incubated for 20 min at 30 °C. Ubiquitylation of H1.2 KxUb click conjugates was performed using 0.5 µM H1.2 KxUb (or H1.2) and incubation for 90 min at 30 °C. The reactions were stopped by the addition of SDS-PAGE loading dye and heating to 95 °C for 5 min and analyzed by SDS-PAGE and Coomassie blue staining.

For the determination of ubiquitylation sites, triplicate samples were fractionated by SDS-PAGE, digested by in-gel digestion, desalted, and analyzed by LC-MS/MS.

Peptides were analyzed on a QExactive HF Hybrid Quadrupole-Orbitrap (Thermo Fisher Scientific) operated with Tune 2.9 (Thermo Fisher Scientific) and coupled to an EASY-nLC 1200 system. The gradient for peptide separation started from 5% ACN, 0.1% formic acid to 35% ACN, 0.1% formic acid in 45 min, followed by 5 min to 45% ACN, 0.1% formic acid, and a washing step of 80% ACN, 0.1% formic acid. The mass spectrometer was operated in data-dependent Top20 mode with dynamic exclusion set to 40 s. Full scan MS spectra were acquired at a resolution of 120,000 (at m/z 200), scan range 350–1600 m/z with an automatic gain control target value of 3e6 and a maximum injection time of 60 ms. Most intense precursors with charge states of 2–6 reaching a minimum automatic gain

control target value of 1e3 were selected for MS/MS experiments. The normalized collision energy was set to 28%. MS/MS spectra were collected at a resolution of 30,000 (at $m/z$ 200), an automatic gain control target value of 1e5 and 100 ms maximum injection time. Each of the triplicates was measured as a technical duplicate.

Raw files were analyzed using Proteome Discoverer 2.2.0.388. Dynamic modifications were set to oxidation (M), attachment of GG (K), and acetylation and addition of GG were set as possible N-terminal modifications. Carbamidomethylation (C) was set as a static modification. Results were filtered for PTM site probability of 99–100% and high confident PSMs. Furthermore, GG-sites were only considered if they were reliably detected in at least four out of six replicates.

**Deubiquitylation assay**. For deubiquitylation of H1.2, ubiquitylated proteins were first generated by in vitro ubiquitylation assay as described above and the reactions stopped by the addition of 10 mM EDTA and incubation on ice for 5 min. Then, the deubiquitylating enzymes (DUBs) *USP15* (5–50 nM), *USP13* (200 nM–2 μM), or *UCHL5* (50–500 nM) (Boston Biochem) were added and incubated at 37 °C for up to 2 h. For time-course analysis, 50 nM *USP15*, 1 μM *USP13*, or 300 nM *UCHL5* were used, respectively. Reactions were stopped by the addition of SDS-PAGE loading dye and heating for 5 min at 95 °C before samples were analyzed by SDS-PAGE and Coomassie blue staining.

**Deacetylation assay**. For deacetylation of p53 K370AcK as substrate protein, 1 μM *SIRT1* was preincubated for 10 min on ice with 10 μM H1.2 or H1.2 K64Ub. Subsequently, *SIRT1* and H1.2 were added to 1 μM p53 K370AcK in 50 mM Tris-HCl (pH 8.0), 150 mM NaCl, 1 mM MgCl$_2$ supplemented with 3 mM fresh NAD$^+$ and incubated at 37 °C for the indicated time points. Reactions were stopped by the addition of SDS-PAGE loading dye and heating for 5 min at 95 °C. Deacetylation was analyzed by western blot with anti-p53 K370AcK (dilution 1:1000, Abcam ab183544), anti-p53 (DO-1, 1:750, Calbiochem OP43-20UG), anti-His HRP conjugated (1:5000, Sigma-Aldrich A7058-1VL) antibodies and Ponceau S (0.1%) staining.

The assay was performed in triplicates, lane intensities were quantified (normalized to the 0 min lane and referenced to the amount of p53) with ImageQuant TL 8.1 (GE Healthcare) and analyzed with GraphPad Prism 6.01. For statistical analysis, a two-way ANOVA with Tukey's multiple comparisons test was performed ($\alpha = 0.05$, 95% confidence interval) with *$p \leq 0.05$ indicating the significant difference of measurements between samples (+H1.2) and (+H1.2 K64Ub) at $t = 30$ min.

**Crosslinking mass spectrometry (XL-MS)**. To investigate interactions of H1.2 and H1.2 KxUb with *SIRT1* on a structural level XL-MS was performed essentially as described[71]. In short, unmodified H1.2, H1.2 K64Ub, Ub (SA) alone, or H1.2 together with Ub (SA) were crosslinked with *SIRT1*. The molar ratio of histones/Ub and *SIRT1* was 1:1. In total, 80 μg or 87.2 μg of protein was incubated in 1× PBS on ice for 5 min in a total volume of 80 μl. Then, BS³-H12/D12 (Creative Molecules) was added to a final concentration of 2.75 mM, incubated on ice for 10 min, followed by 20 min at 30 °C, 600 rpm. Finally, the reaction was quenched by the addition of 1 M NH$_4$HCO$_3$ to a final concentration of 50 mM and incubation for 10 min at 30 °C and shaking at 600 rpm followed by analysis by SDS-PAGE and Coomassie blue staining. Each experiment was performed in three independent biological replicates (i.e., separate protein batches).

After crosslinking, samples were vacuum concentrated followed by in-solution digestion with trypsin overnight and desalting with C18 cartridges. Next, samples were again concentrated in a vacuum concentrator, dissolved in 30% ACN, 0.1% TFA, and crosslinked peptides enriched by peptide size-exclusion chromatography (Superdex 30 Increase 3.2/300) (GE Healthcare). Elution fractions of enriched crosslinked peptides were vacuum concentrated and finally dissolved in 5% ACN, 0.1% formic acid for LC-MS/MS analysis (final volume adjusting 100 mAU in chromatogram to 10 μl).

Peptides were separated on an EASY-nLC 1200 system at a flow rate of 300 nl/min. The gradient started with 4 min at 5% ACN, 0.1% formic acid followed by a linear gradient of 45 min up to 35% ACN, 0.1% formic acid, and a washing step of 11 min at 80% ACN, 0.1% formic acid. Mass spectra were recorded on an Orbitrap Fusion Tribrid mass spectrometer as described above for AP-MS experiments but with dynamic exclusion set to 60 s. Full scan MS spectra were acquired in the orbitrap at scan range 400–1500 $m/z$ and automatic gain control ion target value set to 2e5. Most intense precursors with charge states of 3–8 and intensities greater than 5e3 were selected for MS/MS experiments using a linear ion trap. Isolation was performed in the quadrupole with an automatic gain control ion target value of 1e4 and a maximum injection time of 35 ms. Each of the biological triplicates was measured in technical duplicate.

Data were searched using xQuest 2.1.3 in ion-tag mode with a precursor mass tolerance of 10 ppm. For matching of fragment ions, tolerances of 0.2 Da for common ions and 0.3 Da for crosslink ions were applied. Carbamidomethylation (C) was used as a static modification. Enzyme specificity was set to trypsin, allowing up to three missed cleavages. Samples that contained unmodified H1.2 or H1.2 K64Ub were searched against a database containing the sequence of Ub

G76M, samples with Ub (SA) were searched against the wild-type Ub sequence. In addition, all databases contained the sequences of *SIRT1* including a C-terminal His$_6$-tag, and H1.2 including a Strep-tag II and His$_6$-tag. For visualization in 2D plots, target crosslinks were filtered for ld score > 30, deltaS < 0.95, FDR < 0.05, and only unique crosslinking sites identified in all three biological replicates are shown.

**Integrative structural modeling**. We used the Integrative Modeling Platform (IMP)[72] for modeling the interactions of H1.2: *SIRT1* and H1.2 K64Ub: *SIRT1* resulting from XL-MS experiments (see above). The approach using crosslinks as restraints in a Bayesian scoring scheme is described in detail in[58]. Accordingly, there are four main steps: (A) gathering of data, (B) representation of subunits and translation of the crosslinking data and the prior knowledge into a Bayesian scoring function, (C) configurational sampling to produce an ensemble of models that minimize the Bayesian scoring function and (D) analysis of the ensemble. IMP allows for coarse-grained modeling, i.e., the inclusion of different resolution levels into a model. Different resolution levels are represented by accordingly sized beads (spheres) in a modeling run. In the Bayesian scoring scheme models are ranked according to their likelihood and prior probability. The score is the negative logarithm of their product. The likelihood contains the crosslinking data, while the priors contain information about sequence connectivity in protein chains, as well as the excluded volume between all pairs of beads. The likelihood is defined through a forward model, which quantifies the probability of the formation of a crosslink given the distance between two residues in the model, as well as a noise model, which weighs the deviation between observed crosslinks and the forward model.

For our modeling runs, we employed the crystal structures of the chicken H1 GD (PDB: 1ghc; the first model used), *SIRT1* (PDB: 4ig9; chain A and B used), and Ub (PDB: 1ubq). All residues found in the crystal structures were represented as one residue per bead and constrained into rigid bodies. The following aa not found in the crystal structures were represented by flexible beads (20 aa per bead): for H1.2 aa 27–44 and aa 120–221 and for *SIRT1* aa 424–552. These flexible regions were chosen as they contain crosslinks; this hybrid approach of mixing rigid beads (known crystal structure) and flexible beads (unknown structure) allowed us to incorporate all experimental crosslinks into our modeling approach and enabled us to include long unstructured and to date non-crystallized protein domains to be included in our models. Ub was completely represented by the crystal structure (aa 1–76). The crosslink input databases for the modeling included all links with ld score ≥ 20, deltaS < 0.95, FDR < 0.05 which were found in at least one out of the three biological replicates. From each replicate, the highest-scoring link was chosen. This means a crosslink found in all biological replicates had a weight three times higher than a crosslink only found in one biological replicate. Crosslinks were classified by ld score into three classes.

The actual models were computed by Replica Exchange Gibbs sampling, based on Metropolis Monte Carlo sampling. The Monte Carlo movements included random translation and rotation of rigid bodies with a maximum of 10 Å and 1 radian, respectively.

The sampling was run for 15000 frames with 32 replicas, with temperatures ranging from 1.0 to 2.5 (technical units). The 25% best scoring models were saved, resulting in 120000 saved models overall. These replicas were run in three independent sampling runs with random initial configurations to assess convergence and accounting for 360000 saved models overall. The models from all three sampling runs were pooled and the overall 500 best scoring models were clustered using the root-mean-square deviation (rmsd) of H1.2 and Ub (when Ub was included in the sampling). *SIRT1* was used as a reference for alignment, as its position was fixed during the sampling (except for the flexible beads regions). The rmsd cut-off for clustering was set to 10 Å. The cluster center was defined as the cluster member with minimum rmsd concerning the other members. It was used to represent the atomistic coordinates of the system. The overall precision of each protein was calculated as the root-mean-square fluctuation concerning the cluster center. Furthermore, to represent the variance of the solutions, the superposed structures of each cluster were converted into a localization density. Note that the localization densities are significantly bigger than the crystal structures due to the inclusion of the flexible beads.

For H1.2: *SIRT1*, the sampling converged onto one main cluster while we found two main clusters for the run of H1.2 K64Ub: *SIRT1*. To assess convergence, all replicates were also clustered independently resulting in the same main clusters as the pooled clustering for H1.2: *SIRT1*. For H1.2 K64Ub: *SIRT1* the first two replicates preferentially converged to either of the main clusters while the third replicate contained both clusters. The average precision of H1.2 for the H1.2: *SIRT1* main cluster was ~5 Å and for the two main clusters of H1.2 K64Ub: *SIRT1*, it was ~3/6 Å (H1.2) and ~8/6 Å (Ub). *SIRT1* served as the alignment reference.

**Nucleosome and chromatosome assembly**. The 12-mer DNA array contained twelve repeats of a 207 bp DNA fragment based on the 601 nucleosome positioning sequence[63] cloned into the pUC18 vector by unit assembly:

5'-CTAGTTCGGACCCTATACGCGGCCGCCCTGGAGAATCCCGGTGCC
GAGGCCGCTCAATTGGTCGTAGCAAGCTCTAGCACCGCTTAAACGCACG
TACGCGCTGTCCCCCGCGTTTTAACCGCCAAGGGGATTACTCCCTAGTCT
CCAGGCACGTGTCAGATATATACATCCTGTGCATGTGGATCCGAATACA
TATTAATTAATACG-3'

The DNA array was purified from *E. coli* MDS42 ΔrecA using the Plasmid Plus Maxi Kit (Qiagen). Purified DNA was digested by FastDigest Eco32I and FastDigest DraI (Thermo Fisher Scientific) for 30 min at 37 °C, resulting in the linearization of the DNA array and digestion of the plasmid DNA into three fragments used as competitor DNA. Digested DNA fragments were purified by QIAEX II suspension (Qiagen).

Nucleosome arrays were reconstituted by stepwise salt dialysis. 0.14 μM DNA array, recombinant chicken histone octamer (Abcam) and 0.13 μg/μl BSA (Thermo Fisher Scientific) were dialyzed in 10 mM Tris-HCl (pH 7.6), 2 M NaCl, 1 mM EDTA, 0.05% (w/v) Nonidet P-40 Substitute, 1 mM β-mercaptoethanol followed by stepwise reduction of the NaCl concentration to 0.23 M NaCl in 1 h intervals. Finally, nucleosomes were dialyzed overnight at 4 °C in 10 mM Tris-HCl (pH 7.6), 50 mM NaCl, 1 mM EDTA, 0.05% (w/v) Nonidet P-40 Substitute. The DNA/octamer ratio was determined by titration. Chromatosomes were reconstituted by incubation of nucleosomes with 1.5x molar excess of H1.2 or H1.2 KxUb conjugates for 25 min at 19.5 °C and analyzed by agarose gel electrophoresis.

**Analysis of nucleosome and chromatosome arrays.** *PacI digest*: To analyze saturation of the nucleosome array and evaluate H1.2-binding to the nucleosome, 79 fmol arrays were digested with 5 U of the restriction enzyme PacI (NEB) in 10 mM HEPES/KOH (pH 7.6), 50 mM KCl, 1.5 mM MgCl₂, 0.5 mM EGTA for 1 h at 37 °C. Next, samples were analyzed by agarose gel electrophoresis followed by ethidium bromide staining.

*MgCl₂ precipitation*: For analysis of array-compaction, 95 fmol nucleosome and chromatosome arrays were incubated with increasing concentrations of MgCl₂ (up to 6 mM MgCl₂) for 15 min on ice. After centrifugation (30 min, 15,000×*g*, 4 °C), the supernatant was analyzed by agarose gel electrophoresis and visualized with ethidium bromide. The soluble 12-mer array was then quantified densitometrically with ImageQuant TL. The assay was performed in triplicates and analyzed with GraphPad Prism 6.01. For statistical analysis, the area under the curve was determined and a one-way ANOVA with Tukey's multiple comparisons test was performed ($\alpha = 0.05$, 95% confidence interval).

*Micrococcal Nuclease (MNase) digestion*: 57 fmol array was digested with 0.022 U Micrococcal Nuclease (NEB) in 50 mM Tris-HCl (pH 7.9), 5 mM CaCl₂ supplemented with 100 μg/ml BSA (NEB). The reaction was incubated up to 40 min at 37 °C and quenched by the addition of 0.4% (w/v) SDS, 10 mM EDTA. After incubation for 10 min on ice, proteins were digested by 0.1 mg/ml proteinase K (Roth) for 1 h at 37 °C. After agarose gel electrophoresis and ethidium bromide staining, undigested 12-mer array DNA was quantified densitometrically by ImageQuant TL. The software's 1D gel analysis mode was used for background subtraction and to determine the intensities of all undigested array bands. Intensity values were normalized to the 0 min-time point and plotted over time. The assay was performed in triplicates and analyzed with GraphPad Prism 6.01. For statistical analysis, the area under the curve was determined followed by a one-way ANOVA with Tukey's multiple comparisons test ($\alpha = 0.05$, 95% confidence interval).

**Phase separation assays.** H1.2 and H1.2 KxUb conjugates were labeled with Alexa Fluor 488 NHS-Ester (Thermo Fisher Scientific) by incubating 2–5 mg/ml of protein with the dye dissolved to 10 mg/ml in DMSO in 100 mM NaHCO₃ (pH 8.3) while shaking at 4 °C in the dark. After incubation overnight, the reaction was quenched by the addition of Tris-HCl (pH 8.3) to a final concentration of 20 mM for 1 h at RT. Excess of dye was removed by ultrafiltration. For microscopic experiments, labeled proteins were mixed with the respective unlabeled protein in a molar ratio of 1:50.

Fluorescently labeled DNA (sequence see below) was generated by PCR using a reverse primer labeled at the 5' end with Atto 390 fluorescent dye (biomers.net). The 601 DNA sequence was used as a template. Finally, DNA was purified by ethanol precipitation and for microscopic experiments, labeled DNA and unlabeled DNA were mixed in a molar ratio of 1:1.

5'-CTATACGCGGCCGCCCTGGAGAATCCCGGTGCCGAGGCCGCTCA ATTGGTCGTAGCAAGCTCTAGCACCGCTTAAACGCACGTACGCGCTG TCCCCCGCGTTTTAACCGCCAAGGGGATTACTCCCTAGTCTCCAGGCAC GTGTCAGATATATACATCCTGTGCATGTGGATCCGAAT-3'

For microscopic analysis of droplet formation, samples were mixed at RT directly before analysis. Unless otherwise stated, 2.5 μM proteins were mixed with 2.5 μM DNA in 1× PBS in 10 μl total volume and transferred to a 384 Well Non-binding Microplate (μCLEAR, black) (Greiner Bio-One). For chromatosome condensates, a 1.5-molar excess per nucleosomal binding site of the H1.2 or H1.2 KxUb variants was mixed with 218 nM 12-mer nucleosome array in 5 μl sample volume. Samples were analyzed from four independent experiments (generated from two independent protein batches and labeling) with three positions per well. For FRAP (fluorescence recovery after photobleaching) analysis, 5 μM histones and 5 μM DNA were used. Recovery curves for DNA and histones were generated from three independent droplets per channel and sample in two replicates.

*Microscopy*: Confocal fluorescence microscopy images were captured on a Zeiss CellObserver HS Spinning Disk microscope with a ×63/1.40 Plan-Apochromat oil immersion objective and a Photometrics EVOLVE 512 imaging device at three randomly selected positions per well (sample). FRAP analyses were performed on a Zeiss LSM 700 AxioObserver using a ×63/1.40 Plan-Apochromat oil immersion objective. A circular region of interest (ROI) with a 10% diameter was bleached in

the middle of the respective condensate to averagely 25%/40% of the original intensity for bleaching with 405 nm or 488 nm. 200 × 200 pixels scaled to 0.5 μm × 0.5 μm were collected with a pixel dwell time of 6.53 μs at 405 nm and 1 s intervals for 30 s followed by 5 s intervals for 400 s and for excitation at 488 nm 1 s intervals for 40 s and subsequent 5 s intervals over 550 s with a pixel dwell time of 8.08 μs. Before bleaching, ten 1 s intervals were collected for data normalization.

*Quantification and statistical analysis*: Image analysis was performed with Fiji (ImageJ 2.0.0-rc-69 with Java 1.8.0_201, 64-bit). Microscopy data processed by Fiji was analyzed using the R Statistical Package 3.6.3 and plotted using the ggplot2 package.

To describe phase separation behavior of histones and DNA, the amount of condensed protein was defined as the ratio of $I_{droplet}$ to $I_{outside}$, with $I_{droplet}$ as the integrated intensity inside the droplets and $I_{outside}$ as the integrated intensity outside the droplets. To dissect the droplets, a mask of the condensates was built by thresholding the images according to[73] with a radius of 15 and default values for parameter 1 and 2, a filter was applied to reduce noise detection and only objects with a minimum size of 4 pixels and a circularity value ≥0.85 were considered as condensates. Images for analysis were taken 20 min after sample mixture and application to the microplate. For statistical analysis, a two-sample *t*-test was performed. The obtained values of the condensates containing ubiquitylated variants of H1.2 were compared to the non-modified H1.2, respectively.

All points in the fluorescence recovery curves were normalized to the pre-bleach intensity and corrected for photobleaching by multiplication with a factor $f_t$ determined by the acquisition of a reference condensate within the same image as the bleached condensate, according to

$$f_t = I_{Ref\_pre}/(I_{Bleach\_pre} \cdot I_{Ref,t}) \quad (1)$$

with $I_{Ref\_pre}$ and $I_{Bleach\_pre}$ defined as the mean intensity of reference or bleached ROI before bleaching and $I_{Ref,t}$ as the mean intensity of the reference ROI at time point *t*. The mean intensity of the bleached ROI was subtracted from all data points. The obtained FRAP curves were fitted to a self-starting nonlinear least-square asymptotic regression model with

$$y = y_{\infty} + (y_0 - y_{\infty}) \exp(-\exp(\ln(k)))t \quad (2)$$

The rate constant *k* and the maximum relative recovery $y_{\infty}$ were calculated for each FRAP curve. For statistical analysis, a two-sample *t*-test was performed. The curve parameters of the condensates containing H1.2 KxUbs were compared to the non-modified H1.2, respectively.

**Statistics and reproducibility.** Generation of ubiquitylated H1.2 via click reaction was independently performed several times for all histone variants (H1.2 KxPlk) with $n > 20$. The stability of the H1.2 KxUb conjugates in human cell lysates was tested in two independent replicates with similar results. Ubiquitylation assay using H1.2 KxUbs from click reaction as substrate proteins was repeated twice. AP-MS experiments were performed in triplicates. Co-immunoprecipitation analysis of potential interactors of endogenous H1.2 was performed in duplicates. Enzymatic ubiquitylation of H1.2 and deubiquitylation assays were performed in duplicates. Deacetylation analysis of *SIRT1* and model substrate protein p53 K370AcK were performed in triplicates. XL-MS experiments were carried out in triplicates. MNase digestion and MgCl₂ precipitation assays were performed in triplicates with independently assembled chromatosome arrays. The EMSA of the nucleosome and the chromatosome arrays were performed in triplicates. The EMSA with titrated H1.2, PacI digests, and SDS-PAGE analyses were repeated at least twice. Phase separation assays were carried out in four independent experiments and FRAP condensate assays in six independent experiments. For details of statistical analysis see respective methods section and figure legends.

## Data availability

The data that support this study are available from the corresponding authors upon reasonable request. The MS raw data including all databases used in this study have been deposited to the ProteomeXchange Consortium via the PRIDE[74] partner repository with the dataset identifier PXD025258. Figures with associated raw data: Figs. 1–3 and Supplementary Figs. 1–7. Source data are provided with this paper.

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

## Acknowledgements
The authors thank the current and former members of the Marx, Stengel, and Scheffner labs for valuable discussions. We thank K. Stuber and F. Offensperger (Departments of Chemistry and Biology, University of Konstanz) for providing purified E1/E2/E3 enzymes. We also thank A. Marquardt and A. Sladewska-Marquardt (Proteomics Center, University of Konstanz) for discussions and helpful tips. We also thank the team of the Bioimaging Center, University of Konstanz, and especially M.T. Stöckl for help with microscopy experiments. This work was supported by the DFG (SFB969, SPP1623). E.H., S.G., and M.L.N. thank the Konstanz Research School Chemical Biology for support by a fellowship. S.G. and M.L.N. acknowledge the Zukunftskolleg of the University of Konstanz for a doctoral fellowship. F.S. is grateful for funding from the DFG Emmy Noether Program (STE 2517/1-1).

## Author contributions
E.H., S.G., M.S., A.M., and F.S. conceived the study and experimental approach; E.H. generated H1.2 and H1.2 KxUb conjugates. E.H. performed AP-MS and XL-MS experiments; E.H. und M.L.N. carried out phase separation experiments; S.G. conducted nucleosome and chromatosome array experiments. K.M.K. carried out the integrative modeling; S.M.K. performed deacetylase assays. E.H., S.G., M.L.N., K.M.K., D.R., M.S., A.M., and F.S. analyzed the data, and E.H., M.S., A.M., and F.S. wrote the paper with input from all authors.

## Funding

## Competing interests
The authors declare no competing interests.
