## [Peer Review File · Nature Communications]

REVIEWER COMMENTS

Reviewer #1 (Remarks to the Author):

In the present manuscript, Höllmüller and colleagues investigate possible functions of linker histone H1.2 ubiquitylation at three different sites. Using an elegant method previously developed in the same labs, the authors generate recombinant H1.2-ubiquitin analogs and compare them in a number of *in vitro* assays:

- interactors of three recombinant H1.2-ubiquitin analogs versus unmodified H1.2 in cell lysates were determined by proteomics
- One candidate interaction with histone deacetylase SIRT1 was characterized using enzymatic assays and crosslink-mass spec (XL-MS)
- liquid-liquid phase condensate formation assays with DNA were performed
- *In vitro* chromatosome assembly was performed and resulting chromosomes were analyzed for their stability and resistance to nuclease

Together, the results of the assays support a role for H1.2 ubiquitylation in the opening of chromatin and possibly promoting a more active state, thus counteracting to some extent the overall function of H1.2.

The experiments, data, and data representation are generally of high quality and the results are well presented. Text is well-written and appropriately discussed on the background of the existing literature. Thus, the manuscript is certainly well publishable with respect to all technical standards (only technical concerns see Fig 3F discussed below).

However as discussed below I would argue the state of the experiments is premature for publication in Nature Communications. My main concern is the lack of more definite experiments, particularly experiments performed in cells, that could support the conclusions drawn towards the function of H1.2 in a cellular system. I will discuss this in more detail below, but one important note: albeit statistically significant differences between ubiquitinated and unmodified H1.2 are seen in the functional *in vitro* experiments presented in Fig 3, 4, 5, the effect sizes are relatively small. I think that going from a small effect size *in vitro* to the hypothesis “that site-specific ubiquitylation of H1 has a general regulatory role and promotes a transcriptionally active chromatin state” is premature in the absence of additional experiments.

Note that the part “H1 has a general regulatory role” in the abstract is a relatively hollow statement, and at several instances in the manuscript similar non-descriptive wording is used such as “X affects Y”, “X modulates Y” without specifying the detail or direction of the effect.

Going through the presented data by Figure:

Fig 1: Synthesis of recombinant ubiquitylated H1.2 is elegant and, of course, exact due to the precise bioorthogonal chemistry used. The triazole linkage is considered a good mimetic of the Lys(GlyGly) isopeptide bond in the field, and thus I would argue that the experiments with these recombinant H1.2-ubiquitin analogs are robust and meaningful with respect to the natively ubiquitylated H1.2. In addition, some of the experiment would be potentially impossible to do with a native H1.2 due to concurrent deubiquitylation reactions in cellular lysate.

The choice of three sites is rationalized in the manuscript but it is of course always a question if the most functionally relevant sites can be known *a priori*. At least ten sites are known to exist, according to ref 49 (<https://doi.org/10.1074/mcp.M111.013284>) Supplementary Figure 3: K17, K34, K46, K64, K75, K85, K90, K97, K106, K206.

To my understanding, nothing is known about the relative abundance of mono-ubiquitylation at all the known sites, which would at least be one variable to take into account when choosing sites. A highly abundantly modified site may be considered more consequential *in vivo* than a site that is barely detectable by sensitive mass spec.

Fig 2: Interaction partners of H1.2-ubiquitin species. This is a very interesting experiment and it comprehensively surveys interaction partners that are either sensitive or insensitive to ubiquitylation. However, I am missing here some crucial, more physiological, validation of the key interaction partners highlighted in the figure and text. Pulldowns from lysates are accepted in the field and have in a myriad of cases lead to the identification of relevant functional interactions in the cell, but it is to expect that a subset of the hits are actually false-positives, since the stoichiometries and binding conditions are not really comparable to an endogenous setting. E.g. some interactions partners may simply pull down through favorable but unspecific hydrophobic or electrostatic interactions. Thus, validation is needed! While I fully agree with the authors that there is no equivalent clean experiment that can be done in cellulo, i.e. it seems not possible to pull out ubiquitylation-specific interaction partners on endogenous H1.2. However, I could think of a number of simple experiments, each being better than no validation experiments at all:

- A co-IP from cell lysates using anti-H1.2 antibody
- a co-IP of transiently transfected, epitope-tagged H1.2
- a co-IP from cells stably expressing epitope-tagged H1.2
- any above performed after formaldehyde crosslinking to stabilize additional transient interactions

All of the above would clearly be expected to co-purify at least a subset of interactors found in the lysate pulldown. There are some details to figure out here, e.g. what an appropriate control would be (another H1 variant? A core histone?). Interactions via chromatin need to be removed (e.g. by an adequate salt concentration to remove H1 from chromatin and then work with supernatant). But importantly, any interactors validated through this route will be much more believable since they would be proven to be present in cellulo.

Of course the Co-IP would occur from a mixed H1.2 species, thus it would not be possible to distinguish specificity of the interactors to unmodified H1.2, ubiquitylated H1.2, or H1.2 modified in any other way. However, expecting that the top interactors found here are also indeed relevant to H1.2's function in the cell, clearly those hits should show up. And assuming for a moment that ubiquitylation was a reasonably abundant modification on H1.2, the Co-IP should also pull down ubiquitylation-specific binding partners.

And it should even be possible to validate these ubiquitylation-specific interactors:

- a pulldown of transiently or stable transfected, epitope-tagged H1.2 with the specific K->A or K->R mutation(s) should remove those interactors that require the presence of site-specific ubiquitylation. K->A/R mutations also abrogate any other modification at the site (e.g. acetylation), but since the main and only important point of this experiments is to validate the role of ubiquitylation I believe this is not a problem.
- a pulldown of transiently or stable transfected, epitope-tagged H1.2 with all ubiquitylation sites mutated to A or R should further remove any of the ubiquitylation-specific interactors.

I believe that a selection of experiments described above would be crucial validation and in particularly necessary for implying a functional role for the SIRT1 interaction.

Fig 3: Modulation of SIRT1 activity by H1.2 K64ub. Following up on SIRT1 found in the MS in Fig 2, the authors continue to characterize this putative interaction in vitro. As said above, my instinct would have been to first validate that SIRT1 can be a H1.2 interactor in vivo/in cellulo, and this could have again been easily done by

- co-IP of endogenous H1.2 or SIRT1 and western for the respective other
- Transient co-transfection of epitope-tagged SIRT1 and H1.2 (including H1.2 K56A etc. control(s)) and reciprocal pulldown.

Only if at least some hint for an interaction to occur in the cell, then Fig 3 makes sense. However I do have some minor issues:

- Fig 3B: if it is relevant to know if the same DUBs are also acting on H1.2 in vivo, then epitope-tagged H1.2 could be transiently transfected into cells +/- siRNA against the DUB, then western blot for the epitope.

Maybe in the control condition there would already be additional bands appearing for the epitope-tagged H1.2, corresponding to ubiquitylated species running at higher MW. And upon knockdown of a relevant DUB, those bands should become more intense or appear.

– Fig 3D: XL-MS. I am not an expert in this, so I assume that crosslinking between proteins that do not interact also doesn't occur at a significant level. Or to put in a question: is peptide-level evidence XL-MS considered enough in the field to imply a direct biochemical interaction? Could the complex shown to be stable on a gel filtration column? By pulldowns using the recombinant interaction partners?

– Fig 3E is nice and I believe the XL-MS technology is elaborate enough to allow fairly precise models, but would it be possible to take this information, make a set of mutations, domain deletion, and check if the interaction is indeed disrupted by manipulating the predicted interface?

and one major issue:

– Fig 3F is the only panel where I can't attest the otherwise high data quality. The quantification in Ext Data Fig 5 shows just one significant difference at the 30 min time point, so any effect of H1.2 or H1.2ub is marginal in the shown experiment. But why are the signals anti-p53 K370AcK and anti-p53 so much higher in the middle assay "+ H1.2 K64Ub"? Shouldn't be the first two lanes of each assay, "- NAD+" and "- SIRT1" be exactly the same? Otherwise how can the assays be quantitatively compared, if the starting amounts aren't even the same? Especially since HRP-western has been used, the signal is known not to be linear, which means that essentially one cannot and should not make a quantitative comparison across assays with vastly different starting intensities. Since these blots are the entire basis for the conclusion that "H1.2 ubiquitylation affects SIRT1 activity", I believe this conclusion needs to be corroborated.

Fig 4: Phase condensates: I am not an expert in this, certainly phase condensates are 'in' and clearly are relevant for regulation of chromatin. The assays seem performed to the standards of the field. I understand there is no assay that could test any of this behaviour in cells. Maybe transiently overexpression of epitope-tagged H1.2 and K->A mutants thereof followed by IF could say something about how H1.2 partitions in the nucleus (if it is not just relatively diffuse, in which case not much would be to learn...)

Fig 5: Chromosomes: Data convincingly shows that H1.2 makes arrays more inaccessible and ubiquitylation can reverse some of these effects, with some site-specificity. Could the authors do a complementary experiment where chromosomes are formed with unmodified H1.2 and subsequently ubiquitylated enzymatically and subsequently subjected to the MNase assay? Or if the reaction conditions were incompatible with the chromosome, then simply, enzymatically ubiquitylated H1.2 could be used side-by-side with the recombinant site-specific analogs. This would simply provide some additional evidence that the effect is also true for the 'real' ubiquitylation linkage and in general robust.

Reviewer #2 (Remarks to the Author):

In this manuscript, the authors reported the chemical synthesis of site-specific, non-hydrolysable mono-ubiquitylated H1.2 variants and their modification-dependent interactome. The functional relevance of some representative interactions was further examined on three deubiquitylation enzymes and deacetylase SIRT1. Additional biochemical assays on condensate formation and chromosome assembly suggested a direct role of H1 ubiquitylation on the regulation of chromatin structure beyond modulating protein-protein interactions. Overall, the functional characterization on linker histone H1 ubiquitylation seems thorough with multiple approaches and the scope suitable for Nature Communications. However, some experimental details need to be clarified and discussions included to ensure scientific rigor:

1. Substantial difference was observed between the expected and measured mass of ubiquitylated H1 (Fig. 1C). What might cause the 2-5 Da differences in ubiquitylated H1 variants? Were chemically modified ubiquitylated peptides (with +283 Da mass addition as described in Methods) detected to validate the

chemistry?

2. There is some discrepancy in the description of AP-MS. AP-MS experiments were done in three biological replicates (page 4, line 31) in Results section. However, "at least five out of six replicates" were required to be detected in Methods section (page 21, line 27).
3. For Fig. 2B, it is not clear how ANOVA statistics was done to determine the 270 proteins that were enriched. It is not clear what was used as the baseline to define enrichment, monomeric Ub, beads or unmodified H1.2? In addition, the number shown in Fig. 2C 165 and 37 doesn't add up to 270 proteins.
4. For AP-MS and XL-MS experiments in Methods, the mass analyzer used for MS/MS data acquisition needs to be specified, as well as the mass tolerance of parent and fragment ions for database search. False discovery rate for XL-MS also needs to be assessed.
5. Different crosslinking patterns were observed in H1-SIRT1 and H1-Ub-SIRT1 complexes. However, H1 is largely unstructured with only ~80 residues in the GD. Were H1 and SIRT1 crosslinks mainly in GD, NTD or CTD? If not mainly in GD, have the authors consider the difference in crosslinking pattern might be due to conformational changes in H1.
6. Fig. 3D and 3E don't seem consistent. In Fig. 3D, the H1/SIRT1 crosslinking sites seem to cluster in a similar region on SIRT1. But in Fig. 3E, H1 were placed were placed at drastically different position relative to SIRT1.
7. In Fig. 4B, less condensates can be observed in unmodified H1. Is there a control to show that the same concentration of unmodified H1 were used for this experiment?
8. The EMSA data shown in Fig. 5B is not conclusive. For instance, the addition of H1 to chromosome array made it migrate slower, probably due to the mass of H1 instead of the chromatin condensing feature of H1. Similarly, slower migration of ubiquitylated H1 might be largely due to the addition of ubiquitin, instead of the open, relaxed chromatin structure as suggested in the Results. The MNase assay in Fig. 5C also doesn't seem conclusive. The K206Ub H1 seemed to be more resistant towards MNase digestion from the gel image shown. How images were analyzed, and data points presented in 5C needs to be included in the Methods section.
9. Presentation of the materials needs to be clearer in Results section. For example, the incorporation of tags needs to be included in Results to provide context on how interactomes were characterized. The meaning if "-" in the annotations of Fig. 3F and Extended Fig. 3B was not clear and the Figure Legends were not helpful.

Reviewer #3 (Remarks to the Author):

Summary and significance

The manuscript describes the chemical synthesis and biochemical evaluation of a number of ubiquitylated histone H1 proteins. Incorporation of alkyne and azide functionalities, followed by CuAAC, generates non-hydrolyzable H1.2-Ub linkages at H1.2K17, K64, and K206. The modified proteins are then used as bait in AP-MS workflows to determine interacting proteins. Follow-up studies validate hits including DUBs and SIRT1, including a proposed binding model for the latter. Finally, the authors examine the phase-separation behavior of each of the modified proteins in in vitro assays, and probe the influence of H1.2Ub on chromosome assembly.

The study is thorough, well executed, and broadly compelling and represents an important addition to our understanding of H1 post-translational modification – an underexplored area within chromatin and epigenetics due to the challenges of obtaining pure material for biochemical evaluation. The study is of sufficient interest to the broad readership of Nature Communications and I would recommend publication of the manuscript if the following specific points are sufficiently addressed by the authors.

Specific points to address

Major

Fig. 4. The formation of liquid-like condensates between H1 and DNA is well established and, in the absence of any nucleosomal context, carries debatable biological significance. Recent work has shown that addition of H1 to chromatin arrays results in a greater propensity to phase separate (Gibson et al. Cell 2019, 179, 470). Studying the phase separation behavior of chromatin arrays in the presence of H1.2Ub would be more compelling given this previous work. How does H1.2Ub affect droplet formation in the presence of chromatin arrays?

In Fig. 5 the authors show that chromatin arrays formed in the presence of H1.2Ub are more rapidly digested by MNase or PacI. These data would also be contextualized by the results of the experiment above.

Minor

1. Abstract. Reword the first sentence. The final sentence: "promotes a transcriptionally active chromatin state" is fully not backed up by the forthcoming experiments, consider rewording.
2. Fig. 1 B&C. The authors present SDS-PAGE and mass spec analysis of the H1.2-Ub conjugates generated by CuAAC. The authors should provide a conversion estimate and isolated yield after cycloaddition. The panels in Fig. 1C should be expanded to allow the reader to identify contaminants in the purified proteins after conjugation.
3. Extended data figure 1D. The authors display LCMS-MS data confirming the desired triazole linkages. The H1.2K206Ub spectrum in particular is not good quality.
4. Fig. 3F & Extended data figure 5B. Include western blotting data for the experimental replicates.

Reviewer #4 (Remarks to the Author):

In this study by Höllmüller & Geigges et al., the authors have produced ubiquitylated linker histone (H1) to study its interactome, phase separation behavior and impact on chromosome assembly. The study contains biochemical data of high quality, but I am missing a validation in cells to judge the relevance of the presented findings.

Major points:

1. The authors generate and characterize site-specifically ubiquitylated H1.2 (K17Ub, K64Ub, K206Ub). At which abundance are such H1.2 versions present in "unperturbed" cells (that have not been treated with DNA-damaging agents)? How does this compare to other PTMs on H1? Are these PTMs co-occurring with other PTMs? These questions could be addressed by IP/WB/MS assays to make sure that the recombinant H1KxUbs mimic a significant cellular H1 pool.
2. For the interactome presented in Fig. 2, I am missing some statistical analysis to quantify "site-specificity". I understand that all candidates in the heatmap in Fig. 2B are significant interactors, but how significant are the differences between interactions with different H1KxUbs? The fold-enrichment between the individual H1KxUbs look pretty similar to me, in all of the six groups, which would mean that the site-specificity is moderate. Furthermore, are these interactions also seen in IPs of endogenous H1? This would be reassuring as it would imply that a significant H1 pool carries the respective KxUb modifications and that the interactions also are present in cells.
3. The model that H1.2 K64Ub binds SIRT1 and regulates its activity should be confirmed in cells, otherwise it is difficult to judge how relevant the observed "slight" decrease in SIRT1 activity is. For example, recombinant H1.2 K64Ub or a plasmid encoding a H1.2 K64 mutant that cannot be ubiquitylated could be brought into cells, and the acetylation level of SIRT1 targets could be monitored.

4. The authors show that both H1.2 and H1.2 KxUbs can form condensates, which look pretty similar to each other in the microscopy images, but that different numbers of condensates are observed for some of the variants. Could the different numbers be the result of condensates interacting more or less with the microplate? Does the number of condensates decrease if the solution is incubated longer in a tube before pipetting them into a microplate? As liquid condensates are supposed to fuse while they are allowed to come in contact with each other, I expect these parameters to affect the observed numbers, even if condensate formation itself occurred in a very similar way for the different proteins. Do the critical concentrations, at which condensates start to form, differ for the different proteins? This could be assessed by turbidity measurements to show potential differences more clearly.

Minor points

1. The authors claim that H1 PTMs are an understudied aspect of the histone code, which is supposedly due to a lack of appropriate tools. However, H1 behaves fundamentally different from core histones as it dynamically binds to chromosomes (residence time of a few minutes) while core histones stay at their place for hours. Therefore, PTMs on core histones can make up a stable "code", unlike PTMs on H1 that constantly exchanges between different loci. Thus, I find the analogy misleading, although both core histones and linker histones are called "histones" and one might call their PTMs in principle "histone code".
2. The authors claim in the Introduction that it has recently been shown that chromatin and heterochromatin undergo LLPS. Recently, several studies have questioned this view and have outlined the challenges in detecting LLPS in cells, which would be needed to settle this question. I think this point could be described in a more careful and balanced way.
3. I find the strong claims about the "function" of H1 problematic, as the authors conduct only in vitro assays and it is not clear to me how much these reflect the "function" of H1 in the cell. I think these claims should be toned down, unless they are supported by functional assays that go beyond an analysis of protein-protein interactions.

Point-by-point response to the reviewers' comments for "site-specific ubiquitylation is a general functional regulator of linker histone H1" (NCOMMS-20-36862-T)

REVIEWER COMMENTS

Reviewer #1 (Remarks to the Author):

In the present manuscript, Höllmüller and colleagues investigate possible functions of linker histone H1.2 ubiquitylation at three different sites. Using an elegant method previously developed in the same labs, the authors generate recombinant H1.2-ubiquitin analogs and compare them in a number of in vitro assays:

- interactors of three recombinant H1.2-ubiquitin analogs versus unmodified H1.2 in cell lysates were determined by proteomics
- One candidate interaction with histone deacetylase SIRT1 was characterized using enzymatic assays and crosslink-mass spec (XL-MS)
- liquid-liquid phase condensate formation assays with DNA were performed
- In vitro chromatosome assembly was performed and resulting chromosomes were analyzed for their stability and resistance to nuclease

Together, the results of the assays support a role for H1.2 ubiquitylation in the opening of chromatin and possibly promoting a more active state, thus counteracting to some extent the overall function of H1.2.

The experiments, data, and data representation are generally of high quality and the results are well presented. Text is well-written and appropriately discussed on the background of the existing literature. Thus, the manuscript is certainly well publishable with respect to all technical standards (only technical concerns see Fig 3F discussed below).

However as discussed below I would argue the state of the experiments is premature for publication in Nature Communications. My main concern is the lack of more definite experiments, particularly experiments performed in cells, that could support the conclusions drawn towards the function of H1.2 in a cellular system. I will discuss this in more detail below, but one important note: albeit statistically significant differences between ubiquitinated and unmodified H1.2 are seen in the functional in vitro experiments presented in Fig 3, 4, 5, the effect sizes are relatively small. I think that going from a small effect size in vitro to the hypothesis "that site-specific ubiquitylation of H1 has a general regulatory role and promotes a transcriptionally active chromatin state" is premature in the absence of additional experiments.

Note that the part "H1 has a general regulatory role" in the abstract is a relatively hollow statement, and at several instances in the manuscript similar non-descriptive wording is used such as "X affects Y", "X modulates Y" without specifying the detail or direction of the effect.

We thank the reviewer for the careful review of our manuscript and were happy to see that the reviewer considers our experiments and data of high quality and our manuscript well written. In order to hopefully dissipate the remaining concerns, we have carried out extensive

additional experiments – namely to identify and quantify the occurrence of endogenous H1.2 KxUb variants and to verify and validate interactors of endogenous H1.2 in cells (see also comments below and new Extended Data Fig. 1 and 3D and Extended Data Table 1). Last but not least, we have also significantly expanded our *in vitro* assays and have investigated the effect of site-specific ubiquitylation on phase separation also using intact 12-mer nucleosome arrays in order to further close the gap between a molecular *in vitro* and a cellular *in vivo* function for H1 (see new Fig. 5).

We also agree with the reviewer that our wording may appear overambitious at times, and we have tried to tone it down at various places throughout the manuscript: For a full set of revisions, please see the attached revised manuscript with track-changes.

For example, we have amended the manuscript in the Abstract (page 2):

“We show that site-specific ubiquitylation of H1 at position K64 modulates interactions with deubiquitylating enzymes and the ~~inhibits~~ deacetylase SIRT1. Moreover, it affects H1-dependent chromosome assembly and phase separation resulting in a more open chromosome conformation *generally associated with a transcriptionally active chromatin state*. In summary, we propose that site-specific ubiquitylation *plays a general regulatory role for linker histone H1*.”

And have changed the respective subheading (page 6) from:

“Ubiquitylation-dependent regulation of acetylation” to: “*Ubiquitylation-dependent interaction with SIRT1*”

Changed the Discussion (page 9) from “shows and regulates” to:

“Our data also *suggests* that ubiquitylation of H1.2 at position K64 *affects* the deacetylation capacities of SIRT1, thereby potentially counteracting its transcriptional repressive function.” and have also changed the title to:

“Site-specific ubiquitylation *acts as a general* regulator of linker histone H1”

Going through the presented data by Figure:

Fig 1: Synthesis of recombinant ubiquitylated H1.2 is elegant and, of course, exact due to the precise bioorthogonal chemistry used. The triazole linkage is considered a good mimetic of the Lys(GlyGly) isopeptide bond in the field, and thus I would argue that the experiments with these recombinant H1.2-ubiquitin analogs are robust and meaningful with respect to the natively ubiquitylated H1.2. In addition, some of the experiment would be potentially impossible to do with a native H1.2 due to concurrent deubiquitylation reactions in cellular lysate.

The choice of three sites is rationalized in the manuscript but it is of course always a question if the most functionally relevant sites can be known a priori. At least ten sites are known to exist, according to ref 49 (<https://doi.org/10.1074/mcp.M111.013284>) Supplementary Figure 3: K17, K34, K46, K64, K75, K85, K90, K97, K106, K206.

To my understanding, nothing is known about the relative abundance of mono-ubiquitylation at all the known sites, which would at least be one variable to take into account when choosing sites. A highly abundantly modified site may be considered more consequential *in vivo* than a site that is barely detectable by sensitive mass spec.

The reviewer addresses an important point. However, monitoring and quantifying the cellular H1-Ub pool is not trivial due to the lack of H1-Ub-specific antibodies, explaining the current lack of knowledge about relative abundances of mono-ubiquitylated forms of H1.2.

We have nevertheless carried out extensive additional experiments in order to assess both i) the occurrence of endogenous modification sites of H1.2 and ii) the relative abundance of H1.2-Ub variants in cells. We performed immunoprecipitation experiments to enrich endogenous H1.2 from HEK 293T cells and subjected the samples to LC-MS/MS analysis in order to identify endogenous H1.2 modification sites (see below new Extended Data Fig. 1 and please also refer to Extended Data Table 1 and new Methods section in the revised manuscript and to reviewer 4 for an extended answer).

We identified four ubiquitylation sites in the NTD and the GD of H1.2, including K17 and K64. However, probably due to the high lysine content (~ 40 %) of the CTD resulting in low sequence coverage, no ubiquitylation site was detected in the CTD of H1.2. Additionally, we detected multiple acetylation, phosphorylation and methylation sites in H1.2, confirming the relevance of the H1.2 KxUb variants investigated in this study and their co-occurrence with other relevant PTMs in the cell.

For a quantification of the relative abundance of ubiquitylated H1.2 *versus* non-ubiquitylated H1.2, we counted the numbers of peptide spectrum matches of peptides with the GG or LRGG signature motif and peptides without the respective motives. However, it is important to note that this is rather a rough estimate as quantification of different peptides *via* LC-MS/MS is limited by variations in ionization behavior for different peptides (such as ubiquitylated *versus* non-ubiquitylated peptides). Overall, ~ 5% of identified peptides were found to bear the GG or LRGG signature peptides, indicating that roughly 5% of the cellular H1.2 pool are likely to be ubiquitylated, most prominently at positions K17 and K64. Our additional data therefore clearly demonstrates that the H1.2 KxUb variants investigated represent a significant part of the cellular H1.2 pool.

Extended Data Fig. 1: Endogenous modification sites of H1.2.

Schematic depiction of modification sites in H1.2 after IP-enrichment from HEK 293T cells and identification by LC-MS/MS. Positions investigated within this study are indicated in bold (K17, K64 and K206). Probably due to the high lysine content (~ 40 %) of the CTD resulting in relatively low sequence coverage, no ubiquitylation site was detected in the CTD of H1.2.

Fig 2: Interaction partners of H1.2-ubiquitin species. This is a very interesting experiment and it comprehensively surveys interaction partners that are either sensitive or insensitive to ubiquitylation. However, I am missing here some crucial, more physiological, validation of the key interaction partners highlighted in the Figure and text. Pulldowns from lysates are accepted in the field and have in a myriad of cases lead to the identification of relevant functional interactions in the cell, but it is to expect that a subset of the hits are actually false-positives, since the stoichiometries and binding conditions are not really comparable to an endogenous setting. E.g. some interactions partners may simply pull down through favorable but unspecific hydrophobic or electrostatic interactions. Thus, validation is needed! While I fully agree with the authors that there is no equivalent clean experiment that can be done in cellulo, i.e. it seems not possible to pull out ubiquitylation-specific interaction partners on endogenous H1.2.

However, I could think of a number of simple experiments, each being better than no validation experiments at all:

- A co-IP from cell lysates using anti-H1.2 antibody
- a co-IP of transiently transfected, epitope-tagged H1.2
- a co-IP from cells stably expressing epitope-tagged H1.2
- any above performed after formaldehyde crosslinking to stabilize additional transient interactions

All of the above would clearly be expected to co-purify at least a subset of interactors found in the lysate pulldown.

There are some details to figure out here, e.g. what an appropriate control would be (another H1 variant? A core histone?). Interactions via chromatin need to be removed (e.g. by an adequate salt concentration to remove H1 from chromatin and then work with supernatant). But importantly, any interactors validated through this route will be much more believable since they would be proven to be present in cellulo.

Of course the Co-IP would occur from a mixed H1.2 species, thus it would not be possible to distinguish specificity of the interactors to unmodified H1.2, ubiquitylated H1.2, or H1.2 modified in any other way. However, expecting that the top interactors found here are also indeed relevant to H1.2's function in the cell, clearly those hits should show up. And assuming for a moment that ubiquitylation was a reasonably abundant modification on H1.2, the Co-IP should also pull down ubiquitylation-specific binding partners.

And it should even be possible to validate these ubiquitylation-specific interactors:

- a pulldown of transiently or stable transfected, epitope-tagged H1.2 with the specific K->A or K->R mutation(s) should remove those interactors that require the presence of site-specific ubiquitylation. K->A/R mutations also abrogate any other modification at the site (e.g. acetylation), but since the main and only important point of this experiments is to validate the role of ubiquitylation I believe this is not a problem.
- a pulldown of transiently or stable transfected, epitope-tagged H1.2 with all ubiquitylation sites mutated to A or R should further remove any of the ubiquitylation-specific interactors. I believe that a selection of experiments described above would be crucial validation and in particularly necessary for implying a functional role for the SIRT1 interaction.

We agree with the reviewer that further work was required to validate some of the key interaction partners under more physiological conditions. As suggested by the reviewer, we performed initial AP-MS experiments with HA-tagged H1.2 ectopically expressed in HEK 293T cells. In three biological replicates, we consistently detected 71 of the 202 significantly enriched interactors of H1.2 and the H1.2 KxUb variants, representing an overlap of 35%. These interactors include, for example, PPP6R3, SART3, NPM1 and UCHL5. However, since these experiments were performed with overexpressed H1.2, we feel that this approach has no advantages over our approach using recombinant H1.2 as both approaches are very similar to each other (e.g. the overexpression approach does not address the concern of the reviewer of "unspecific hydrophobic or electrostatic interactions"). Thus, we decided not to include this data in the revised manuscript.

As suggested by the reviewer, we have therefore validated some selected interactions from Fig. 2 also in cells for endogenous proteins – including SIRT1 – by co-immunoprecipitation using an anti-H1.2 antibody and western blot analysis.

Extended Data Fig. 3:

(D) Co-immunoprecipitation analysis of interactors of endogenous H1.2 visualized by western blot analysis. Lysate indicates HEK 293T cell lysate as input, IgG indicates the control with an unspecific IgG antibody and H1.2 experiments with the anti-H1.2 antibody. SN marks supernatant.

We added these results in Extended Data Fig. 3D and amended the manuscript accordingly in Results (page 5) and Methods section (page 23):

“Finally, selected interactors co-immunoprecipitated with endogenous H1.2 (Extended Data Fig. 3D) further confirming our AP-MS results.”

We also agree with the reviewer that many more experiments can be done to study the potential interactions identified. However, the main purpose of our study is to provide evidence that site-specific ubiquitylation affects the properties/functions of H1.2. We therefore focused our efforts on two main aspects, identification of the interactomes of the various H1.2 forms and the effect on chromosome structure/compaction. Our data clearly show that both aspects are affected by site-specific ubiquitylation of H1.2. We feel that this data are highly original and of immediate interest to many in the field. While we agree that not all of the

interactions may be of physiological relevance (though the relevance may be difficult to disprove, as some of the interactions may only take place under distinct circumstances, e.g. during certain phases of the cell cycle or in response to certain stress stimuli), we would like to emphasize that our AP-MS data meet the highest scientific standards and are by themselves already highly reproducible: e.g. interactors had to be detected consistently in three independent biological replicates (the standard in the field is two biological replicates for such large-scale proteomics data) and multiple interactors have been already validated by immunoprecipitation and western blotting (see Fig. 2D).

Thus, we hope that the reviewer agrees that while further studies would be justified to prove or disprove some of the interactions, such studies are clearly beyond the scope of this manuscript.

Fig 3: Modulation of SIRT1 activity by H1.2 K64ub. Following up on SIRT1 found in the MS in Fig 2, the authors continue to characterize this putative interaction *in vitro*. As said above, my instinct would have been to first validate that SIRT1 can be a H1.2 interactor *in vivo/in cellulo*, and this could have again been easily done by

- co-IP of endogenous H1.2 or SIRT1 and western for the respective other
- Transient co-transfection of epitope-tagged SIRT1 and H1.2 (including H1.2 K56A etc. control(s)) and reciprocal pulldown.

Only if at least some hint for an interaction to occur in the cell, then Fig 3 makes sense.

However I do have some minor issues:

We agree with the reviewer and hope that the data shown above convince the reviewer that SIRT1 is indeed a *bona fide* interactor of H1.2 *in cellulo*. In addition, in ref. 56 of the manuscript, H1.2 was reported to interact with SIRT1 in cells.

– Fig 3B: if it is relevant to know if the same DUBs are also acting on H1.2 *in vivo*, then epitope-tagged H1.2 could be transiently transfected into cells +/- siRNA against the DUB, then western blot for the epitope. Maybe in the control condition there would already be additional bands appearing for the epitope-tagged H1.2, corresponding to ubiquitylated species running at higher MW. And upon knockdown of a relevant DUB, those bands should become more intense or appear.

We agree with the reviewer that it would be interesting to know if the DUBs we have identified as interactors of H1.2 and shown to deubiquitylate H1.2 *in vitro* would also act on H1.2 *in vivo*. Unfortunately, as only up to 5% of H1.2 is endogenously ubiquitylated (see above), western blot analysis is not sensitive enough to detect such subtle differences. This is particularly the case if additional DUBs act on H1.2-Ub, which is likely to be the case. Moreover, as the focus of our paper is the effect of *site-specific* ubiquitylation on H1.2 function, we feel that although the suggested experiments are interesting in their own right, they are beyond the scope of our manuscript.

– Fig 3D: XL-MS. I am not an expert in this, so I assume that crosslinking between proteins that do not interact also doesn't occur at a significant level. Or to put in a question: is peptide-level evidence XL-MS considered enough in the field to imply a direct biochemical interaction? Could the complex shown to be stable on a gel filtration column? By pulldowns using the recombinant interaction partners?

The reviewer is correct in the assumption that crosslinking between proteins that do not interact does not occur at a significant level, if at all; proof of this can also be found in Extended Data Fig. 6A (formerly Extended Data Fig. 5A) where free Ub was added as a control and no crosslinks to Ub were detected. We and others could also show that XL-MS can not only confirm that proteins interact but even define their exact interaction site (PMID: 29162807); therefore, such a strong crosslinking pattern as detected in this case between SIRT1 and H1.2, where a large number of high-confidence links was detected reliably over biological triplicate measurements (see also new Supplementary Table 2 for details), is a reliable proof that these two proteins indeed interact *in vitro*.

Furthermore, the additionally suggested pulldown experiments have also already been carried out: for recombinant protein, see Fig. 2D and for endogenous H1.2, see new Extended Data Fig. 3D and above.

– Fig 3E is nice and I believe the XL-MS technology is elaborate enough to allow fairly precise models, but would it be possible to take this information, make a set of mutations, domain deletion, and check if the interaction is indeed disrupted by manipulating the predicted interface?

We thank the reviewer for the appreciation of our Figure and agree that it is indeed possible to generate fairly precise models under favorable circumstances (see, for example, our proof-of-concept paper introducing Bayesian crosslinking-based integrated modeling: PMID: 25171412).

Generating a set of mutations and domain deletions to further probe the interaction of SIRT1 with H1.2 is feasible but requires significant amounts of additional work, particularly as H1.2 is a largely unstructured protein and shows no clear interaction site by XL-MS. Therefore and given the fact that knowing the exact interaction site would not add much to what we know already, we believe that the suggested experiments are clearly beyond the scope of this manuscript and better suited for a follow-up study that will have a closer look at the interplay between SIRT1 and H1.2.

and one major issue:– Fig 3F is the only panel where I can't attest the otherwise high data quality. The quantification in Ext Data Fig 5 shows just one significant difference at the 30 min time point, so any effect of H1.2 or H1.2ub is marginal in the shown experiment. But why are the signals anti-p53 K370AcK and anti-p53 so much higher in the middle assay "+ H1.2 K64Ub"? Shouldn't be the first two lanes of each assay, "– NAD+" and "– SIRT1" be exactly the same? Otherwise how can the assays be quantitatively compared, if the starting amounts aren't even the same? Especially since HRP-western has been used, the signal is known not

to be linear, which means that essentially one cannot and should not make a quantitative comparison across assays with vastly different starting intensities. Since these blots are the entire basis for the conclusion that “H1.2 ubiquitylation affects SIRT1 activity”, I believe this conclusions needs to be corroborated.

For unknown reasons, the p53 signal in the presence of H1.2 K64Ub is higher in the Western blot analysis than in its absence, although the actual input level of p53 was identical at all conditions (as a common master mix containing p53 was employed). Since this is a highly reproducible phenomenon and we were not able to figure out the reason for it, we performed an additional experiment with p53 acetylated at position K382. As shown below, similar results were obtained with this p53 variant.

In vitro deacetylation assay for SIRT1 and model substrate protein p53 K382AcK in the presence of H1.2 (+H1 WT), H1.2 K64Ub (+H1 K64Ub) or in the absence of any histone (-H1). – NAD⁺ and – SIRT1 indicate that the respective component was not present in the reaction mixture. Protein and acetylation intensities were visualized by western blot.

For quantification, signal intensities were normalized to the starting point of each set of conditions to account for potential variations in intensities. Furthermore, different exposure times were taken and the signals quantified (an example for p53 K370AcK is shown below) to assure that we are in the linear range of respective signals:

Deacetylation assay for SIRT1 and model substrate protein p53 K370AcK in the presence of H1.2 (+H1 WT), H1.2 K64Ub (+H1 K64Ub) or in the absence of any histone (-H1) as shown in Fig. 3F. Exposure of the anti-p53 K370AcK blot (top) is shown for two different exposure times (labeled as (short) and unlabeled).

Nonetheless and as indicated in the original manuscript ("the deacetylation activity of SIRT1 is slightly decreased after incubation with H1.2 K64Ub relative to H1.2"), we agree that the effects on SIRT1 activity – even though clearly reproducible – are mild. However, the main message of the results shown in Fig. 3, that site-specific ubiquitylation of H1.2 at K64 alters its interaction with SIRT1, is clearly shown by XL-MS and modeling data.

To make this more clear, we have further toned down the manuscript and:

Have changed the respective subheading (page 6) from:

“Ubiquitylation-dependent regulation of acetylation” to “*Ubiquitylation-dependent interaction with SIRT1*”

Removed “inhibit” from the Abstract (page 2) to write:

“We show that site-specific ubiquitylation of H1 at position K64 modulates interactions with deubiquitylating enzymes and the ~~inhibits~~ deacetylase SIRT1.”

And the Results on page 6:

“Having established that H1.2 K64Ub forms a distinct complex with SIRT1, we speculated that the enzymatic activity of SIRT1 is *modulated* by H1.2 in a ubiquitylation-dependent manner.”

And changed the Discussion (page 9) from “shows and regulates” to:

Our data also *suggests* that ubiquitylation of H1.2 at position K64 *affects* the deacetylation capacities of SIRT1, thereby potentially counteracting its transcriptional repressive function.

Fig 4: Phase condensates: I am not an expert in this, certainly phase condensates are ‘in’ and clearly are relevant for regulation of chromatin. The assays seems performed to the standards of the field. I understand there is no assay that could test any of this behaviour in cells. Maybe transiently overexpression of epitope-tagged H1.2 and K->A mutants thereof followed by IF could say something about how H1.2 partitions in the nucleus (if it is not just relatively diffuse, in which case not much would be to learn...)

Unfortunately, there is no *in vivo* assay for (PTM-specific) phase separation. We have performed immunofluorescence staining of endogenous H1.2. However, this method only shows the localization of H1.2 and not if H1.2 is localized within a condensate.

However, we have also carried out extensive additional experiments and have repeated all phase separation experiments with H1.2 and each of the three different H1.2 KxUb variants using 12-mer nucleosome arrays (please see our response to Reviewer 3, new Fig. 5 and Extended Data Fig. 9). We find that all of the effects caused by site-specific ubiquitylation of H1.2 on H1.2-/ H1.2 KxUb-DNA condensates are also seen in condensates of H1.2-/ H1.2 KxUb-nucleosome arrays (see new Fig. 5). All condensates exhibit a similar partition coefficient but for ubiquitylated H1.2 larger condensates (especially with the 12-mer arrays) and most notably more condensates were formed compared to unmodified H1.2 (which again was particularly the case for H1.2 K64Ub).

While the latter set of experiments is arguably still not within cells, we hope that the reviewer agrees that these experiments further close the gap between a molecular *in vitro* and a cellular *in vivo* function for H1-Ub.

Fig 5: Chromosomes: Data convincingly shows that H1.2 makes arrays more inaccessible and ubiquitylation can reverse some of this effects, with some site-specificity. Could the authors do a complementary experiment where chromosomes are formed with unmodified H1.2 and subsequently ubiquitylated enzymatically and subsequently subjected to the MNase assay? Or if the reaction conditions were incompatible with the chromatosome, then simply, enzymatically ubiquitylated H1.2 could be used side-by-side with the recombinant site-specific analogs. This would simply provide some additional evidence that the effect is also true for the 'real' ubiquitylation linkage and in general robust.

To the best of our knowledge, a site-specific E3 ubiquitin ligase for H1 is not known. In fact, ubiquitylation of H1.2, e.g. by the E3 ligase HUWE1, leads to H1.2 forms that are modified at several lysine residues by single ubiquitin moieties (multiple mono-ubiquitylation) and even by ubiquitin chains (poly-ubiquitylation) (see Extended Data Fig. 4). Any site-specific effects would therefore be lost or disguised when using enzymatic ubiquitylation, and we would only be able to investigate the summed effect of inhomogeneous H1-Ub mixtures on chromatosome structure – in contrast to the specific effects caused by our H1.2 KxUb variants. Furthermore, if enzymatic ubiquitylation (by HUWE1) would be carried out on chromatosome arrays, not only H1.2 but also the core histones would probably be ubiquitylated (PMID: 15767685). Ubiquitylation of core histones would additionally affect chromatosome assembly and structure, which could not be separated from the effects of ubiquitylated H1.2.

Importantly, the correct fold (secondary structure) was confirmed for each of the H1.2 KxUb variants by CD spectroscopy and their acceptance as substrates has been shown in an *in vitro* ubiquitylation assay with HUWE1 (see Extended Data Fig. 2B and E), confirming their structural integrity. Previously, we have also demonstrated that the linkage introduced by chemical synthesis mimics the native ubiquitylation linkage well, exhibits a similar high-resolution structure and generally behaves like the natural linkage (PMID: 30920720; PMID: 32301549; PMID: 29045006).

Reviewer #2 (Remarks to the Author):

In this manuscript, the authors reported the chemical synthesis of site-specific, non-hydrolysable mono-ubiquitylated H1.2 variants and their modification-dependent interactome. The functional relevance of some representative interactions was further examined on three deubiquitylation enzymes and deacetylase SIRT1. Additional biochemical assays on condensate formation and chromatosome assembly suggested a direct role of H1 ubiquitylation on the regulation of chromatin structure beyond modulating protein-protein interactions. Overall, the functional characterization on linker histone H1 ubiquitylation seems thorough with multiple approaches and the scope suitable for Nature Communications. However, some experimental details need to be clarified and discussions included to ensure scientific rigor:

1. Substantial difference was observed between the expected and measured mass of ubiquitylated H1 (Fig. 1C). What might cause the 2-5 Da differences in ubiquitylated H1 variants? Were chemically modified ubiquitylated peptides (with +283 Da mass addition as described in Methods) detected to validate the chemistry?

We thank the reviewer for the careful review of our data. We do not exactly know the cause of the slight mass differences for some of the ubiquitylated H1.2 KxUb variants but suspect that they may be caused by residual buffer ions, as is commonly the case in MS measurements of intact proteins. More importantly, our higher-resolved fragmentation MS2 spectra unambiguously prove that the correct H1.2 KxUb variants with the +283 Da mass addition were generated as shown in Extended Data Fig. 2D.

2. There is some discrepancy in the description of AP-MS. AP-MS experiments were done in three biological replicates (page 4, line 31) in Results section. However, “at least five out of six replicates” were required to be detected in Methods section (page 21, line 27).

We apologize for not being clear enough on this issue. All AP-MS experiments were indeed carried out in three independent biological replicates. Additionally, each biological replicate was measured twice as technical duplicates (i.e. three biological replicates, each measured as technical duplicates = six replicates in total). In our analysis of the data, we required that each protein had to be identified in at least five out of these six replicates, such ensuring that it was identified in all biological replicates.

In order to clarify this point, we have amended the manuscript (page 22) to:

“at least five out of six replicates (*three biological replicates, each measured as technical duplicates*)”

3. For Fig. 2B, it is not clear how ANOVA statistics was done to determine the 270 proteins that were enriched. It is not clear what was used as the baseline to define enrichment, monomeric Ub, beads or unmodified H1.2? In addition, the number shown in Fig. 2C 165 and 37 doesn't add up to 270 proteins.

ANOVA statistics as implemented in the Perseus 1.6.1.3 software was used and, as is usually the case for ANOVA statistics, different baits/conditions were not compared pairwise but all different baits/conditions were compared relative to each other in order to determine significantly enriched proteins. As described in the Methods section, parameters FDR and S0 were set to FDR = 0.001 and S0 = 2, resulting in a total of 270 proteins with significant changes between all the different baits used for our AP-MS experiments: unmodified H1.2, H1.2 K17Ub, H1.2 K64Ub, H1.2 K206Ub and including monomeric Ub.

The numbers in Fig. 2C (165 + 37 = 202) refer only to the interactors of H1.2 and the various H1.2 KxUbs variants. The discrepancy in numbers stems from the free Ub-specific interactors.

To clarify this point, we amended the manuscript (Fig. 2, page 12)

“(C) Venn diagrams (bottom) of all proteins that specifically interact with unmodified H1.2 or H1.2 KxUbs, respectively (left) and all *H1.2 KxUb-specific* interacting proteins with site-specific resolution (right), *not including interactors of free Ub-specific interactors (for those see cluster 6)*. Further GO-term analysis of all *ubiquitylation-specific* H1.2 KxUb-interacting proteins based on PANTHER classification (top).”

4. For AP-MS and XL-MS experiments in Methods, the mass analyzer used for MS/MS data acquisition needs to be specified, as well as the mass tolerance of parent and fragment ions for database search. False discovery rate for XL-MS also needs to be assessed.

MS/MS data for both AP-MS and XL-MS experiments were acquired in a linear ion trap. For analysis of AP-MS data, the mass tolerance of parent ions was set to 4.5 ppm and 0.5 Da for fragment ions, respectively. We have amended the Methods section of the manuscript accordingly to include this information (page 21 and page 24).

As written in the Methods section (see page 24 of the manuscript), the mass tolerance of parent and fragment ions for XL-MS was 10 ppm for parent and 0.2 Da for common fragment ions and 0.3 Da for crosslink fragment ions, respectively. The false discovery rate was calculated by xProphet and set to 5%:

“Data were searched using xQuest 2.1.3 in ion-tag mode with a precursor mass tolerance of 10 ppm. For matching of fragment ions, tolerances of 0.2 Da for common ions and 0.3 Da for crosslink ions were applied. (...) crosslinks were filtered for Id score > 30, deltaS < 0.95, FDR < 0.05 and only unique crosslinking sites identified in all three biological replicates are shown.”

5. Different crosslinking patterns were observed in H1-SIRT1 and H1-Ub-SIRT1 complexes. However, H1 is largely unstructured with only ~80 residues in the GD. Were H1 and SIRT1

crosslinks mainly in GD, NTD or CTD? If not mainly in GD, have the authors consider the difference in crosslinking pattern might be due to conformational changes in H1.

Fig. 3D shows that SIRT1 crosslinks mainly *via* a region close to its C-terminal regulatory domain to H1.2. For H1.2, on the contrary, we find crosslinks to SIRT1 in all of its domains (GD, NTD and CTD) and crosslinks to Ub mainly in its NTD and GD. Given the fact that H1 is largely unstructured and contains a large number of lysine residues, this was not an entirely unexpected result.

We certainly did consider that any difference in crosslinking pattern might be caused by conformational changes in H1.2. This is why we additionally performed Bayesian crosslinking guided integrative structural modeling using our crosslinking data as input. However, as H1 is largely unstructured, as the reviewer correctly points out, the long unordered N- and C-terminal tails could not be placed unambiguously, preventing us from making any statement on the exact conformational changes taking place within H1.2 upon complex formation with SIRT1.

Using the crystal structures of SIRT1, Ub and the H1 GD together with our crosslinking data as input, we were able to generate unbiased, highly reproducible and robust models for both the H1.2: SIRT1 and H1.2 K64Ub: SIRT1 complex (Fig. 3E, Extended Data Fig. 7A and B). To account for the long unstructured domains of H1.2 and SIRT1, we further used a modeling strategy where these regions were partially represented by flexible beads as described in the Methods section (page 24). While our models show more than one possible localization for Ub within the H1.2 K64Ub: SIRT1 complex, which indeed suggests multiple conformational states, they also unambiguously demonstrate that ubiquitylation of histone H1.2 significantly impacts the positions of H1.2 and SIRT1 relative to each other, suggesting conformational changes within the H1.2: SIRT1 complex upon ubiquitylation.

To facilitate assessing the various crosslinking patterns even further, we have compiled an additional Supplementary Table 2 containing all crosslinks from our XL-MS experiments.

6. Fig. 3D and 3E don't seem consistent. In Fig. 3D, the H1/SIRT1 crosslinking sites seem to cluster in a similar region on SIRT1. But in Fig. 3E, H1 were placed were placed at drastically different position relative to SIRT1.

Fig. 3D and E only appear to be inconsistent at first sight. As described in the manuscript and in the answer to the previous question, both Figures are based on the exact same crosslinks from our XL-MS experiments. These crosslinks together with the crystal structures of SIRT1, Ub and the H1 GD together as input generated unbiased, highly reproducible and robust models (Fig. 3E, Extended Data Fig. 7A and B; see Methods for more details), which clearly confirmed that ubiquitylation of histone H1 significantly impacts the positions of H1.2 and SIRT1 relative to each other.

The explanation for this seemingly inconsistent behaviour is the following: it is true that for both experiments the crosslinks between H1.2 and SIRT1 are found in roughly the same region. However, in the second experiment, we also have to consider the crosslinks going from Ub to H1.2 and SIRT1, most of which are interacting with the same region of SIRT1 as

the crosslinks from H1.2 (MS is an ensemble technology that will generate data from multiple complexes present in solution). Our modeling framework accommodates for all these crosslinks while respecting the 3D space occupied by the proteins. Therefore H1.2 and Ub compete for the same region of 3D space next to SIRT1, leading to the orientation shift of H1.2 when compared to the experiment without Ub.

7. In Fig. 4B, less condensates can be observed in unmodified H1. Is there a control to show that the same concentration of unmodified H1 were used for this experiment?

The concentrations of the four different H1.2 stock solutions were quantified by SDS-PAGE prior to their application in the phase separation assays (this information is added as Extended Data Fig. 9A).

8. The EMSA data shown in Fig. 5B is not conclusive. For instance, the addition of H1 to chromosome array made it migrate slower, probably due to the mass of H1 instead of the chromatin condensing feature of H1. Similarly, slower migration of ubiquitylated H1 might be largely due to the addition of ubiquitin, instead of the open, relaxed chromatin structure as suggested in the Results. The MNase assay in Fig. 5C also doesn't seem conclusive. The K206Ub H1 seemed to be more resistant towards MNase digestion from the gel image shown. How images were analyzed, and data points presented in 5C needs to be included in the Methods section.

The EMSA assay is used for quality control. It confirms the chromosome arrays' correct formation and shows that all H1.2 KxUb variants were successfully incorporated into the chromosome array (Fig. 4B). The reviewer is correct in pointing out that the migration behavior in the EMSA is influenced by both mass and structure. The slower EMSA migration of chromosome arrays (which include H1) compared to nucleosomes was previously shown by others (PMID: 17962805). However, as all H1.2 KxUb variants have the same mass, the structure and composition of the ubiquitylated chromosome arrays is likely to influence the migration behavior as well explaining their slight variations, as shown in Fig. 4B.

MNase digestion experiments were performed in triplicates and one of the corresponding agarose gels was shown in Fig. 4C (left) exemplarily. The band intensities of the undigested DNA array in the agarose gel images were quantified densitometrically and the mean of the intensities together with their standard deviation was plotted over time, showing a clear and statistically significant difference between both nucleosomes and chromosomes and between H1.2 KxUb chromosome variants (Fig. 4C, middle and right panel). As all intensity values were normalized to the zero time point of the respective sample, a comparison between the different samples at a particular time point may be difficult. Still, even when only one of the exemplary gel images is taken into account, the single images faithfully mirror the overall plotted mean intensities, e.g. for H1.2 K206Ub, the band intensity at 20 min is stronger than the other bands at this time point.

We therefore respectfully disagree with the reviewer and are convinced that the main conclusion concerning compaction of the array and DNA accessibility can be drawn from the MNase digestion assay in Fig. 4C.

In order to meet the reviewer's concerns, we have amended the respective paragraph in our manuscript (page 7):

"However, ubiquitylation of H1.2 resulted in a slightly retarded migration behavior using an electrophoretic mobility shift assay (EMSA) (Fig. 4B), which may result from the additional protein mass of Ub or from a more open or relaxed chromatosome structure or both."

We also agree with the reviewer regarding image analysis and have extended the Methods section as suggested (page 27):

"After agarose gel electrophoresis and ethidium bromide staining, undigested 12-mer array DNA was quantified densitometrically by ImageQuant TL. The software's 1D gel analysis mode was used for background subtraction and to determine the intensities of all undigested array bands. Intensity values were normalized to the 0 min-time point and plotted over time. "

9. Presentation of the materials needs to be clearer in Results section. For example, the incorporation of tags needs to be included in Results to provide context on how interactomes were characterized. The meaning if "-" in the annotations of Fig. 3F and Extended Fig. 3B was not clear and the Figure Legends were not helpful.

We apologize for not being clear enough.

As suggested, we added a description of how the tags were employed to characterize interactomes to the Results section (page 4).

"In addition, H1.2 was equipped with an N-terminal Strep-tag II for affinity purification."

"Building on previous efforts^{52,53}, we adapted an affinity purification-mass spectrometry (AP-MS) based approach to identify Ub-dependent interaction partners of H1 using its N-terminal Strep-tag II for enrichment (Fig. 2A, Extended Data Fig. 2A and Supplementary Table 1)."

The "-" in Fig. 3F and Extended Data Fig. 4B was supposed to indicate "without", e.g. that the respective component was not present in the reaction mixture.

In order to clarify this point, we have amended the manuscript accordingly (page 14) to

"- NAD⁺ and - SIRT1 indicate that the respective component was not present in the reaction mixture."

and (page 33) to:

"The "-" indicates that the respective component was not present in the reaction mixture."

Reviewer #3 (Remarks to the Author):

Summary and significance

The manuscript describes the chemical synthesis and biochemical evaluation of a number of ubiquitylated histone H1 proteins. Incorporation of alkyne and azide functionalities, followed by CuAAC, generates non-hydrolyzable H1.2-Ub linkages at H1.2K17, K64, and K206. The modified proteins are then used as bait in AP-MS workflows to determine interacting proteins. Follow-up studies validate hits including DUBs and SIRT1, including a proposed binding model for the latter. Finally, the authors examine the phase-separation behavior of each of the modified proteins in *in vitro* assays, and probe the influence of H1.2Ub on chromosome assembly.

The study is thorough, well executed, and broadly compelling and represents an important addition to our understanding of H1 posttranslational modification – an underexplored area within chromatin and epigenetics due to the challenges of obtaining pure material for biochemical evaluation. The study is of sufficient interest to the broad readership of Nature Communications and I would recommend publication of the manuscript if the following specific points are sufficiently addressed by the authors.

Specific points to address

Major

Fig. 4. The formation of liquid-like condensates between H1 and DNA is well established and, in the absence of any nucleosomal context, carries debatable biological significance. Recent work has shown that addition of H1 to chromatin arrays results in a greater propensity to phase separate (Gibson et al. Cell 2019, 179, 470). Studying the phase separation behavior of chromatin arrays in the presence of H1.2Ub would be more compelling given this previous work. How does H1.2Ub affect droplet formation in the presence of chromatin arrays?

We are grateful to the reviewer for this excellent suggestion.

We characterized the phase separation behavior of the 12-mer nucleosome array with H1.2 and each of the three different H1.2 KxUb variants. We find that all of the effects caused by site-specific ubiquitylation of H1.2 on H1.2-/ H1.2 KxUb-DNA condensates are also seen in condensates of H1.2-/ H1.2 KxUb-nucleosome arrays (see new Fig. 5 and Extended Data Fig. 9). All condensates exhibit a similar partition coefficient but for ubiquitylated H1.2 larger condensates (especially with the 12-mer arrays) and most notably more condensates were formed compared to unmodified H1.2 (which again was particularly the case for H1.2 K64Ub).

As we agree with the reviewer that these results make our findings much more compelling, we have drafted an extended Fig. 5 and rearranged the manuscript to incorporate our findings and to show the effect of site-specific ubiquitylation of H1.2 on droplet formation side-by-side for DNA and chromatin arrays.

Fig. 5: H1.2 KxUb-dependent condensate formation.

(A) Overview of condensate formation assay using fluorescently labeled H1.2 and H1.2 KxUBs where either the nucleosome positioning DNA or intact 12-mer nucleosome arrays were added. The different linker histone variants were chemically labeled with Alexa-Fluor 488-NHS, DNA was amplified with a 5'-Atto-390-modified primer while the DNA array was not labeled. **(B)** Representative microscopic images of various H1.2- and H1.2 KxUB-DNA condensates. Shown are images after excitation with a 488 nm laser (green) or a 405 nm laser (blue) and the merged images (light blue). Scale bar 10 μm . **(C)** Representative microscopic images of H1.2- and H1.2 KxUB-array condensates. Shown are images after excitation with the 488 nm laser (green). Scale bar 10 μm . **(D)** Characterization of H1.2- and H1.2 KxUB-DNA (top) and H1.2- and H1.2 KxUB-nucleosome array (bottom) condensates. Shown are the partition coefficient (left), the number of droplets (middle) and mean droplet size (right) 20 min after mixing. Data were extracted from the images in the histone (green) channel. Error bars represent standard deviations, t-test, **** $p \leq 0.0001$, *** $p \leq 0.001$, ** $p \leq 0.01$, * $p \leq 0.05$, ns $p > 0.05$.

The text in the manuscript, the Figure legend, the corresponding Extended Data Fig. 9 and the Methods part of the manuscript have been changed accordingly (see the revised manuscript with track changes on).

In Fig. 5 the authors show that chromatin arrays formed in the presence of H1.2Ub are more rapidly digested by MNase or Pacl. These data would also be contextualized by the results of the experiment above.

We fully agree with the reviewer and have decided to switch the place of the previous Fig. 4 (H1.2 KxUb-dependent condensate formation) and Fig. 5 (Impact of H1.2 KxUb variants on chromosome assembly) in order to better contextualize the results from the various chromosome arrays for phase separation.

See in particular the last paragraph on page 8 in the Results section:

“To determine a potential impact of site-specific ubiquitylation, we closely monitored the morphology and dynamics of H1.2 KxUb condensates and analyzed the partition coefficient, number and size of formed droplets (Fig. 5D, top) and their liquid-like dynamics as assessed by FRAP (Extended Data Fig. 9F and G). We found no significant effect of ubiquitylation on the partition coefficients of the resulting condensates (Fig. 5D, left) and detected only a marginal influence on their size (Fig. 5D, right) and dynamics (Extended Data Fig. 9F and G). However, we observed that H1.2 and the H1.2 KxUb variants differ in the total number of formed condensates (Fig. 5D, middle). We additionally assessed phase separation behavior of labeled H1.2 and H1.2 KxUb variants with the 12-mer nucleosome arrays (Fig. 5A and C) and found that their behavior regarding partition coefficient, number and size mirrors our observations with the H1.2/H1.2 KxUb-DNA condensates (Fig. 5D, bottom). While the effect on mean droplet size is even more pronounced for intact chromosomes, histones ubiquitylated in their NTD (H1.2 K17Ub) or GD (H1.2 K64Ub) formed significantly more droplets than non-ubiquitylated H1 in both settings (Fig. 5D). These results suggest that ubiquitylation of the NTD or GD of H1.2 leads to the formation of more but less concentrated condensates as partition coefficients were unchanged while the number and size of formed condensates increased.

Taken together, we therefore find that site-specific ubiquitylation affects and modulates H1.2-mediated condensate formation in the presence of DNA alone or of intact 12-mer nucleosome.”

And the Discussion (page 10):

“Our results showed that ubiquitylation of H1.2 at position K64 leads to the formation of more but less concentrated H1-dependent condensates. Remarkably, these site-specific ubiquitylation effects for condensates consisting of H1 and its nucleosome positioning DNA are fully mirrored by condensates consisting of intact chromosomes. Reducing compactness by obtaining a more open conformation would explain the observed reduction in the concentration of H1-dependent condensates.”

Minor

1. Abstract. Reword the first sentence. The final sentence: “promotes a transcriptionally active chromatin state” is fully not backed up by the forthcoming experiments, consider rewording.

We agree with the reviewer and adjusted the wording accordingly (page 2):

“Moreover, it affects H1-dependent chromosome assembly and phase separation resulting in a more open chromosome conformation *generally associated with a transcriptionally active chromatin state*. In summary, we propose that site-specific ubiquitylation *plays a general regulatory role for linker histone H1*.”

2. Fig. 1 B&C. The authors present SDS-PAGE and mass spec analysis of the H1.2-Ub conjugates generated by CuAAC. The authors should provide a conversion estimate and isolated yield after cycloaddition. The panels in Fig. 1C should be expanded to allow the reader to identify contaminants in the purified proteins after conjugation.

We have addressed this issue of the reviewer by adding the following requested information (page 18-19):

“The H1.2 KxPlk variants were obtained in 0.6 mg (H1.2 K17Plk), 0.3 mg (H1.2 K64Plk) and 1.3 mg (H1.2 K206Plk) yield per liter of expression culture.”

“CuAAC conversion rates were optimized for each batch of protein to obtain > 90-95% of conversion.”

“Isolated yields after conjugation varied with around 25% (H1.2 K17Ub), 5% (H1.2 K64Ub) and 30% (H1.2 K206Ub) of used H1.2 KxPlk, respectively.”

Additionally, we have further expanded the LC-MS spectra shown in Fig. 1C and Extended Data Fig. 2C, as suggested by the reviewer.

Fig. 1C

Extended Data Fig. 2C

3. Extended data figure 1D. The authors display LCMS-MS data confirming the desired triazole linkages. The H1.2K206Ub spectrum in particular is not good quality.

We thank the reviewer for the careful review of our data and agree with the reviewer that perfect experimental data is hard to come by.

H1.2 and especially its CTD consist of a high number of lysine residues (40 %), which complicate the typical tryptic digest and make it difficult to cover the CTD by specific proteases. That is why pepsin was chosen in combination with a protease-unspecific data analysis leading to select MS/MS spectra with partially sub-optimal fragmentation behavior. However, it is important to note that the spectrum clearly and unambiguously shows site-specific ubiquitylation of H1.2 K206PIk *via* CuAAC.

4. Fig. 3F & Extended data figure 5B. Include western blotting data for the experimental replicates.

To address the request of the reviewer, here we show the three replicates of the deacetylation assay:

Replicate 1:

Replicate 2:

Replicate 3:

Replicates 1-3 of the *in vitro* deacetylation assay for SIRT1 and model substrate protein p53 K370AcK in the presence of H1.2 (+H1 WT), H1.2 K64Ub (+H1 K64Ub) or in the absence of any histone (-H1) as shown in Fig. 3F and Extended Data Fig. 6B. Replicate 1 refers to the experiment exemplarily shown in Fig. 3F.

Reviewer #4 (Remarks to the Author):

In this study by Höllmüller & Geigges et al., the authors have produced ubiquitylated linker histone (H1) to study its interactome, phase separation behavior and impact on chromosome assembly. The study contains biochemical data of high quality, but I am missing a validation in cells to judge the relevance of the presented findings.

Major points:1.

The authors generate and characterize site-specifically ubiquitylated H1.2 (K17Ub, K64Ub, K206Ub). At which abundance are such H1.2 versions present in “unperturbed” cells (that have not been treated with DNA-damaging agents)? How does this compare to other PTMs on H1? Are these PTMs co-occurring with other PTMs? These questions could be addressed by IP/WB/MS assays to make sure that the recombinant H1KxUbs mimic a significant cellular H1 pool.

The reviewer addresses an important point. However, monitoring the cellular H1.2-Ub pool is not trivial due to the lack of H1.2-Ub-specific antibodies. Although a quantitative comparison of cellular levels of ubiquitylated H1.2 to other PTMs and assessing the general PTM state of H1 would be clearly interesting, it would take a massive amount of additional work and resources and, if successful, the data would likely warrant a separate manuscript. Thus, we hope the reviewer agrees that such an endeavour is clearly beyond the scope of this manuscript. It is also important to note, that it is not the sheer amount/abundance of a protein or a specific PTM that is defining its relevance in the cell.

Nonetheless, we have carried out extensive additional experiments in order to assess both i) the occurrence of endogenous modification sites of H1.2 and ii) the relative abundance of H1.2-Ub variants in cells. We performed immunoprecipitation experiments to enrich endogenous H1.2 from HEK 293T cells and subjected the samples to LC-MS/MS analysis to identify endogenous H1.2 modification sites (see below new Extended Data Fig. 1 and Extended Data Table 1).

We identified four ubiquitylation sites in the NTD and the GD of H1.2, including K17 and K64. However, probably due to the high lysine content (~ 40 %) of the CTD resulting in low sequence coverage, no ubiquitylation site was detected in the CTD of H1.2. Additionally, we detected multiple acetylation, phosphorylation and methylation sites in H1.2, confirming both the relevance of the H1.2 KxUb variants investigated in this study and their co-occurrence with other relevant PTMs in the cell and our experimental system.

For quantification of the relative abundance of ubiquitylated H1.2 *versus* non-ubiquitylated H1.2, we counted the numbers of peptide spectrum matches of peptides with the GG or LRGG signature motif and peptides without the respective motives. However, it is important to note that this is rather a rough estimate as quantification of different peptides *via* LC-MS/MS is limited by variations in ionization behavior for different peptides (such as ubiquitylated *versus* non-ubiquitylated peptides). Overall, around 5 % of identified peptides were found to bear the GG or LRGG signature peptides, indicating that roughly 5 % of the cellular pool of H1.2 are likely to be ubiquitylated, most prominently at positions K17 and K64. Our additional data

therefore clearly demonstrates that the H1.2 KxUb variants investigated represent a significant part of the cellular H1.2 pool.

We have therefore amended the manuscript in the Results section (page 4):

“These positions had been identified to be ubiquitylated in several previous studies ⁴⁵⁻⁵⁰ (note that ubiquitylation of endogenous H1.2 at positions K17 and K64 was confirmed in HEK 293T cells; Extended Data Fig. 1 and Extended Data Table 1).”

We have included the results of these additional experiments in the new Extended Data Fig. 1 and Extended Data Table 1:

Extended Data Fig. 1: Endogenous modification sites of H1.2.

Schematic depiction of modification sites in H1.2 after IP-enrichment from HEK 293T cells and identification by LC-MS/MS. Positions investigated within this study are indicated in bold (K17, K64 and K206). Probably due to the high lysine content (40%) of the CTD resulting in a relatively low sequence coverage, no ubiquitylation site was detected in the CTD of H1.2.

Extended Data Table 1. Modification sites of endogenous H1.2.

Detected modifications of endogenous H1.2 after IP-enrichment are indicated within the identified peptide sequence: [-M] loss of initial methionine, (Ox) oxidation of methionine, (Ac) acetylation of lysine or protein N-terminus, (GG) or (LRGG) ubiquitylation, (Ph) phosphorylation, (Me) methylation. Only PTM-containing peptides are shown.

Peptide sequence	Precursor m/z [Da]	Charge	MH+(ex) [Da]	MH+(calc) [Da]	ΔM [ppm]
M (Ox)SETAPAAPAAAPPAEK(GG)	870.41443	2+	1739.82158	1739.82688	-3.04
M (Ox) S (Ph)ETAPAAPAAAPPAEK(LRGG)	696.99860	3+	2088.98124	2088.7838	1.37
[-M]SETAPAAPAAAPPAEK	739.87659	2+	1478.74590	1478.74855	-1.79
[-M](Ac)- SET APAAPAAAPPAEK	760.88397	2+	1520.76067	1520.75911	1.02
[-M] S (Ph)ETAPAAPAAAPPAEK	520.23877	3+	1558.70176	1558.71488	-8.42
[-M]SET(Ph)APAAPAAAPPAEK	779.85529	2+	1558.70329	1558.71488	-7.43
[-M](Ac)- SET (Ph)APAAPAAAPPAEK	800.86456	2+	1600.72185	1600.72544	-2.25
[-M](Ac)- SET APAAPAAAPPAEK(Ac)	781.88147	2+	1562.75566	1562.76968	-8.97
MSETAPAAPAAAPPAEKAPVK(Ac)	683.02234	3+	2047.05246	2047.05285	-0.19
[-M](Ac)- SET APAAPAAAPPAEKAPVK	958.50702	2+	1916.00676	1916.01237	-2.93
[-M](Ac)- SET APAAPAAAPPAEK(Ac)APVK	979.51355	2+	1958.01982	1958.02293	-1.59

SGVS(Ph)LAALKK(GG)ALAAAGYDVEK(Ac)	1149.59167	2+	2298.17607	2298.17410	0.86
K(Ac)ALAAAGYDVEK	639.33850	2+	1277.66973	1277.67359	-3.03
ALAAAGYDVEK(Me)	561.29419	2+	1121.58110	1121.58372	-2.33
SLVSK(Ac)GTLVQTK(LRGG)GTGASGSFK(LRGG)	477.76913	6+	2861.57842	2861.59557	-5.99
GT(Ph)LVQTK(LRGG)GTGASGSFK(Ac)LNK	800.42303	3+	2399.25455	2399.24425	4.29
GTLVQTK(Ac)GTGASGSFK	790.91498	2+	1580.82268	158.82786	-3.28
K(Me)AGGTK(Ac)PK(Ac)	442.76260	2+	884.51793	884.51999	-2.33
AAKSAAKAVKPKAAK(Ac)PK(Ac)	438.02280	4+	1749.06936	1749.07452	-2.95

We have also updated the Methods section (page 21):

“Identification of endogenous posttranslational modification sites of H1.2 in HEK 293T cells

To identify endogenous posttranslational modifications (PTMs) of H1.2 in HEK 293T cells, H1.2 was enriched by immunoprecipitation followed by identification of potential PTMs and modification sites via LC-MS/MS. In the first step, the primary antibody H1.2 (Abcam) was immobilized (5 µg antibody/12.5 µl beads) on Pierce Protein G Magnetic beads (Thermo Fisher Scientific). After washing, 0.5 mg HEK 293T cell lysate was added, incubated for 2 h at 4 °C and washed with 1x PBS, 2 mM MgCl₂, 1 mM DTT, 100 µM Pefabloc SC, 1 µg/ml Leupeptin, 1 µg/ml Aprotinin, 1x cOmplete Protease Inhibitor Cocktail. Samples were eluted in SDS-PAGE loading dye supplemented with 200 µM DTT and fractionated by SDS-PAGE followed by tryptic digest using trypsin and 2-chloroacetamide as alkylation agent.

Digested and desalted peptides were analyzed on an Orbitrap Fusion Tribrid mass spectrometer coupled to an EASY-nLC 1200 system. Peptides were separated in a 90 min gradient starting from 5% ACN, 0.1% formic acid to 35% ACN, 0.1% formic acid in 70 min, followed by 10 min to 45% ACN, 0.1% formic acid and a washing step of 10 min at 80% ACN, 0.1% formic acid. MS spectra were recorded in the orbitrap at 120000 (at m/z 200) resolution and scan range 300-1500 m/z, automatic gain control ion target value of 4e5 and a maximum injection time of 50 ms. Intensity threshold was set to 5e3, included charge states to 2-7. Exclusion duration was 45 s. Fragmentation was performed by CID in the ion trap using 35% collision energy. The automatic gain control was set to 2e3 with a maximum injection time of 300 ms. The system was operated in data dependent top-speed mode with 3 s cycle time. All samples were measured in technical duplicates.

Samples were analyzed with Proteome Discoverer 2.2.0.388 (Thermo Fisher Scientific). Dynamic modifications were set to oxidation (M), acetylation and methylation (K), attachment of GG or LRGG (K) and phosphorylation (S, T or Y). Possible protein N-terminal modifications included were acetylation, M-loss or M-loss with acetylation. Modification site-probability threshold was set to 75%. Results were further filtered for no more than three modifications per peptide and only unambiguous PSMs were considered.”

2. For the interactome presented in Fig. 2, I am missing some statistical analysis to quantify “site-specificity”. I understand that all candidates in the heatmap in Fig. 2B are significant interactors, but how significant are the differences between interactions with different H1KxUbs? The fold-enrichment between the individual H1KxUbs look pretty similar to me, in all of the six groups, which would mean that the site-specificity is moderate. Furthermore, are these interactions also seen in IPs of endogenous H1? This would be reassuring as it would imply that a significant H1 pool carries the respective KxUb modifications and that the interactions also are present in cells.

As mentioned by the reviewer, we determined the statistically significant interactors in this experiment by ANOVA statistics (Fig. 2B) to account for our experimental setup consisting of six parallel baits/conditions in an optimal way. Intensities were normalized by Z-scoring and averaged. As additional multiple comparisons tests for pairwise comparisons (e.g. Tukey’s or Dunnett’s multiple comparisons test) result in large and convoluted tables due to the size of the dataset, a Z-score cut-off was set as a threshold to define relative enrichment between samples. If needed for further categorization, e.g. site-specificity of interactors for the different H1.2 KxUb variants, proteins with a minimum Z-score of 0.3 ($Z < 0.3$ representing 62% of the population) were considered for further analysis within their respective groups (see also Methods (page 22) and Supplementary Table 1). Z-scores (also called standard scores) are a measure of how many standard deviations a raw score is below or above the population mean, thereby representing the relationship of a value to the mean. However, it is important to note that this indeed moderate significance is only referring to the overall groups presented in Fig. 2, whereas for select interactors, as for example the interactors studied in more detail (i.e. USP15, USP13, UCHL5 and SIRT1) relative enrichment patterns are highly significant based on ANOVA statistics and Tukey’s multiple comparisons test as shown in Fig. 3A and C and Extended Data Fig. 5.

As suggested by the reviewer, we also validated some selected interactions from Fig. 2 with endogenous proteins in the cell by co-immunoprecipitation using an anti-H1.2 antibody and western blot analysis.

Extended Data Fig. 3:

(D) Co-immunoprecipitation analysis of interactors of endogenous H1.2 visualized by western blot analysis. Lysate indicates HEK 293T cell lysate as input, IgG indicates the control with an unspecific IgG antibody and H1.2 experiments with the anti-H1.2 antibody. SN marks supernatant.

We added these results in Extended Data Fig. 3D and amended the manuscript accordingly (page 5):

“Finally, selected interactors co-immunoprecipitated with endogenous H1.2 (Extended Data Fig. 3D) further confirming our AP-MS results.”

And Methods section (page 23):

“For the co-immunoprecipitation analysis of potential interactors, endogenous H1.2 was enriched from HEK 293T cells as described above. As control sample, an IgG isotype control antibody (Thermo Fisher Scientific) was immobilized. If probed for CDC27, beads were additionally incubated before elution in 1x PBS, 1 mM MnCl₂, lambda protein phosphatase, 1x cOmplete Protease Inhibitor Cocktail (Roche) for 40 min at RT. Finally, elution fractions were analyzed by western blot and Ponceau S staining. For western blot, primary antibodies were directed against H1 (Santa Cruz), CDC27 (in-house generation), PPP6R3 (Novus Biologicals), SIRT1 (Merck), NPM1 (Abcam) and p150 (BD Transduction Laboratories).”

3. The model that H1.2 K64Ub binds SIRT1 and regulates its activity should be confirmed in cells, otherwise it is difficult to judge how relevant the observed “slight” decrease in SIRT1 activity is. For example, recombinant H1.2 K64Ub or a plasmid encoding a H1.2 K64 mutant that cannot be ubiquitylated could be brought into cells, and the acetylation level of SIRT1 targets could be monitored.

SIRT1 is a relatively broad deacetylase and targets various substrate proteins such as histones and the tumor suppressor p53. The other way around, it cannot be excluded that SIRT1 substrates are also targets of other deacetylases. Introduction of recombinant H1.2 K64Ub into cells would be rather cumbersome and we do not see how this would allow us to quantitatively assess the acetylation status of SIRT1 substrates. Similarly, overexpression of an H1.2 mutant may cause severe cellular alterations hampering the significance of such an experiment. Furthermore, although mainly studied in this context, SIRT1 was also found as an interactor of H1.2 K17Ub and H1.2 K206Ub, indicating its binding to different H1.2-Ub variants. Mutating several lysines of H1.2 might rather hamper its endogenous role in general.

While additional experiments to elucidate how H1.2 K64Ub binds SIRT1 and regulates its activity in cells are therefore beyond the scope of this manuscript, we agree with the reviewer that a deeper understanding of the role ubiquitylation of H1 plays in regulating SIRT1 activity would be desirable.

Nonetheless and as indicated in the original manuscript (“the deacetylation activity of SIRT1 is slightly decreased after incubation with H1.2 K64Ub relative to H1.2”), we agree that the effects on SIRT1 activity – even though clearly reproducible – are mild. To make this more clear, we have further toned down the manuscript and:

Removed “inhibit” from the Abstract (page 2) to write:

“We show that site-specific ubiquitylation of H1 at position K64 modulates interactions with deubiquitylating enzymes and the ~~inhibits~~ deacetylase SIRT1.”

Have changed the respective subheading (page 6) from:

“Ubiquitylation-dependent regulation of acetylation” to “*Ubiquitylation-dependent interaction with SIRT1*”

And the Results (page 6):

“Having established that H1.2 K64Ub forms a distinct complex with SIRT1, we speculated that the enzymatic activity of SIRT1 is *modulated* by H1.2 in a ubiquitylation-dependent manner.”

and changed the Discussion (page 9) from “shows and regulates” to:

“Our data also *suggests* that ubiquitylation of H1.2 at position K64 *affects* the deacetylation capacities of SIRT1, thereby potentially counteracting its transcriptional repressive function.”

4. The authors show that both H1.2 and H1.2 KxUbs can form condensates, which look pretty similar to each other in the microscopy images, but that different numbers of condensates are observed for some of the variants. Could the different numbers be the result of condensates interacting more or less with the microplate? Does the number of condensates decrease if the solution is incubated longer in a tube before pipetting them into a microplate? As liquid condensates are supposed to fuse while they are allowed to come in contact with each other, I expect these parameters to affect the observed numbers, even if condensate formation itself occurred in a very similar way for the different proteins. Do the critical concentrations, at which condensates start to form, differ for the different proteins? This could be assessed by turbidity measurements to show potential differences more clearly.

The reviewer is correct in assuming that interaction of biomolecules with a microplate is possible and exactly for this reason molecules were mixed and directly transferred to a 384-well plate with a special non-binding surface to avoid potential surface interactions (see also Methods section (page 28)). We have also carefully kept conditions comparable between all experiments and have measured and quantified multiple wells at different time points for each of the replicate experiments and see a very consistent and reproducible effect.

We indeed observed an initial decrease in the number of condensates over a period of 60 min accompanied by an increase in droplet size as a result of fusion of multiple droplets and wetting effects on the surface. Therefore number, size and partitioning coefficient for all experiments shown were evaluated 20 min after mixing of molecules, when condensates had stopped moving in Z-direction (i.e. up and down and such out of focus of the microscope) but still retained their spherical shape. More importantly, a filter was applied to reduce noise detection and only objects with a minimum size of 4 pixels and a circularity value ≥ 0.85 were considered as condensates (see Methods for more details). Using these filter criteria, the mean size of monitored condensates remained constant for 60 min and these criteria were therefore applied to determine the influence of site-specific ubiquitylation on condensate formation.

Critical concentrations were also carefully determined and phase diagrams showing droplet formation as a function of salt concentration for both H1.2 and DNA and H1.2 and 12-mer nucleosome arrays evaluated in order to experimentally probe all H1.2 KxUb variants under optimal droplet forming conditions (Extended Data Fig. 9).

Moreover, we have also carried out extensive additional experiments and have repeated all phase separation experiments with H1.2 and each of the three different H1.2 KxUb variants using 12-mer nucleosome arrays (please see the extensive answer to Reviewer 3, new Fig. 5 and Extended Data Fig. 9). We find that all of the effects caused by site-specific ubiquitylation of H1.2 on H1.2-/ H1.2 KxUb-DNA condensates are also seen in condensates of H1.2-/ H1.2 KxUb-nucleosome arrays (see new Fig. 5). All condensates exhibit a similar partition coefficient but for ubiquitylated H1.2 larger condensates (especially with the 12-mer arrays) and most notably more condensates were formed compared to unmodified H1.2 (which again was particularly the case for H1.2 K64Ub).

In summary, we are confident that the difference in numbers between the non-ubiquitylated H1.2 and in between the H1.2 KxUb variants (which only differ in the exact localization of their respective ubiquitylation site but carry the same mass and overall charge) is real and due to their PTM-specific ubiquitylation site.

Minor points

1. The authors claim that H1 PTMs are an understudied aspect of the histone code, which is supposedly due to a lack of appropriate tools. However, H1 behaves fundamentally different from core histones as it dynamically binds to chromosomes (residence time of a few minutes) while core histones stay at their place for hours. Therefore, PTMs on core histones can make up a stable “code”, unlike PTMs on H1 that constantly exchanges between different loci. Thus, I find the analogy misleading, although both core histones and linker histones are called “histones” and one might call their PTMs in principle “histone code”.

We agree with the reviewer that H1 is fundamentally different from core histones in many aspects and that its dynamic behavior with short residence times on the nucleosomes likely precludes the establishment of an inheritable modification pattern. However, when the focus is on histone PTMs in terms of chromatin modulation, variation and the regulation of chromatin structure and function, the linker histone H1 has to be considered as a part of the histone code. Moreover, there is established crosstalk between modifications of core and linker histones (e.g. PMID: 23289424) and we would argue that the dynamic binding behavior does not preclude H1 from recruiting regulatory proteins. Thus, we stand by our opinion that the study of this particular part of the histone code has been significantly hampered by the lack of appropriate tools.

However, in order to meet the concerns of the reviewer, we have amended the manuscript (page 2):

“The linker histone H1 additionally binds at the nucleosome entry and exit sites *in a dynamic manner* to form higher-order chromatin structures.”

“Unlike the core histones, H1 binds dynamically to chromosomes and plays a fundamental role in the formation of higher order chromatin. Yet, although H1 is closely linked to the regulation of DNA structure and dynamics^{1,3-5}, a lack of appropriate technologies, in particular the absence of site- and modification-specific antibodies, has handicapped research on H1 and has significantly hampered our ability to decipher its contribution to the histone code.”

2. The authors claim in the Introduction that it has recently been shown that chromatin and heterochromatin undergo LLPS. Recently, several studies have questioned this view and have outlined the challenges in detecting LLPS in cells, which would be needed to settle this question. I think this point could be described in a more careful and balanced way.

We thank the reviewer for pointing this out and we fully agree with the reviewer that the question if and how chromatin and heterochromatin undergo LLPS in cells is under debate and requires additional investigation. We changed the wording accordingly (page 2):

“Recent studies indicate that LLPS plays a role in chromatin maintenance and chromatin organization, even though the extent to which these processes are affected by LLPS in cells requires further investigations⁷⁻¹⁰.”

We have additionally cited a recent review that provides an excellent overview of the current positions and the debate (Lyon and Rosen, Nature Reviews Molecular Cell Biology, 2020).

3. I find the strong claims about the “function” of H1 problematic, as the authors conduct only *in vitro* assays and it is not clear to me how much these reflect the “function” of H1 in the cell. I think these claims should be toned down, unless they are supported by functional assays that go beyond an analysis of protein-protein interactions.

We agree with the reviewer that it would be desirable to know more about the function of H1 in the cell. However, results obtained in cellular assays remain frequently at a rather descriptive/correlative level, as too many variables contributing to the eventual outcome are unknown and/or cannot be influenced, and do not provide much insight into the mechanistic/molecular basis of individual processes. Only the careful disassembly and reconstruction of critical parts using pure material under experimentally controllable conditions often allows us to assign specific functions to the molecule of interest on the molecular/mechanistic level. More importantly, there is currently not a single assay available that will enable us to monitor the effects of *site-specific* ubiquitylation in the cell.

We also would respectfully like to point out that we have carried out quite extensive *in vitro* assays that clearly go beyond the mere analysis of protein-protein interactions and, in our opinion, are all suitable to contribute to the elucidation of the function of linker histone H1.2, e.g. DUB assays, phase-separation assays and multiple chromatosome array assays.

Still, in order to meet the reviewer’s concerns, we have altered the title of the manuscript to:

“Site-specific ubiquitylation acts as a regulator of linker histone H1”

REVIEWERS' COMMENTS

Reviewer #1 (Remarks to the Author):

I appreciate the efforts made by the authors to improve the manuscript and to answer my specific questions. It is a technically sound and well executed study which leverages precise chemistry to study the effect of ubiquitination of linker histone.

Three major experiments have been added during revisions:

Extended Data Figure 1: Endogenous modifications sites of H1.2: These mass spec dataset now confirms that that H1.2 is ubiquitinated in cells. The present datasets adds to existing studies (e.g. (<https://doi.org/10.1074/mcp.M111.013284>)) an estimate of abundance of the ubiquitinated fragment was estimated at up to 5%. Since ubiquitination may be a function of specific genomic loci, H1.2 ubiquitination may be considerably abundant in some regions of the genome.

Extended Data Figure 3: Validation of selected interaction candidates by Co-IP of endogenous proteins with H1.2. I think this is reassuring, albeit the results are not super clean (some bands in the IgG, suggesting that washing conditions were kept rather mild.

Figure 5 - phase separation assay with chromatosomes, which nicely validates and extends prior experiments with DNA.

The scope remains focussed on an *in vitro* characterization of possible functional consequences of site-specific ubiquitination of H1.2. From the *in vitro* data, two possible functions are hypothesized: first, H1.2K64ub inhibits SIRT1 and thus H1.2K64ub may promote histone acetylation. Second, H1-containing chromosomes phase-separate differently *in vitro*, suggesting that K64ub antagonizes chromatin compaction *in vitro*. In the absence of functional validation in cells, both of these aspects are speculative and thus despite the beautiful chemistry approach I believe the overall expected impact of the study on the chromatin field is limited.

While the revision experiments have corroborated aspects where there was a clear need for additional validation and extended the phase-separation aspect with a more physiological and relevant assay, the revision have not significantly bolstered the biological conclusions or provided new insights beyond the original assertions regarding SIRT1, in particular:

While it has been confirmed that endogenous H1.2 and SIRT1 can interact above background, the dependence on site-specific ubiquitination of H1.2 has not been further investigated in cells. Below I copied again my specific comments from the first review process:

>>>>>>>>>>>>

– a pulldown of transiently or stable transfected, epitope-tagged H1.2 with the specific K->A or K->R mutation(s) should remove those interactors that require the presence of site-specific ubiquitylation. K->A/R mutations also abrogate any other modification at the site (e.g. acetylation), but since the main and only important point of this experiments is to validate the role of ubiquitylation I believe this is not a problem.

– a pulldown of transiently or stable transfected, epitope-tagged H1.2 with all ubiquitylation sites mutated to A or R should further remove any of the ubiquitylation-specific interactors.

<<<<<<<<<<<<<<<

I don't think these experiment are particularly complex and would at least validate one key conclusion by the authors.

Beyond a mere binding, the originally asserted inhibition of SIRT1 activity by ubiquitinated H1.2 remains questionable:

The authors provided further information regarding the assay in Figure 3F, mainly that the western signal for

acetylated and non-acetylated p53 is dramatically increased upon addition of ubiquitinated H1.2 for unknown reasons. While I feel with the authors that unexplained things happen in biology all the time, I think here is a simple assay where it should be possible to figure out what is going in! The problem is, if the authors cannot explain a 'magic' increase in starting signal of acetylated-p53, how can they be sure that whatever decrease they observe subsequently is in any way comparable to the other conditions?

I wonder if p53 and SIRT1 would not be visible on a coomassie gel as well? And is there more signal for p53 in the Ponceau/Coomassie then in the presence of H1.2 K64Ub? Unfortunately the authors only show a narrow cut-out of the Ponceau stain.

Looking at this Ponceau also made me wonder about the experiment setup. The experiment is done with equimolar amounts of SIRT1 and p53, and a 10fold excess of H1.2. So essentially there is one SIRT1 enzyme molecule per p53 substrate molecule, which seems not adequate to assay catalytic activity of an enzyme. On top of that, adding at 10-fold excess of H1.2 K64Ub may not be an adequate way to measure a physiologic inhibitory effect.

Then there is the general problem of quantifying HRP signal (which is non-linear even before saturating the detector). There seems to be some variability amongst the controls (timepoint 0h, no NAD+, no SIRT1) - all of them should be the same as a baseline for any significance calculations. Also, my impression is that p53 K370 acetylation signal decreases very rapidly only in the presence of H1.2 but not in the control without any H1.2. Can the authors clarify if there is any statistically significant difference in the deacetylation kinetics of H1.2K64Ub versus the control without linker histone. If not, would the hypothesis be that non-ubiquitinated H1.2 stimulates SIRT1 activity but H1.2K64Ub reverses this trend?

So in summary, I think the SIRT1 part, which is a major part of the manuscript as is, lacks a clear trajectory towards functional relevance in cells.

Reviewer #2 (Remarks to the Author):

In this revision, the authors have addressed most of the concerns raised from last review. Although explanations in the rebuttal letter were helpful, some should be incorporated into the manuscript for a better presentation. Here are some specific suggestions they can consider:

Explanations on previous Point 5 and 6 provided arguments for the results, however without providing specific evidence. For instance, what is the quality of modeling (or were there any assessments on modeling)? It will help to point out that Fig. 3E is only one of multiple conformational clusters in text. It will be visually helpful to map the crosslinks between SIRT1-H1.2 (maybe as sticks between crosslinked residues) in the three-dimensional structures (Fig. 3D) figure to help visual illustration.

Reviewer #3 (Remarks to the Author):

The authors provide an updated Fig. 5 using 12-mer arrays. The experiments show a small increase in particle size in the presence of H1Ub, along with an increase in particle number in at least the K64Ub case.

In the absence of reliable experiments for the in-cell validation of phase separation / condensate formation, these data do hold greater value than the previous experiments performed with free H1 proteins.

Other points have been sufficiently addressed and I support the publication of the manuscript, providing the concerns of the other reviewers have been addressed.

Reviewer #4 (Remarks to the Author):

The authors have improved the manuscript by adding additional data and toning down some conclusions that I found unjustified. Despite the high quality of the data and although the authors have clearly put a lot of work into this study, I am still left with the question how important H1 ubiquitylation actually is for the cell.

The H1 pool that is ubiquitylated amounts to approx. 5%, and most of the Ub-dependent effects described here (SIRT1 modulation, site-specific interactome, condensate formation) seem pretty mild (albeit significant). While I agree with the authors that it is not the sheer abundance of a PTM that defines its relevance in the cell, I disagree with the authors that it is "only the careful disassembly and reconstruction of critical parts using pure material under experimentally controllable conditions" that will define this relevance. I agree that experiments in cells are oftentimes descriptive and correlative, and that a system of purified components has its advantages. However, I find that prior knowledge from cells will be critical to decide how to set up the purified system and how to interpret the outcome. For example, the authors promote a model in which H1 forms condensates in the cell, but they did not determine if the interaction partners identified here actually localize in these condensates - they might localize to a different set of condensates that reside somewhere else in the cell and that are immiscible with H1 condensates, so that the interactions seen here might never (or rarely) occur in the cell. Preparation of a cell lysate for IP-MS studies, as done here by the authors, will arguably destroy the compartmentalization of proteins into distinct condensates that is supposedly found in the cell.

In summary, I find that the manuscript would be much stronger if the authors complemented their in vitro data with a cell-based assay that shows how important (or not) H1 ubiquitylation is in the cell. One way to do this with state-of-the-art technology would be to replace the respective residue(s) in the endogenous H1 gene using CRISPR/Cas9, and to check if there is a phenotype. Expressing non-conjugatable mutants in cells has taught us a great deal about the relevance of ubiquitylation (and PTMs in general) after all. Despite the high quality of the study, I doubt that the paper in its present form appeals to the broad readership of a multi-disciplinary journal like Nature Communications.

Point-by-point response to the reviewers' comments for "site-specific ubiquitylation is a general functional regulator of linker histone H1" (NCOMMS-20-36862A)

REVIEWERS' COMMENTS

Reviewer #1 (Remarks to the Author):

I appreciate the efforts made by the authors to improve the manuscript and to answer my specific questions. It is a technically sound and well executed study which leverages precise chemistry to study the effect of ubiquitination of linker histone. Three major experiments have been added during revisions: Extended Data Figure 1: Endogenous modifications sites of H1.2: These mass spec dataset now confirms that that H1.2 is ubiquitinated in cells. The present datasets adds to existing studies (e.g. (<https://doi.org/10.1074/mcp.M111.013284>)) an estimate of abundance of the ubiquitinated fragment was estimated at up to 5%. Since ubiquitination may be a function of specific genomic loci, H1.2 ubiquitination may be considerably abundant in some regions of the genome.

Extended Data Figure 3: Validation of selected interaction candidates by Co-IP of endogenous proteins with H1.2. I think this is reassuring, albeit the results are not super clean (some bands in the IgG, suggesting that washing conditions were kept rather mild. Figure 5 - phase separation assay with chromatosomes, which nicely validates and extends prior experiments with DNA.

The scope remains focussed on an in vitro characterization of possible functional consequences of site-specific ubiquitination of H1.2. From the in vitro data, two possible functions are hypothesized: first, H1.2K64ub inhibits SIRT1 and thus H1.2K64ub may promote histone acetylation. Second, H1-containing chromosomes phase-separate differently in vitro, suggesting that K64ub antagonizes chromatin compaction in vitro. In the absence of functional validation in cells, both of these aspects are speculative and thus despite the beautiful chemistry approach I believe the overall expected impact of the study on the chromatin field is limited.

While the revision experiments have corroborated aspects where there was a clear need for additional validation and extended the phase-separation aspect with a more physiological and relevant assay, the revision have not significantly bolstered the biological conclusions or provided new insights beyond the original assertions regarding SIRT1, in particular:

While it has been confirmed that endogenous H1.2 and SIRT1 can interact above background, the dependence on site-specific ubiquitination of H1.2 has not been further investigated in cells. Below I copied again my specific comments from the first review process:

>>>>>>>>>>>>>>

– a pulldown of transiently or stable transfected, epitope-tagged H1.2 with the specific K->A or K->R mutation(s) should remove those interactors that require the presence of site-specific ubiquitylation. K->A/R mutations also abrogate any other modification at the site (e.g. acetylation), but since the main and only important point of this experiments is to validate the role of ubiquitylation I believe this is not a problem.

– a pulldown of transiently or stable transfected, epitope-tagged H1.2 with all ubiquitylation sites mutated to A or R should further remove any of the ubiquitylation-specific interactors.

I don't think these experiment are particularly complex and would at least validate one key conclusion by the authors.

Beyond a mere binding, the originally asserted inhibition of SIRT1 activity by ubiquitinated H1.2 remains questionable:

The authors provided further information regarding the assay in Figure 3F, mainly that the western signal for acetylated and non-acetylated p53 is dramatically increased upon addition of ubiquitinated H1.2 for unknown reasons. While I feel with the authors that unexplained things happen in biology all the time, I think here is a simple assay where it should be possible to figure out what is going in! The problem is, if the authors cannot explain a 'magic' increase in starting signal of acetylated-p53, how can they be sure that whatever decrease they observe subsequently is in any way comparable to the other conditions?

I wonder if p53 and SIRT1 would not be visible on a coomassie gel as well? And is there more signal for p53 in the Ponceau/Coomassie than in the presence of H1.2 K64Ub? Unfortunately the authors only show a narrow cut-out of the Ponceau stain.

Looking at this Ponceau also made me wonder about the experiment setup. The experiment is done with equimolar amounts of SIRT1 and p53, and a 10fold excess of H1.2. So essentially there is one SIRT1 enzyme molecule per p53 substrate molecule, which seems not adequate to assay catalytic activity of an enzyme. On top of that, adding at 10-fold excess of H1.2 K64Ub may not be an adequate way to measure a physiologic inhibitory effect. Then there is the general problem of quantifying HRP signal (which is non-linear even before saturating the detector). There seems to be some variability amongst the controls (timepoint 0h, no NAD⁺, no SIRT1) - all of them should be the same as a baseline for any significance calculations. Also, my impression is that p53 K370 acetylation signal decreases very rapidly only in the presence of H1.2 but not in the control without any H1.2. Can the authors clarify if there is any statistically significant difference in the deacetylation kinetics of H1.2K64Ub versus the control without linker histone. If not, would the hypothesis be that non-ubiquitinated H1.2 stimulates SIRT1 activity but H1.2K64Ub reverses this trend?

So in summary, I think the SIRT1 part, which is a major part of the manuscript as is, lacks a clear trajectory towards functional relevance in cells.

We are grateful to hear that the reviewer finds our study "a technically sound and well executed study which leverages precise chemistry to study the effect of ubiquitination of linker histone". We are also pleased to see that the reviewer acknowledges that we validated some of the identified interactions in cells in Co-IP experiments with endogenous H1.2 and assessed both the occurrence of endogenous modification sites of H1.2 and the relative abundance of H1.2 KxUb variants in cells during this revision in order to know more about the PTM-specific function(s) of H1 in the cell.

As pointed out by the reviewer, we have also extended our characterization of phase separation behavior to 12-mer nucleosome arrays and find that all of the effects caused by site-specific ubiquitylation of H1.2 on histone/DNA condensates are also seen in condensates of intact chromatosomes. While the latter set of experiments is arguably still not within cells, we hope that the reviewer agrees that these experiments further close the gap between a molecular *in vitro* and a cellular *in vivo* function for H1-Ub.

Despite this progress, the reviewer is apparently not fully satisfied, but we respectfully disagree with the statement "the revision have not significantly bolstered the biological conclusions or provided new insights beyond the original assertions regarding SIRT1", in particular for the following reasons.

- 1) We are convinced that the careful disassembly and reconstruction of critical parts of the cellular machinery using pure material under experimentally controllable conditions is best suited to assign specific functions to a protein of interest *on a molecular and mechanistic level*, thereby constituting an essential basis for understanding/elucidating biological processes.

2) Results obtained in pulldown experiments with ectopically expressed H1 mutants will be difficult to interpret, as the H1 mutants would be most likely integrated into chromatin and, thus, endogenous H1 that can still be ubiquitylated would be coprecipitated. This is one of the reasons why we decided against doing such experiments.

3) Regarding the interaction of H1 with SIRT1. We are happy to see that the reviewer now agrees that our data shows that SIRT1 is indeed a *bona fide* interactor of H1.2 *in cellulo*. We also want to reiterate that the main message of the results shown in Fig. 3, that *site-specific ubiquitylation of H1.2 at K64 alters its interaction with SIRT1*, was clearly shown by XL-MS and modeling data.

For the remaining question, if ubiquitylated H1 is also able to modulate the “deacetylation activity of SIRT1”, we want to emphasize: levels of p53 were controlled and identical within biological replicates (as acetylated p53 was included in the master mix). The amounts of p53 and SIRT1 used in these assays are too low to be visualized by Coomassie or Ponceau staining. Also, while the observed effect has been moderate, as stated by us throughout the manuscript, assays have been performed in biological replicates and for two different acetylation-variants, all showing the same trend.

More importantly, this remaining issue does clearly not represent “a major part of the manuscript” (e.g. - we are arguing about one panel of one figure (Figure 3F) of a manuscript consisting in total of 5 Figures and 22 panels).

Taken together, we feel that our data constitutes an important step forward in the quest for a comprehensive understanding of the role of H1 in the histone code. Our study also serves as a general example of how chemical protein conjugation, proteomic profiling and biochemical characterization can be deployed in a powerful approach to dissect the PTM-specific modular proteome and to study its functional consequences on a molecular level.

Reviewer #2 (Remarks to the Author):

In this revision, the authors have addressed most of the concerns raised from last review. Although explanations in the rebuttal letter were helpful, some should be incorporated into the manuscript for a better presentation. Here are some specific suggestions they can consider: Explanations on previous Point 5 and 6 provided arguments for the results, however without providing specific evidence. For instance, what is the quality of modeling (or were there any assessments on modeling)? It will help to point out that Fig. 3E is only one of multiple conformational clusters in text. It will be visually helpful to map the crosslinks between SIRT1-H1.2 (maybe as sticks between crosslinked residues) in the three-dimensional structures (Fig. 3D) figure to help visual illustration.

We are happy to hear that we were able to address the concerns of the reviewer and respectfully point out that all the explanations in the rebuttal were already incorporated in the revised manuscript.

1) There are several approaches we incorporated to ensure a high quality of the resulting models (see also methods). First, all modeling runs were run with 32 replicas over a temperature scale resulting in a wide coverage of the model space which was even further increased by starting with initially random configurations. Every modeling run was also

performed in triplicates and only the 25% best scoring models were saved for further analysis. For our final clusters the models from these three independent runs were pooled. In addition, we also ensured that each replicate converged onto the same clusters as the pooled models.

2) We have indicated at multiple occasions in the manuscript that Fig. 3E is only one of multiple conformational clusters:

- When referring to Fig. 3E, we indicate that there is more than one possible localization for Ub, i.e. page 6 of the manuscript: "While our models show more than one possible localization for Ub within the H1.2 K64Ub: SIRT1 complex, suggesting multiple conformational states ...".
- We also note in the figure legend of Fig. 3E that the second cluster for H1.2 K64Ub is found in Fig S7.
- Fig S7 then shows that we have indeed identified two main clusters for the H1.2 K64Ub: SIRT1 complex, which is additionally depicted by its RMSD matrix in Fig S7B.

While the reviewer is certainly correct in pointing out that visualizing the crosslinks may help in understanding the cluster localization of our model, we have decided against showing them because of the sheer number of crosslinks involved. For example, for the model containing Ub we have identified 139 unique crosslinks, resulting in an overall confusing picture if all of them were to be visualized.

Reviewer #3 (Remarks to the Author)

The authors provide an updated Fig. 5 using 12-mer arrays. The experiments show a small increase in particle size in the presence of H1Ub, along with an increase in particle number in at least the K64Ub case.

In the absence of reliable experiments for the in-cell validation of phase separation / condensate formation, these data do hold greater value than the previous experiments performed with free H1 proteins.

Other points have been sufficiently addressed and I support the publication of the manuscript, providing the concerns of the other reviewers have been addressed.

We thank the reviewer again for the comments on the original version which really helped to improve the quality of our manuscript and are pleased to hear that the reviewer now supports publication of our manuscript.

Reviewer #4 (Remarks to the Author)

The authors have improved the manuscript by adding additional data and toning down some conclusions that I found unjustified. Despite the high quality of the data and although the authors have clearly put a lot of work into this study, I am still left with the question how important H1 ubiquitylation actually is for the cell.

The H1 pool that is ubiquitylated amounts to approx. 5%, and most of the Ub-dependent effects described here (SIRT1 modulation, site-specific interactome, condensate formation) seem pretty mild (albeit significant). While I agree with the authors that it is not the sheer abundance of a PTM that defines its relevance in the cell, I disagree with the authors that it is "only the careful disassembly and reconstruction of critical parts using pure material under experimentally controllable conditions" that will define this relevance. I agree that experiments in cells are oftentimes descriptive and correlative, and that a system of purified components has its advantages. However, I find that prior knowledge from cells will be critical to decide

how to set up the purified system and how to interpret the outcome. For example, the authors promote a model in which H1 forms condensates in the cell, but they did not determine if the interaction partners identified here actually localize in these condensates - they might localize to a different set of condensates that reside somewhere else in the cell and that are immiscible with H1 condensates, so that the interactions seen here might never (or rarely) occur in the cell. Preparation of a cell lysate for IP-MS studies, as done here by the authors, will arguably destroy the compartmentalization of proteins into distinct condensates that is supposedly found in the cell.

In summary, I find that the manuscript would be much stronger if the authors complemented their *in vitro* data with a cell-based assay that shows how important (or not) H1 ubiquitylation is in the cell. One way to do this with state-of-the-art technology would be to replace the respective residue(s) in the endogenous H1 gene using CRISPR/Cas9, and to check if there is a phenotype. Expressing non-conjugatable mutants in cells has taught us a great deal about the relevance of ubiquitylation (and PTMs in general) after all. Despite the high quality of the study, I doubt that the paper in its present form appeals to the broad readership of a multi-disciplinary journal like Nature Communications.

We are grateful to hear that the reviewer finds our manuscript improved and our data once again of high quality.

We also agree with the reviewer that it would be great to know more about the PTM-specific function(s) of H1 in the cell. However, we would like to point out that we validated some of the identified interactions in cells in Co-IP experiments with endogenous H1 and assessed both the occurrence of endogenous modification sites of H1.2 and the relative abundance of H1.2 KxUb variants in cells during this revision. Moreover, we have extended our characterization of phase separation behavior to 12-mer nucleosome arrays and find that that all of the effects caused by site-specific ubiquitylation of H1.2 on histone/DNA condensates are also seen in condensates of intact chromatosomes. While the latter set of experiments is not within cells, we hope that the reviewer agrees that these experiments further close the gap between a molecular *in vitro* and a cellular *in vivo* function for H1-Ub.

More importantly, there is not a single assay available that would have enabled us to monitor PTM-specific interactions of H1 in the cell, left alone in condensates. Likewise, replacing the respective residue(s) in the endogenous H1.2 gene using CRISPR/Cas9, as suggested by the reviewer, will not tell us much, if anything at all, about the effect that *site-specific ubiquitylation* has on H1.2 function (e.g., lysine residues are subject to several PTMs; thus, even if a phenotype could be observed with a respective H1 mutant, it would not be possible to attribute it to H1 ubiquitylation). Thus, we feel that we have done everything that could be reasonably expected.

Taken together, we feel that our data constitutes an important step forward in the quest for a comprehensive understanding of the role of H1 in the histone code. Our study also serves as a general example of how chemical protein synthesis, proteomic profiling and biochemical characterization can be deployed in a powerful approach to dissect the PTM-specific modular proteome and to study its functional consequences on a molecular level.